# Migration of CD8 + TSCM cells into intestine via PPBP–CXCR2 axis increases host stress susceptibility by inhibiting gut microbiome-derived homovanillic acid

Yuan Zhang[1,11], Minzi Ju[1,11], Suzhen Chen[2,3,11], Wendi Yang[1,11], Yang Cai[1], Xiaoyu Yu[1], Gang Chen[2,3,4], Zhongxia Shen[2,3,5], Ying Bai[1], Hui Ren[1], Yinghui Li[2,3], Ling Shen[1], Junxu Li [6], Peng Shi [7], Yonggui Yuan [2,3] ✉, Bing Han [1,8] ✉ & Honghong Yao [1,9,10] ✉

Psychosocial stress impacts immune system and brain function, yet mechanisms linking peripheral immune dysregulation to major depressive disorder remain unclear. Here, we demonstrate that a specific subset of T cells, the stem cell-like memory CD8+ T ($T_{SCM}$) cells, is elevated in patients and stress-susceptible mice. CD8+ $T_{SCM}$ cells from patients display unique transcriptional programs and correlated with depression severity. Adoptive transfer of stress-derived CD8+ $T_{SCM}$ cells induced depressive-like behavior and neuroinflammation in recipients, without brain migration. Employing a whole-body immunolabeling technology, we discover CD8+ $T_{SCM}$ cells migrated to intestine via the interaction of pro-platelet basic protein and C-X-C motif chemokine receptor 2. CD8+ $T_{SCM}$ cells decrease the abundance of tyrosine-metabolizing bacteria to reducing homovanillic acid production, triggered neuroinflammation and depressive symptoms. Thus, our findings uncover a complex interplay between CD8+ $T_{SCM}$ cells and gut microbial metabolism, shedding light on potential mechanisms underlying depression and suggesting avenues for therapeutic intervention.

Neuropsychiatric disorders precipitated by stress, including major depressive disorder (MDD), have global prevalence and impose a significant individual burden[1-3]. Chronic psychosocial stress stands out as a significant risk factor for the development of depression[4]. Despite the availability of efficacious therapeutic interventions for MDD[3], a substantial fraction (over one-third) of afflicted individuals fail to attain complete remission following treatment with conventional antidepressant medications or established psychotherapies[5-7].

[1]Department of Pharmacology, Jiangsu Provincial Key Laboratory of Critical Care Medicine, School of Medicine, Southeast University, Nanjing, Jiangsu, China. [2]Department of Psychosomatics and Psychiatry, Zhongda Hospital, School of Medicine, Jiangsu Provincial Key Laboratory of Brain Science and Medicine, Southeast University, Nanjing, Jiangsu, China. [3]Institute of Psychosomatics, School of Medicine, Southeast University, Nanjing, Jiangsu, China. [4]Department of Psychiatry, the Third People's Hospital of Huai'an, Huai'an, Jiangsu, China. [5]Department of Psychosomatics, the Third People's Hospital of Huzhou, Huzhou, Zhejiang, China. [6]Department of Pharmacology and Toxicology, University at Buffalo, Buffalo, NY, USA. [7]Department of Cardiology of the Second Affiliated Hospital, Zhejiang University School of Medicine, Hangzhou, Zhejiang, China. [8]Southeast University Affiliated Liangjiang Hospital, Nanjing Pukou People's Hospital, Southeast University, Nanjing, China. [9]Co-innovation Center of Neuroregeneration, Nantong University, Nantong, Jiangsu, China. [10]Institute of Life Sciences, Key Laboratory of Developmental Genes and Human Disease, Southeast University, Nanjing, Jiangsu, China. [11]These authors contributed equally: Yuan Zhang, Minzi Ju, Suzhen Chen, Wendi Yang. ✉e-mail: 101011406@seu.edu.cn; hanb@seu.edu.cn; yaohh@seu.edu.cn

Consequently, elucidating the intricate pathophysiological mechanisms that underpin the impact of psychosocial stress is imperative for advancing our comprehension of disorders like MDD. This understanding is foundational for emerging treatment modalities and preventive strategies.

Owing to the intricate bi-directional communication between the central nervous system (CNS) and the immune system, recent research has underscored the significance of immune dysregulation in the etiology and progression of depression[8,9]. Accumulating evidence points to T lymphocytes as key players in the pathology of depression[8]. A previous study examined changes in lymphocyte subsets among MDD patients and determined that CD8+ T cell levels were elevated compared to those in healthy controls[10]. Remarkably, MDD patients unresponsive to antidepressant therapy presented with elevated counts of CD8+ cells relative to those who responded favorably to treatment[11]. Additionally, another study demonstrated that perforin produced by colonic CD8+ T cells stimulates CXCL9 production in intestinal epithelial cells, which subsequently induces neuronal endoplasmic reticulum stress in the hippocampus, contributing to depression-like symptoms in inflammatory bowel disease[12]. Moreover, the proportion of effector memory CD8+ T ($T_{EM}$) cells was observed to be heightened in MDD patients and was positively correlated with sleep disruptions[13]. Despite these insights indicating the involvement of peripheral immune cells in the pathophysiology of depression, a comprehensive understanding of immune cell alteration, particularly that of CD8+ T cells, and their interplay with MDD remains lacking.

Additionally, it is widely recognized that the intestinal microbiota plays a crucial role in shaping the host immune system, as well as brain functions, including cognition and mood[14,15]. In the interaction between peripheral immunity, gut microbiota, and brain function, microbial metabolites emerge as a critical intermediate. Various bacterial compounds, predominantly from the small intestine, have been found to extensively permeate to blood, with microbially-derived amino acids even penetrating the CNS[16,17]. Therefore, changes in gut microbiota metabolites can affect peripheral immunity and directly or indirectly influence the pathogenesis and pathological progression of depression[18–20]. However, whether alterations in peripheral immunity resulted from depression can influence microbiota metabolism remains elusive at present.

In this study, using single-cell RNA sequencing (scRNA-seq) coupled with flow cytometric analysis, we identified stem cell-like memory CD8+ T ($T_{SCM}$) cells as a pivotal subset driving depression pathology. Subsequent nanobody($V_HH$)-boosted 3D imaging of solvent-cleared organs (vDISCO), a pressure-driven, nanobody-based whole-body immunolabeling technology[21], visualized the migration of CD8+ $T_{SCM}$ cells to the intestine via the pro-platelet basic protein (PPBP) and its receptor C-X-C motif chemokine receptor 2 (CXCR2) axis, resulting in a decrease of microbiota associated with tyrosine metabolism and a reduction of homovanillic acid (HVA). This cascade triggers neuroinflammation within the brain and precipitates depressive phenotypes.

## Results
### Impact of chronic stress on T cells
To validate the immunophenotypic characteristics of peripheral immune cells in individuals with MDD, a rigorous participant screening process was undertaken. 583 potential participants were screened by phone, 296 were assessed for eligibility onsite, and 230 participants (including 115 MDD patients and 115 psychiatrically healthy controls) were meticulously selected and stratified into four distinct cohorts (Supplementary Data 1). In cohort 1, peripheral blood samples were comprehensively analyzed using flow cytometry to discern any MDD-induced modifications in cell populations and phenotypes (Supplementary Fig. 1A and Supplementary Data 2–3). The peripheral immune cell populations exhibited a significantly higher proportion of T cells and a significantly lower proportion of NK cells compared to

psychiatrically healthy controls, whereas the proportions of myeloid cells and B cells showed no significant differences in MDD patients (Fig. 1A and Supplementary Fig. 1B–D). However, only the T cell proportion positively correlated with the severity of depressive symptoms, as measured by the 24-item Hamilton Depression Rating Scale (HAMD-24) scores (Fig. 1B, Supplementary Fig. 1E–G, and Supplementary Data 4). There were also positive correlations between the proportion of T cells and the patients' feelings of despair and cognitive impairment as measured by HAMD-24 (Fig. 1C, D). Similarly, T cell proportion was positively associated with the severity of observer-rated depressive symptoms (Montgomery Asberg Depression Rating Scale, MADRS) and the severity of anhedonia (Snaith-Hamilton Pleasure Scale, SHAPS) (Fig. 1E, F). Additionally, the proportion of T cells negatively correlated with both the total Temporal Experience of Pleasure Scale (TEPS) scores and its anticipatory pleasure subscale (Fig. 1G, H).

To test whether T cells play a role in stress-induced depression, we further examined the proportion of T cells in a mouse depression model of chronic social defeat stress (CSDS) (Fig. 1I). C57BL/6 J mice were classified into susceptible or resilient groups according to their social interaction ratio in social interaction test (SIT) after exposure to CSDS (Fig. 1J). Susceptible mice displayed depressive-like behaviors, as indicated by the reduced sucrose uptake in the sucrose preference test (SPT) and prolonged immobility time in both tail suspension test (TST) and forced swim test (FST) (Fig. 1K–M). Notably, compared with both control and resilient mice, CSDS exposure increased the proportions of T cells in peripheral blood mononuclear cells (PBMCs) of susceptible mice (Fig. 1N), which were negatively correlated with sucrose uptake in SPT and positively correlated with immobility time in both TST and FST (Fig. 1O–Q). Overall, these results confirmed that T cells may contribute to immune system abnormalities in MDD patients, and play a potential role in the pathophysiology of depression.

### CD8+ $T_{SCM}$ cells increased host stress susceptibility
To delineate the comprehensive landscapes of circulating immune cell phenotypes in individuals with MDD, we conducted scRNA-seq using the 10x genomics platform on PBMCs samples obtained from cohort 2 b Fig. 2A and Supplementary Data 5). Through scRNA-seq analysis, we identified nine clusters of T cells, each corresponding to well-defined T cell subsets based previously defined marker genes[22] and the R package *SingleR* (v1.0.1) (Fig. 2B, Supplementary Fig. 2A, B, and Supplementary Data 6). According to the differentially expressed genes (DEGs) analysis, we identified a total of 48 DEGs across various T cell subsets by comparing MDD patients with psychiatrically healthy controls, and 33 of these DEGs were specifically from CD8+ $T_{SCM}$ cells, indicating that these cells in MDD patients exhibit the most significantly distinct transcriptional profile (Fig. 2C and Supplementary Data 7). This unique phenotype was characterized by the activation of highly divergent cell-communication networks, which may contribute to immune system abnormalities in MDD (Supplementary Fig. 2C, D, and Supplementary Data 8).

Given the significant increase in CD8+ $T_{SCM}$ cells observed in the scRNA-seq study (Fig. 2D), we collected peripheral blood samples from a discovery cohort of 39 MDD patients and 39 psychiatrically healthy controls (cohort 3), which was specifically assembled to investigate changes in the proportions of various T cell subsets (Supplementary Fig. 3A and Supplementary Data 9, 10). Flow cytometry revealed significant increases in the proportions of CD8+ $T_{SCM}$, CD8+ $T_{CM}$, and CD8+ $T_{EM}$ cells, and significant decreases in the proportions of CD4+ and CD8+ $T_{Naive}$ cell proportion in blood of MDD patients (Fig. 2E and Supplementary Fig. 3B–I). Among the various T cell subsets evaluated, only the proportions of CD8+ $T_{SCM}$ cells positively correlated with HAMD-24 scores (Fig. 2F and Supplementary Fig. 3J–M). There were also positive correlations between the proportion of CD8+ $T_{SCM}$ cells and the patients' feelings of despair and sleep disturbance as measured

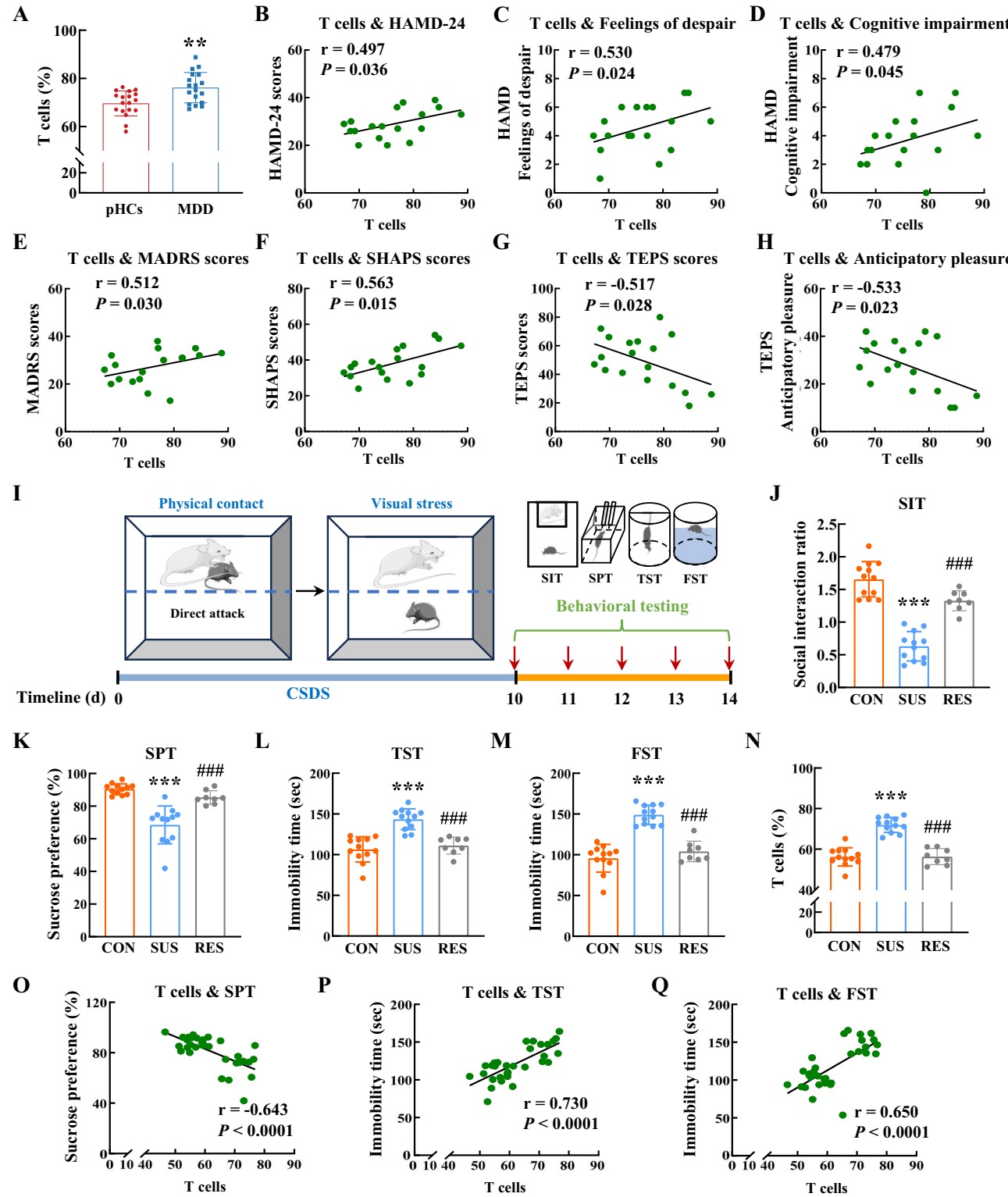

by HAMD-24 (Fig. 2G, H). Positive correlations were also found between the proportion of CD8+ $T_{SCM}$ cells and the severity of observer-rated (MADRS) as well as self-reported (the 9-item Patient Health Questionnaire, PHQ-9) depressive symptoms (Fig. 2I, J). Additionally, CD8+ $T_{SCM}$ cells exhibited positive correlations with Hamilton Anxiety Scale (HAMA) total scores, as well as the scores of physical anxiety and psychological anxiety (Fig. 2K–M and Supplementary Data 11).

Given that these CD8+ $T_{SCM}$ cells appear to be phenotypically and functionally distinct, we hypothesized that their differentiation is

regulated by respective transcriptional programs. Single-cell regulatory network inference and clustering (SCENIC)[23] revealed highly cluster-specific regulon activities and non-overlapping transcription factor profiles (Supplementary Fig. 4A). CD8+ $T_{SCM}$ cells in MDD patients exhibited high activity of several regulons, including XBP1, IRF1, and RUNX3. Since CD8+ $T_{SCM}$ cells are involved in the progression of depression, we next sought to determine the key molecules associated with the distinct ability of CD8+ $T_{SCM}$ cells to drive depression, and 33 specific DEGs were identified in CD8+ $T_{SCM}$ cells from MDD patients by scRNA-seq (Supplementary Fig. 4B). Compared with CD8+

**Fig. 1 | Impact of chronic stress on T cells. A** Proportions of T cells in psychiatrically healthy controls and MDD patients. **\*\*$P$ = 0.0018 versus the pHCs group, $n$ = 18/group. **B** Correlations between T cell proportions and HAMD-24 scores in MDD patients. Correlations between the proportions of T cells with the scores of HAMD-24 factors feelings of despair (**C**) and cognitive impairment (**D**) in MDD patients. $n$ = 18. Correlations between the proportions of T cells with the scores of MADRS (**E**), SHAPS (**F**), TEPS total (**G**) and TEPS-anticipatory pleasure (**H**) in MDD patients. $n$ = 18. **I** Overview of animal experiments of CSDS-induced depression mouse models. Components of this figure were created using Servier Medical Art templates, which are licensed under a Creative Commons Attribution 3.0 Unported License; https://smart.servier.com. **J** CSDS-exposed mice were separated into susceptible or resilient groups according to the social interaction ratio. CON ($n$ = 12), SUS ($n$ = 12), RES ($n$ = 8). **\*\*\*$P$ < 0.0001 versus the CON group, ###$P$ < 0.0001 versus the SUS group. The depressive-like behaviors of control, susceptible or resilient mice after CSDS by the SPT (**K**), TST (**L**), and FST (**M**). CON ($n$ = 12), SUS ($n$ = 12), RES

($n$ = 8). **\*\*\*$P$ < 0.0001 versus the CON group, ###$P$ < 0.0001 versus the SUS group. **N** Proportions of T cells were increased in the susceptible group. CON ($n$ = 12), SUS ($n$ = 12), RES ($n$ = 8). **\*\*\*$P$ < 0.0001 versus the CON group, ###$P$ < 0.0001 versus the SUS group. Correlations of T cell proportions with sucrose uptake in SPT (**O**) and immobility time in both TST (**P**) and FST (**Q**). $P$ < 0.0001 versus the T cell proportions in (**O–Q**), $n$ = 32. **A** Student's $t$ test (two-sided). **J–N** One-way ANOVA followed by two-sided Holm-Sidak pos*t* hoc multiple comparison test. **B–H**, **O–Q** Two-sided Pearson's or Spearman's correlation coefficient. **A**, **J–N** Data were presented as mean ± SD. Source data are provided as a Source Data file. pHCs psychiatrically healthy controls, MDD major depressive disorder, HAMD-24 the 24-item Hamilton Depression Rating Scale, MADRS Montgomery Asberg Depression Rating Scale, SHAPS Snaith-Hamilton Pleasure Scale, TEPS Temporal Experience of Pleasure Scale, CSDS chronic social defeat stress, CON control, SUS susceptible, RES resilient, SIT social interaction test, SPT sucrose preference test, TST tail suspension test, FST forced swim test.

T$_{SCM}$ cells from psychiatrically healthy controls, Kyoto Encyclopedia of Genes and Genomes (KEGG) enrichment analysis revealed that these DEGs from CD8$^+$ T$_{SCM}$ cells in MDD patients were mainly involved in the chemokine signaling pathway (Supplementary Fig. 4C). To further study the interactions among robust DEGs, a visualized PPI network was constructed by the STRING database (https://cn.string-db.org/), which identified 33 nodes and 90 edges (Supplementary Fig. 4D). Subsets of the PPI network showed that certain genes, such as *GZMB*, *GZMH*, and *TBX21*, contributed to cytolysis (Supplementary Fig. 4E). Other genes, including *CX3CR1*, *CCL4*, and *CCL4L2*, contributed to the chemokine signaling pathway (Supplementary Fig. 4F). Next, the interaction network was imported into cytoHubba from Cytoscape software to screen hub genes using 12 algorithms[24]. The interaction network with the top 10 genes with the highest degrees was created (Supplementary Data 12). Moreover, Molecular Complex Detection (MCODE) was also used to generate a significant module containing 9 genes, which included 3 genes, and these genes completely overlapped with the hub genes determined by cytoHubba (Supplementary Fig. 4G). We then validated our hub DEGs using fluorescence-activated cell sorting to separate CD8$^+$ T$_{SCM}$ cells and then performed qPCR to verify that the expression of *TBX21*, *PRF1*, and *GNLY* were significantly increased in CD8$^+$ T$_{SCM}$ cells from MDD patients compared to psychiatrically healthy controls (Supplementary Fig. 4H–J and Supplementary Data 13). These results indicated that *TBX21*, *PRF1*, and *GNLY* might be key molecules that regulate the function of CD8$^+$ T$_{SCM}$ cells in the context of depression.

We further confirmed the proportions of CD8$^+$ T$_{SCM}$ cells in CSDS mouse model, compared with both control and resilient mice, CSDS exposure increased the proportions of CD8$^+$ T$_{SCM}$ cells in PBMCs of susceptible mice (Fig. 2N and Supplementary Data 14). The proportion of CD8$^+$ T$_{SCM}$ cells was negatively correlated with sucrose uptake in SPT and positively correlated with immobility time in TST and FST (Fig. 2O–Q). Additionally, the proportions of CD8$^+$ T$_{SCM}$ cells also increased in mice induced by chronic unpredictable stress (CUS) and correlated with depressive-like behaviors (Supplementary Fig. 5A–H). To investigate the unique properties of pathological CD8$^+$ T$_{SCM}$ cells in driving depression, we separated CD8$^+$ T$_{SCM}$ cells from control and CUS mice by flow cytometry. Among the 33 DEGs identified in human CD8$^+$ T$_{SCM}$ cells from the scRNA-seq study, 20 homologous DEGs between humans and mice were significantly upregulated in CD8$^+$ T$_{SCM}$ cells from CUS mice (Supplementary Fig. 6A). Furthermore, 4 DEGs exhibited no significant changes in CUS mice, and 9 DEGs did not have conserved genes in the mouse species (Supplementary Fig. 6B). This finding suggests a potential role for CD8$^+$ T$_{SCM}$ cells in the pathophysiology of MDD, warranting further investigation into their exact contributions to disease progression and potential therapeutic targeting. Therefore, we next focused on the mechanisms by which peripheral CD8$^+$ T$_{SCM}$ cells can affect neurological function and behavior.

## Pathological CD8$^+$ T$_{SCM}$ cells transmitted depressive symptoms to the host

To determine whether CD8$^+$ T$_{SCM}$ cells induced by depression are pathogenic, CD8$^+$ T$_{SCM}$ cells were isolated from CUS and control mice and adoptively transferred into recombination activating gene 1 (*Rag1*)$^{-/-}$ mice (immunodeficient mice lacking mature T and B lymphocytes). Subsequently, the behavioral phenotypes of the recipient *Rag1*$^{-/-}$ mice were assessed using standard behavioral tests (Fig. 3A and Supplementary Fig. 7A). It is important to note that baseline behavioral assessments of WT and *Rag1*$^{-/-}$ mice, conducted using the SPT, TST, and FST, revealed no significant differences in their behavioral performance (Supplementary Fig. 7B-D). The *Rag1*$^{-/-}$ mice that received CD8$^+$ T$_{SCM}$ cells isolated from stressed mice induced by CUS model exhibited depressive-like behaviors, as indicated by reduced sucrose uptake in the SPT and prolonged immobility time in the TST and FST (Fig. 3B-D). In order to further confirm the role of CD8$^+$ T$_{SCM}$ cells in the transmission of depressive behavior, the adoptive transfer experiments were also conducted in WT mice (Supplementary Fig. 7E). Since technical limitations hindered the depletion of CD8$^+$ T$_{SCM}$ cells, we opted to deplete CD8$^+$ T cells through intravenous (i.v.) injection of the anti-CD8 neutralizing antibody (αCD8) in WT mice. The efficiency of depletion was confirmed through flow cytometry analysis (Supplementary Fig. 7F). The CD8$^+$ T cell-depleted WT mice that received pathological CD8$^+$ T$_{SCM}$ cells isolated from stressed mice also displayed depressive-like behaviors in the SPT, TST, and FST (Supplementary Fig. 7G–I). These results suggest that pathological CD8$^+$ T$_{SCM}$ cells retain transcriptional imprints that can induce depressive-like behaviors in CD8$^+$ T cell-depleted WT mice.

Next, we evaluated whether the expansion of pathological CD8$^+$ T$_{SCM}$ cells affect CNS functions. Growing evidence shows that impairments in the normal structure and function of glial cells in the context of neuroinflammation can lead to depression[25,26]. Astrocytes are key players that sense homeostatic disturbances in the CNS, and astrocyte dysfunction is evident in stressed mice[27]. Thus, immunostaining was used to examine astrocytes in the brain of the recipient *Rag1*$^{-/-}$ mice that underwent adoptive transfer of pathological CD8$^+$ T$_{SCM}$ cells, which revealed astrocyte dysfunction (Fig. 3E), as shown by the decreased number of glial fibrillary acidic protein (GFAP)-positive cells (Fig. 3F) and significantly decreased astrocyte branch number, volume, and length (Fig. 3G-I). Immunostaining also revealed that adoptive transfer of pathological CD8$^+$ T$_{SCM}$ cells induced microglial activation in the *Rag1*$^{-/-}$ recipient mice (Fig. 3J), as indicated by increases in the microglial soma size, branch number, branch length, and branch volume (Fig. 3K–N) in the hippocampus of the *Rag1*$^{-/-}$ recipient mice. This is noteworthy because microglia are capable of conducting immune surveillance in various neurologic pathologies[28]. To examine the molecular mechanisms of CD8$^+$ T$_{SCM}$ cells in neuroinflammation in the *Rag1*$^{-/-}$ recipient mice, RNA-seq was performed on hippocampal tissue. RNA-seq results identified 418 DEGs (240 upregulated genes

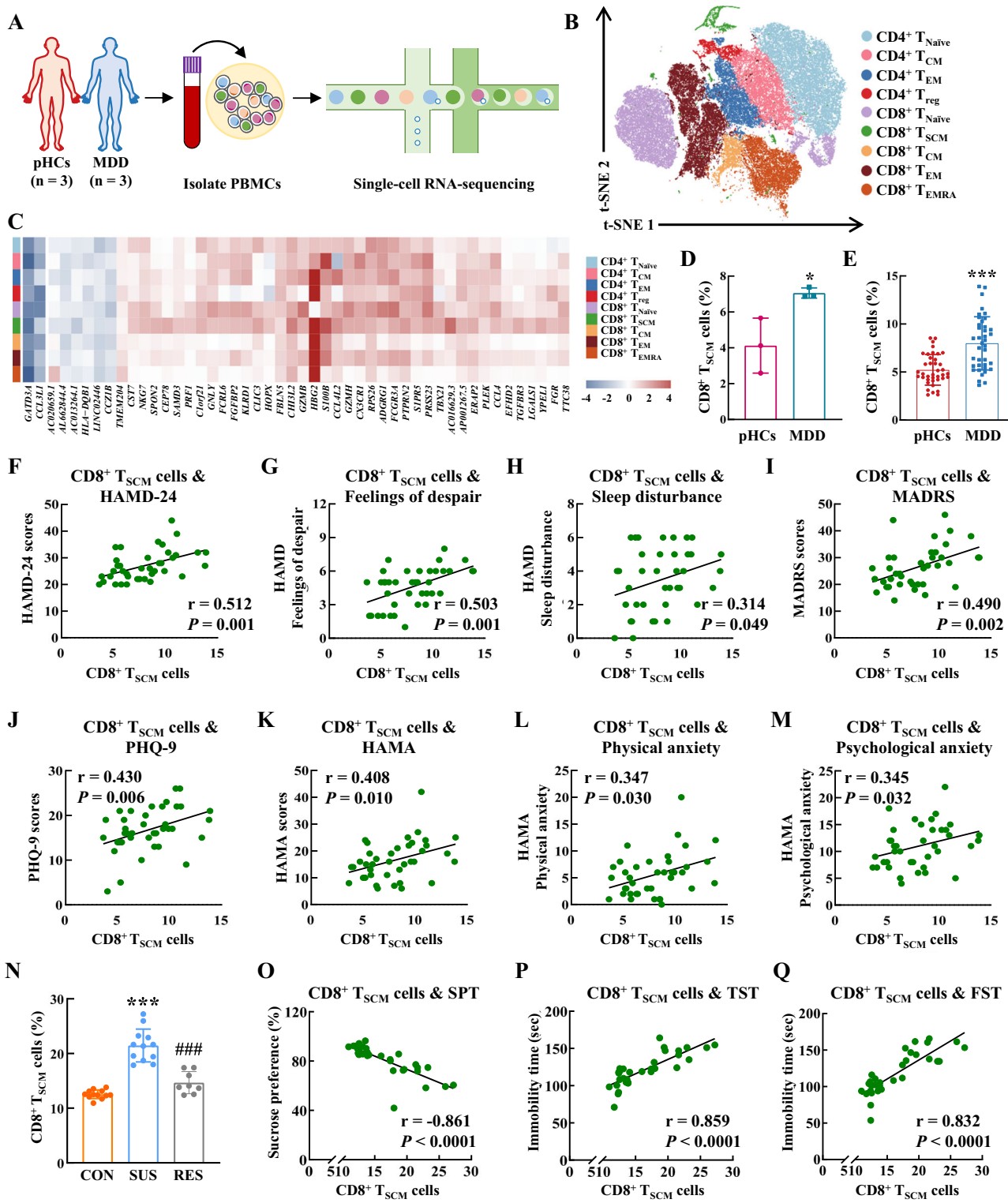

and 178 downregulated genes) in *Rag1*[−/−] recipient mice that received CD8+ T_SCM cells from stressed mice compared with those that received CD8+ T_SCM cells from control mice (Fig. 3O). Using DisGeNET (https://www.disgenet.org/), a publicly available database[29], we compared these 418 DEGs with previously reported findings in MDD and found that 26 genes were associated with MDD-related terms (Fig. 3P and Supplementary Data 15, 16). Next, the KEGG database was used to examine the biological pathways associated with these 26 DEGs (Fig. 3Q). The analysis identified several key pathways implicated in the neuroinflammatory cascade: (i) the NF-κB inflammatory signaling

pathway, TNF signaling pathway, and Ras signaling pathway were significantly enriched, suggesting their roles in driving microglial activation[30,31]. (ii) the phospholipase D signaling pathway and estrogen signaling pathway were enriched, indicating their involvement in astrocyte loss[32,33]. (iii) proinflammatory pathways, including NF-κB, TNF, and chemokine signaling pathways, were associated with compromised blood-brain barrier (BBB) integrity[34,35]. Overall, these findings suggested that CD8+ T_SCM cells are involved in the cascades of neuroinflammation by inducing microglial activation and astrocyte dysfunction, thus inducing depressive-like behaviors.

**Fig. 2 | CD8$^+$ T$_{SCM}$ cells increased host stress susceptibility. A** Schematic of the scRNA-seq workflow. Components of this figure were created using Servier Medical Art templates, which are licensed under a Creative Commons Attribution 3.0 Unported License; https://smart.servier.com. **B** Nine transcriptionally distinct T cell clusters were overlaid on the t-SNE representation. **C** Hierarchical clustering of the log-fold change in DEGs ([log$_2$(fold change)] ≥ 1 and FDR < 0.05) in MDD patients and psychiatrically healthy controls for each cell type. **D** Proportions of CD8$^+$ T$_{SCM}$ cells in pHCs and MDD patients in the scRNA-seq analysis. *$P = 0.0312$ versus the pHCs group, $n = 3$/group. **E** Proportions of CD8$^+$ T$_{SCM}$ cells in pHCs and MDD patients. ***$P < 0.0001$ versus the pHCs group, $n = 39$/group. The proportion of CD8$^+$ T$_{SCM}$ cells correlated with HAMD-24 total scores (**F**), and the factor scores of feelings of despair (**G**) and sleep disturbance (**H**) in MDD patients. $n = 39$. Correlations between CD8$^+$ T$_{SCM}$ cell proportions and the scores of MADRS (**I**) and PHQ-9 (**J**) in MDD patients. $n = 39$. The proportion of CD8$^+$ T$_{SCM}$ cells correlated with HAMA total scores (**K**), and the factor scores of physical anxiety (**L**) and psychological anxiety (**M**) in MDD patients. $n = 39$. **N** Proportions of CD8$^+$ T$_{SCM}$ cells

increased in the susceptible group. CON ($n = 12$), SUS ($n = 12$), RES ($n = 8$). ***$P < 0.0001$ versus the CON group, ###$P < 0.0001$ versus the SUS group. Correlations of CD8$^+$ T$_{SCM}$ cell proportions with sucrose uptake in SPT (**O**) and immobility time in both TST (**P**) and FST (**Q**). $P < 0.0001$ versus the T cell proportions in (**O**–**Q**), $n = 32$. **D, E** Student's $t$ test (two-sided). **N** one-way ANOVA followed by two-sided Holm-Sidak post hoc multiple comparison test. **F**–**M, O**–**Q** two-sided Pearson's or Spearman's correlation coefficient. **D, E, N** Data were presented as mean ± SD. Source data are provided as a Source Data file. pHCs psychiatrically healthy controls, MDD major depressive disorder, T$_{Naïve}$ cells naïve T cells, T$_{CM}$ cells central memory T cells, T$_{EM}$ cells effector memory T cells, T$_{reg}$ regulatory T cells, T$_{SCM}$ cells stem cell-like memory T cells, T$_{EMRA}$ cells terminally differentiated effector memory T cells. HAMD-24 the 24-item Hamilton Depression Rating Scale, MADRS Montgomery Asberg Depression Rating Scale, PHQ-9 the 9-item Patient Health Questionnaire, HAMA Hamilton Anxiety Scale, CON control, SUS susceptible, RES resilient, SPT sucrose preference test, TST tail suspension test, FST forced swim test.

## PPBP-CXCR2 interaction mediated the migration of CD8$^+$ T$_{SCM}$ cells to the intestine

To investigate the potential contribution of CD8$^+$ T$_{SCM}$ cells to brain pathology, additional experiments were conducted using the enhanced green fluorescent protein (EGFP) transgenic mice as donors to isolate CD8$^+$ T$_{SCM}$ cells, which were then adoptively transferred into *Rag1$^{-/-}$* mice as EGFP-labeled cells (Supplementary Fig. 8A). The EGFP-labeled CD8$^+$ T$_{SCM}$ cells isolated from CSDS mice successfully induced depression-like behavior in the *Rag1$^{-/-}$* recipient mice (Supplementary Fig. 8B–D). Unexpectedly, no adoptively transferred EGFP-labeled CD8$^+$ T$_{SCM}$ cells, whether derived from stressed or control mice, were detected in the brain parenchyma of the *Rag1$^{-/-}$* recipient mice through immunostaining and flow cytometry analysis (Fig. 4A, B). To track the distribution of EGFP-labeled CD8$^+$ T$_{SCM}$ cells, we employed vDISCO, a whole-body immunolabeling method, which further confirmed the absence of these cells in the brain parenchyma (Fig. 4C, D and Supplementary Movie 1, 2). Surprisingly, the presence of EGFP-labeled CD8$^+$ T$_{SCM}$ cells in the intestine of the *Rag1$^{-/-}$* recipient mice were detected, with a notably more accumulation of these cells migrating to the intestine in *Rag1$^{-/-}$* mice that received CD8$^+$ T$_{SCM}$ cells isolated from CSDS mice (Fig. 4E and Supplementary Movie 3, 4). Furthermore, the presence of EGFP-labeled CD8$^+$ T$_{SCM}$ cells was also substantiated in heart, liver, lung and lymph node (Supplementary Fig. 9A–D). A growing body of literature indicates that the intricate interplay between the CNS, gut microbiota, and immune function, collectively referred to as the brain-gut-microbiota axis, may critically influence an individual's propensity toward developing stress-related disorders[36]. Accordingly, we investigated the underlying mechanisms that regulated the migration and accumulation of CD8$^+$ T$_{SCM}$ cells within the intestine.

According to the scRNA-seq analysis, CD8$^+$ T$_{SCM}$ cells displayed significantly elevated expression levels of PPBP compared to other T cell subsets (Fig. 4F). Additionally, CD8$^+$ T$_{SCM}$ cells from CSDS susceptible mice showed significantly higher PPBP levels compared with both control and resilient mice (Fig. 4G). Furthermore, an increased number of CD8$^+$ T$_{SCM}$ cells were observed in the colon of mice exposed to CSDS compared with control mice (Fig. 4H). Given that PPBP binds to its receptor CXCR2[37], we utilized the CXCR2 inhibitor (SB265610) to disrupt this interaction. CD8$^+$ T$_{SCM}$ cells isolated from both control and CSDS mice were adoptively transferred into *Rag1$^{-/-}$* mice. In a subgroup of *Rag1$^{-/-}$* mice that received CSDS-derived CD8$^+$ T$_{SCM}$ cells were intraperitoneally injected with SB265610 (2 mg/kg, daily). The transferred cells were identified using CD8 antibody, and the immunofluorescence staining revealed that treatment with SB265610 significantly reduced the accumulation of CD8$^+$ T cells in the colon (Fig. 4I). This finding was further corroborated by flow cytometry analysis, which demonstrated a decreased proportion of CD8$^+$ T$_{SCM}$ cells in the colon (Fig. 4J).

We further employed an in vitro transwell system, which freshly isolated CD8$^+$ T$_{SCM}$ cells from control or CSDS-treated mice were fluorescently labeled by CellTracker Green for visualization and seeded in the upper chamber, and intestinal microvascular endothelial cells (IMECs) were cultured to confluence on permeable supports to assess CD8$^+$ T$_{SCM}$ cell migration (Fig. 4K). We demonstrated that CD8$^+$ T$_{SCM}$ cells from CSDS mice migrated more robustly toward IMECs than the CD8$^+$ T$_{SCM}$ cells from normal mice (Fig. 4L, M). Importantly, PPBP blockade, achieved through either pharmacological inhibition of CXCR2 with SB265610 or lentiviral-mediated PPBP knockdown in CD8$^+$ T$_{SCM}$ cells from CSDS mice, significantly reduced the migration of these cells (Fig. 4L, M and Supplementary Fig. 10). These data confirm that PPBP-CXCR2 is essential for directing CD8$^+$ T$_{SCM}$ cell migration to the colon.

## Blockage of CXCR2 alleviated pathological CD8$^+$ T$_{SCM}$ cell-induced HVA level reduction in brain

As we have noticed higher presence of CD8$^+$ T$_{SCM}$ cells in the colon of the *Rag1$^{-/-}$* mice received pathological CD8$^+$ T$_{SCM}$ cells, this finding prompted us to examine the underlying functions for the infiltrated pathological CD8$^+$ T$_{SCM}$ cells into the colon. We firstly detected the proinflammatory cytokines, particularly those related to CD8$^+$ T$_{SCM}$ cells, such as IFN-γ, TNF-α, and IL-2[38–40], in the plasma of adoptive transfer *Rag1$^{-/-}$* mice. The plasma levels of IFN-γ, TNF-α, and IL-2 were higher in *Rag1$^{-/-}$* mice received pathological CD8$^+$ T$_{SCM}$ cells from CUS mice than those that received CD8$^+$ T$_{SCM}$ cells from control mice (Supplementary Fig. 11A–C). This elevation in cytokine levels was accompanied by an increased overall histology score for mucosal damage and reduced expression of tight junction proteins (Occludin, ZO-1, and Claudin-5) in the colon (Supplementary Fig. 11D, E). Having examined that pathological CD8$^+$ T$_{SCM}$ cells caused the inflammation of intestine, we next investigated the levels of proinflammatory cytokines in colon of *Rag1$^{-/-}$* mice. The levels of IFN-γ, TNF-α, and IL-2 were higher in the colon of *Rag1$^{-/-}$* mice received pathological CD8$^+$ T$_{SCM}$ cells than those received control CD8$^+$ T$_{SCM}$ cells, while these changes were inhibited by SB265610 treatment (Supplementary Fig. 11F–H). To further validate the role of the PPBP-CXCR2 axis in CD8$^+$ T$_{SCM}$ cell induced intestinal inflammation, we also transfected CD8$^+$ T$_{SCM}$ cells with PPBP knockdown lentivirus and found that this intervention significantly reduced the levels of IFN-γ, TNF-α, and IL-2 in the lower chamber of transwell system (Supplementary Fig. 11I–L). These findings suggested that the pathological CD8$^+$ T$_{SCM}$ cells induced the proinflammatory environment in the colon.

Accumulating evidence indicates that gut microbiota-derived metabolites are critical molecular mediators between the microbiota and the host[41,42]. The differential metabolites were identified via metabolomics analysis in microbiota and brain in *Rag1$^{-/-}$* recipient mice, respectively (Supplementary Fig. 12A, B). Upon mapping the

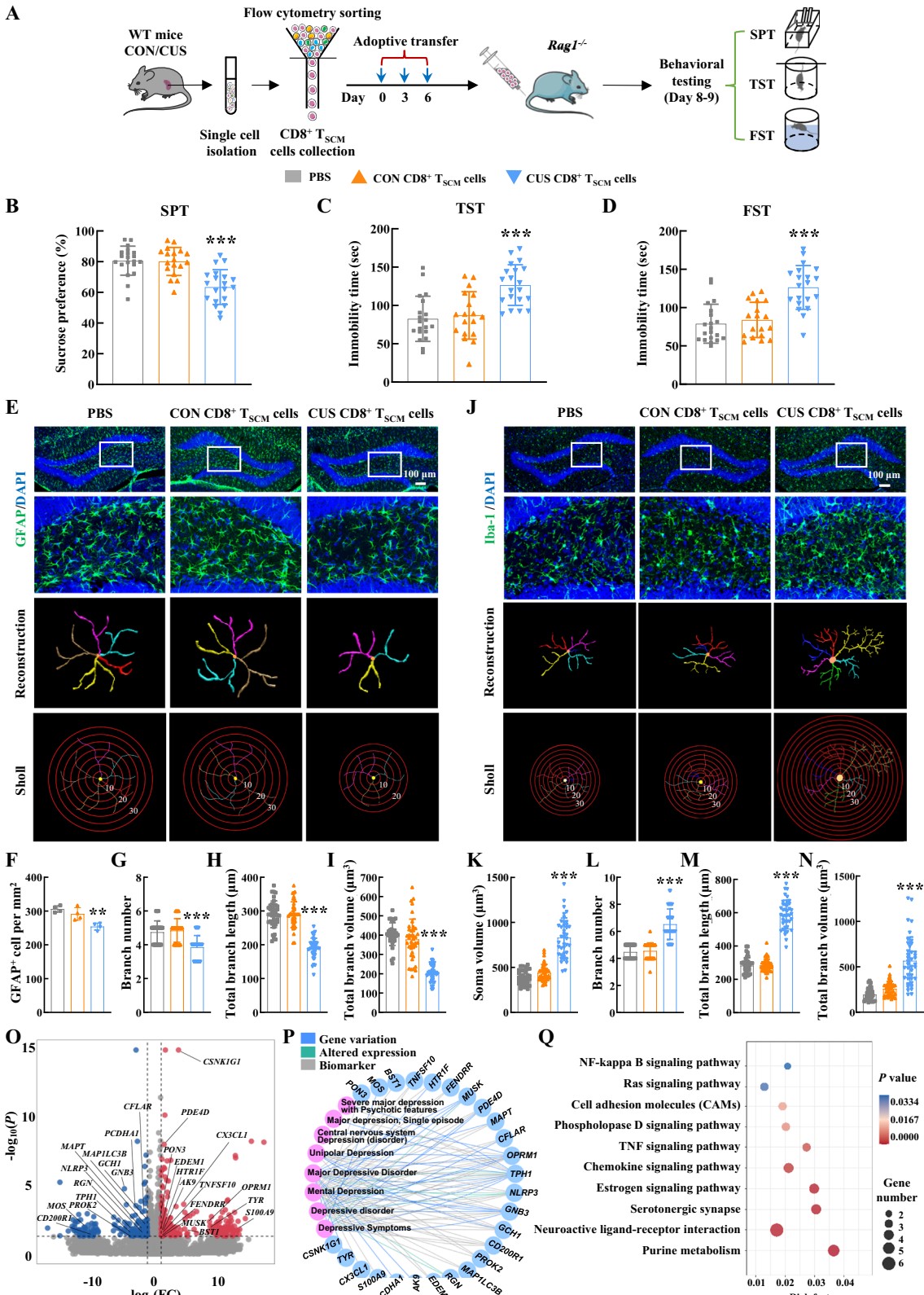

identified differential metabolites to the KEGG pathway database, the pathways engaged in "phenylalanine, tyrosine, and tryptophan biosynthesis", "phenylalanine metabolism", "tyrosine metabolism", and "purine metabolism" were significantly enriched in both microbiota and brain (Fig. 5A, B and Supplementary Fig. 12C). This finding was consistent across both microbial and brain-derived metabolites, suggesting a potential interplay between these metabolites and their

associated pathways in the context of our experimental model. Among these metabolic pathways, the tyrosine metabolism has drawn particular attention due to its reported role as a key pathway in the pathogenesis of depression[43]. Within the brain, we identified 11 tyrosine pathway metabolites in our metabolomics dataset: homovanillic acid (HVA), adenosine, L-tryptophan, L-tyrosine, tyrosine, 3,4-dihydroxymandelate, adenine, succinic acid semialdehyde, glycine,

**Fig. 3 | Pathological CD8$^+$ T$_{SCM}$ cells transmitted depressive symptoms to the host. A** Schematic showing the experimental design for CD8$^+$ T$_{SCM}$ cell adoptive transfer. Components of this figure were created using Servier Medical Art templates, which are licensed under a Creative Commons Attribution 3.0 Unported License; https://smart.servier.com. The depressive-like behaviors of recipient *Rag1*$^{-/-}$ mice were evaluated by the SPT (**B**), TST (**C**) and FST (**D**). ***$P$ < 0.0001 (**B**), ***$P$ = 0.0002 (**C**) and ***$P$ = 0.0006 (**D**) versus the control CD8$^+$ T$_{SCM}$ cells group. PBS ($n$ = 20), CON CD8$^+$ T$_{SCM}$ cells ($n$ = 18), CUS CD8$^+$ T$_{SCM}$ cells ($n$ = 20). Adoptive transfer of CD8$^+$ T$_{SCM}$ cells isolated from stressed mice induced astrocyte dysfunction in the hippocampus of *Rag1*$^{-/-}$ mice. Representative images showing astrocyte immunostaining for GFAP in the *Rag1*$^{-/-}$ mouse hippocampus, followed by three-dimensional reconstruction and Sholl analysis (**E**); Quantification of GFAP-positive cells per square millimeter (**F**); Average branch number (**G**), total branch length (**H**), and total branch volume (**I**) in the hippocampus. **$P$ = 0.0073 (**F**), ***$P$ < 0.0001 (**G**), ***$P$ < 0.0001 (**H**) and ***$P$ < 0.0001 (**I**) versus the control CD8$^+$ T$_{SCM}$ cells group, $n$ = 4/group, 40 cells/group. Adoptive transfer of CD8$^+$ T$_{SCM}$ cells isolated from stressed mice induced activation of microglia in the hippocampus of *Rag1*$^{-/-}$ mice. Representative images showing microglial immunostaining for Iba-1 in the *Rag1*$^{-/-}$ mouse hippocampus, followed by three-dimensional reconstruction and Sholl analysis (**J**); Average soma size (**K**), branch number (**L**), total branch length (**M**), and total branch volume (**N**). ***$P$ < 0.0001 versus the control CD8$^+$ T$_{SCM}$ cells group in (**K–N**), $n$ = 4/group, 40 cells/group. **O** Volcano plot showed the DEGs ([log$_2$(fold change)] ≥1 and $P$ < 0.05) in the hippocampal tissue of *Rag1*$^{-/-}$ recipient mice by R package edgeR. $n$ = 3/group. **P** A total of 26 genes were associated with depression-related terms in DisGeNET. **Q** KEGG annotation of DEGs in the hippocampus of *Rag1*$^{-/-}$ mice after CD8$^+$ T$_{SCM}$ cell adoptive transfer by DAVID tools. (**B–D**, **F–I**, **K–N**) n test. **B–D**, **F–I**, **K–N** Data were presented as mean ± SD. Source data are provided as a Source Data file. CON control, CUS chronic unpredictable stress, CD8$^+$ T$_{SCM}$ cells stem cell-like memory CD8$^+$ T cells, SPT sucrose preference test, TST tail suspension test, FST forced swim test, GFAP glial fibrillary acidic protein, Iba-1 Ionized calcium-binding adapter molecule 1.

p-coumaric acid, and L-glutamine. Of these, resorting to the orthogonal partial least squares-discriminant analysis (OPLS-DA), our reanalysis revealed five metabolites with statistically significant alterations, including HVA, adenosine, L-tryptophan, L-tyrosine, and tyrosine, exhibited statistically significant alterations (VIP > 1, $p$ < 0.05) (Fig. 5C).

Of particular neurobiological relevance, HVA-the primary dopamine metabolite, displayed the most substantial reduction, potentially reflecting compensatory monoaminergic adaptation in depression pathophysiology. This pronounced decrease aligns with evidence of reduced HVA levels in the cerebrospinal fluid and plasma of MDD patients[44–46]. Animal study also showed HVA supplementation restores synaptic function and ameliorates depressive-like behaviors in chronic unpredictable mild stress (CUMS) and corticosterone (CORT) models of depression[47]. Thus we focus on HVA in our study. To clarify the mechanistic link between CXCR2 inhibition and its dual effects on HVA levels and inflammation. In these experiments, *Rag1*$^{-/-}$ mice receiving adoptive transfers of pathological CD8$^+$ T$_{SCM}$ cells were treated with SB265610 or vehicle. Levels of HVA in microbiota, brain homogenates, and plasma were quantified using ELISA. As shown in Fig. 5D–F, in *Rag1*$^{-/-}$ mice receiving pathological CD8$^+$ T$_{SCM}$ cells, SB265610 treatment promoted an increase of HVA levels in the microbiota, although this increase did not reach statistical significance; however, it significantly elevated HVA concentrations in both brain and plasma.

To further explore the relevance of HVA in MDD, we assessed HVA levels in human plasma (Cohort 1) and microbiota samples (Cohort 4) (Supplementary Data 17). We observed consistently lower HVA concentrations in both the plasma and microbiota of MDD patients relative to psychiatrically healthy controls (Fig. 5G, H). To investigated whether HVA had therapeutic potential for depression in mouse models, we administered HVA at doses of 300 mg/kg twice a day for 10 days in CSDS mouse models based upon previous study[47] (Fig. 5I). Behavior phenotype test demonstrated that HVA inhibited the sucrose uptake reduction in the SPT and immobility time prolongation in the TST and FST (Fig. 5J–L). These findings suggested that the adoptively transferred pathological CD8$^+$ T$_{SCM}$ cells induced depressive behaviors through HVA levels reduction.

### Blockage of CXCR2 inhibited the decrease of tyrosine-metabolizing bacteria contributing to the recovery of HVA levels

The gut-brain axis has been proven to play important roles in chronic stress-induced depression, and within this axis, gut microbiota are key players[48]. Thus, we investigated whether pathological CD8$^+$ T$_{SCM}$ cells could alter the abundance of specific bacterial taxa in the intestine, leading to the reduced levels of HVA in *Rag1*$^{-/-}$ recipient mice. Metagenomic sequencing analysis revealed the most identified differential microbiota belonged to four phyla, *Bacillota*, *Bacteroidota*, *Pseudomonadota*, and *Actinomycetota* in *Rag1*$^{-/-}$ recipient mice received pathological CD8$^+$ T$_{SCM}$ cells from CSDS mice (Supplementary Fig. 13A–C). KEGG pathway enrichment analysis revealed that these differential microbiotas were primarily enriched in carbohydrate metabolism and amino acid metabolism pathway (Supplementary Fig. 13D). To ensure consistency between the gut microbiota alterations induced by adoptively transferred pathological CD8$^+$ T$_{SCM}$ cells and those observed in CSDS models, we performed metagenomic sequencing of control and CSDS mice. Among the changes of 137 species have been found consistent with CSDS model, 9 species involved in tyrosine metabolism (Fig. 6A, B). Subsequently, employing spearman correlation analyses, we delineated the relationships between HVA levels, the abundance of microbiota, and depressive behavior of *Rag1*$^{-/-}$ recipient mice. Four microbiotas including *Bifidobacterium scardovii* (*B. scardovii*), *Alistipes indistinctus*, *Mucinivorans hirudinis*, and *Marinilabiliales bacterium* correlated with both sucrose preference in the SPT and immobility times in the TST and FST. Other four microbiota (*Bifidobacterium cuniculi*, *Bacteroidota bacterium*, *Chitinophaga terrae*, and *Alloprevotella sp. oral taxon 473*) have one or two correlations with depressive behavior (Fig. 6C). Importantly, only one microbiota *B. scardovii* positively correlated with HVA levels (Fig. 6C and Supplementary Data 18), suggesting that a deficiency in *B. scardovii* may contribute to the reduced HVA levels in the *Rag1*$^{-/-}$ mice that received pathological CD8$^+$ T$_{SCM}$ cells.

The abundance of *B. scardovii* was decreased in stressed mice and the *Rag1*$^{-/-}$ mice received pathological CD8$^+$ T$_{SCM}$ cells, while these changes were reversed by SB265610 treatment (Fig. 6D, E). To investigate whether *B. scardovii* produce HVA, we cultured them in vitro with DSMZ medium 104. The results showed that *B. scardovii* produced HVA (Fig. 6F). We further investigated whether *B. scardovii* had therapeutic potential for depression in mouse models. We then treated CSDS mouse model with *B. scardovii* ($1 \times 10^9$ colony-forming units (CFU)/day) once a day for 10 days (Fig. 6G). Behavior test demonstrated significant increase in sucrose preference and reduction in immobility time after intervention (Fig. 6H, J). We also observed that treatment with *B. scardovii* in CSDS mice resulted in increased concentrations of HVA in both the plasma and brain (Fig. 6K, L). These findings indicate the potential of *B. scardovii* in alleviating depressive phenotypes in mice.

### Blockage of CXCR2 alleviated neuroinflammation and improved depressive symptoms

According to previous studies, lower levels of HVA correlated with a higher burden of neuroinflammation and depression in people[49,50]. To further investigate the underlying mechanisms by which pathological CD8$^+$ T$_{SCM}$ cells induce depressive symptoms, we evaluated whether these cells contribute to neuroinflammation. As shown in Supplementary Fig. 14A–C, the level of CD8$^+$ T$_{SCM}$ cell-secreted cytokines (IFN-γ, TNF-α, and IL-2) was elevated in the hippocampus of *Rag1*$^{-/-}$

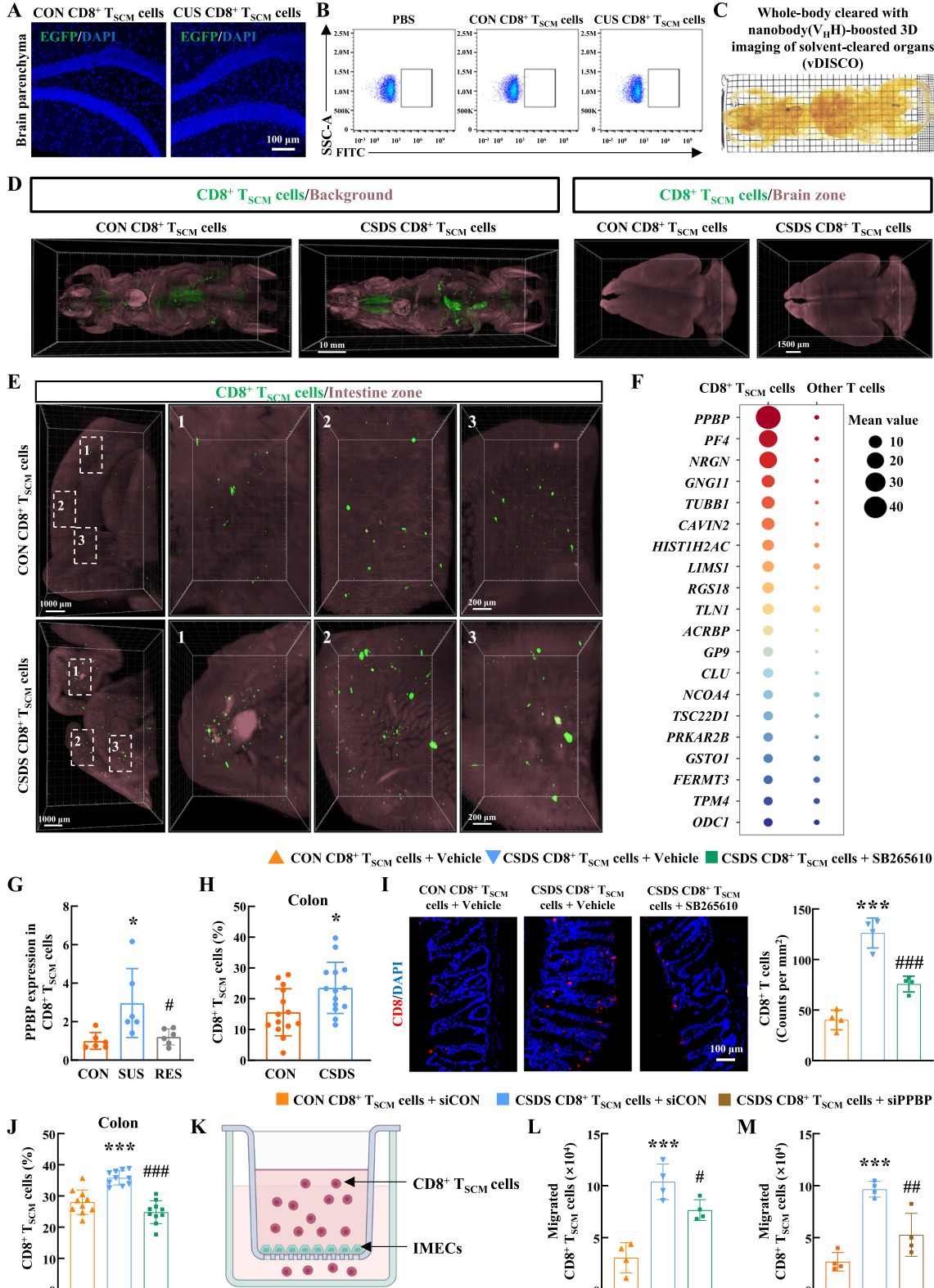

mice that received pathological CD8+ T$_{SCM}$ cells compared to those receiving control CD8+ T$_{SCM}$ cells.

Given that the CXCR2 inhibitor SB265610 reversed the reduction in HVA levels induced by CD8+ T$_{SCM}$ cells, we further investigated whether this inhibitor could alleviate depressive symptoms by inhibiting neuroinflammation. In *Rag1*$^{-/-}$ mice that received CD8+ T$_{SCM}$ cells isolated from stressed mice, SB265610 treatment restored

sucrose uptake in the SPT and reduced immobility time in both the TST and FST (Fig. 7A–C). Concurrently, this intervention reduced pro-inflammatory cytokines IFN-γ, TNF-α, and IL-2 in plasma and brain of the same mice (Fig. 7D-I). Immunostaining revealed that SB265610 treatment alleviated astrocyte dysfunction induced by pathological CD8+ T$_{SCM}$ cells, evidenced by an increased number of GFAP-positive cells (Fig. 7J, K) and significantly increased astrocyte branch numbers,

**Fig. 4 | PPBP-CXCR2 interaction mediated the migration of CD8+ T$_{SCM}$ cells to the intestine. A** Adoptive transfer EGFP labbed-CD8+ T$_{SCM}$ cells in the brain parenchyma of *Rag1*$^{-/-}$ recipient mice detected by immunostaining. *n* = 5/group. **B** EGFP labbed-CD8+ T$_{SCM}$ cells in the brain parenchyma of *Rag1*$^{-/-}$ recipient mice detected by flow cytometry. **C** The *Rag1*$^{-/-}$ recipient mice were cleared with SHANEL and imaged by light sheet microscopy. **D** The distribution of adoptively transferred CD8+ T$_{SCM}$ cells in *Rag1*$^{-/-}$ recipient mice. *n* = 2/group. **E** Image of the brain parenchyma and intestine in *Rag1*$^{-/-}$ recipient mice. **F** The top 20 genes highly expressed in the CD8+ T$_{SCM}$ cells through scRNA-seq analysis. **G** The PPBP levels of CD8+ T$_{SCM}$ cells from control, susceptible and resilient mice. *$^*P$* = 0.021 versus the CON group, *$^#P$* = 0.0277 versus the SUS group, *n* = 6/group. **H** The proportion of CD8+ T$_{SCM}$ cells in the colon samples from control and CSDS mice detected by flow cytometry. *$^*P$* = 0.0141 versus the CON group, *n* = 14/group. **I** The immunostaining images and quantification of CD8+ T cells in colon of *Rag1*$^{-/-}$ recipient mice. *$^{***}P$* < 0.0001 versus the control CD8+ T$_{SCM}$ cells group, *$^{###}P$* = 0.0003 versus the

CSDS CD8+ T$_{SCM}$ cells group, *n* = 4/group. **J** The proportion of CD8+ T$_{SCM}$ cells within the colon of *Rag1*$^{-/-}$ recipient mice detected by flow cytometry. *$^{***}P$* < 0.0001 versus the control CD8+ T$_{SCM}$ cells group, *$^{###}P$* < 0.0001 versus the CSDS CD8+ T$_{SCM}$ cells group, *n* = 10/group. **K** Schematic depicting the experimental setup of transwell. Created in BioRender. Yao, H. (2026) https://BioRender.com/iqfc57z. The migration of CD8+ T$_{SCM}$ cells treated with SB265610 (**L**) or siPPBP lentivirus (**M**). *$^{***}P$* = 0.0001 (**L**) and *$^{***}P$* = 0.0002 (**M**) versus the control CD8+ T$_{SCM}$ cells group, *$^#P$* = 0.0241 (**L**) and *$^{##}P$* = 0.0029 (**M**) versus the CSDS CD8+ T$_{SCM}$ cells group, *n* = 4/group. **G, I–J, L–M** one-way ANOVA followed by two-sided Holm-Sidak post hoc multiple comparison test. **H** Student's *t* test (two-sided). **G–J, L–M** Data were presented as mean ± SD. Source data are provided as a Source Data file. CON control, CUS chronic unpredictable stress, CD8+ T$_{SCM}$ cells stem cell-like memory CD8+ T cells, CSDS chronic social defeat stress, IMECs intestinal microvascular endothelial cells.

volume, and length (Fig. 7L–N). Furthermore, SB265610 treatment inhibited the microglial activation induced by pathological CD8+ T$_{SCM}$ cells, as indicated by reductions in microglial soma size, branch number, branch length, and branch volume in the hippocampus of *Rag1*$^{-/-}$ recipient mice (Fig. 7O–S). Moreover, CSDS treatment significantly decreased the spine number of neurons in the hippocampus, as shown by Golgi staining, and these phenotypes were ameliorated by SB265610 treatment (Fig. 7T, U). Additionally, SB265610 treatment significantly mitigated the decreased expression of presynaptic membrane protein synapsin-1 (SYN1) and brain-derived neurotrophic factor (BDNF) in the hippocampus of CSDS mice (Supplementary Fig. 15A). Overall, these findings suggested that CXCR2 inhibitor SB265610 alleviated neuroinflammation and enhanced neuronal plasticity, thereby improving depressive symptoms.

### HVA alleviates depression through inhibition of neuroinflammation and enhancement of neuronal plasticity

As behavior phenotype test demonstrated that HVA had a therapeutic effect on depression, we hypothesized that gut-derived HVA could cross the BBB and directly exert antidepressant effects within the brain. To further investigate whether HVA alleviates depressive symptoms by inhibiting neuroinflammation, we conducted immunostaining analyses. Our results showed that HVA treatment alleviated astrocyte dysfunction induced by CSDS (Fig. 8A), as evidenced by an increased number of GFAP-positive cells (Fig. 8B) and significantly enhanced astrocyte branch numbers, volume, and length (Fig. 8C–E). Additionally, HVA treatment inhibited CSDS-induced microglial activation, as indicated by reductions in the microglial soma size, branch number, branch length, and branch volume in the hippocampus (Fig. 8F–J). Furthermore, we observed that HVA treatment significantly upregulated the spine number of neurons and the expression of SYN1 and BDNF in the hippocampus of CSDS mice (Fig. 8K, L, Supplementary Fig. 15B). These results indicate that HVA not only alleviates neuroinflammation but also promotes neuronal plasticity, leading to improvements in depressive symptoms.

### Discussion

The intricate interplay between host peripheral immunity and gut microbiota in the pathogenesis of depression represents one of the most captivating yet contentious areas of inquiry in psychiatric research. Here, we unveil a mechanism through which T cell immunity escalates the cascade of neuroinflammatory events in depression. Our findings reveal a significant enrichment of the CD8+ T$_{SCM}$ cell subset in patients with MDD, with the proportion of these cells positively correlating with the severity of depressive symptoms. Utilizing a pressure-driven, nanobody-based whole-body immunolabeling technology named vDISCO, we visualized the migratory pattern of CD8+ T$_{SCM}$ cells, pinpointing the intestine as a primary destination facilitated through the PPBP-CXCR2 axis, as no such migration was observed in the brain.

Remarkably, the pathological CD8+ T$_{SCM}$ cells instigate a decrease of gut microbiota integral to tyrosine metabolism, concurrently leading to a decline in HVA. This metabolic perturbation triggers neuroinflammation and results in depressive behaviors. The administration of CXCR2 inhibitor effectively disrupted the molecular interaction between PPBP and CXCR2, leading to a significant diminution in the localization of CD8+ T$_{SCM}$ cells in the intestine (Supplementary Fig. 16). This finding underscores the central role of CD8+ T$_{SCM}$ cells in the pathophysiological mechanisms underlying depression, highlighting the potential of CXCR2 inhibitor as a therapeutic target for modulating T cell-mediated neuroinflammation.

T$_{SCM}$ cells represent a recently characterized subset of T cells, distinguished by their expression of a suite of naïve cell markers, including CCR7, lack of CD45RO, presence of CD45RA and CD62L, high levels of the memory cell-associated marker CD95/Fas, and high levels of the stem cell-linked surface antigen CD127 in human contexts[51,52]. Functionally, T$_{SCM}$ cells act as multipotent progenitors endowed with superior self-renewal capabilities, giving rise to diverse memory T cell populations and playing a critical role in orchestrating immune responses[51]. Emerging data underscore the potency of CD8+ T$_{SCM}$ cells in catalyzing robust and enduring tumor regression[51,53]. These cells have been identified as a reservoir for pathogenic effector T cells during autoimmune relapses post-immunotherapy and during islet transplantation in type 1 diabetes mellitus[54,55]. Moreover, these cells are intricately involved in the induction and progression of atherosclerosis[56]. Our investigation revealed that in MDD patients, CD8+ T$_{SCM}$ cells predominantly facilitated the activation, expansion, and differentiation of T cells, thereby amplifying systemic immune inflammation characteristic of MDD. Our study provides the foundational elucidation of the regulatory functions exerted by CD8+ T$_{SCM}$ cells in depression, demonstrating their capacity to exacerbate neuroinflammation and consequently intensify depressive phenotypes. This insight paves the way for a deeper understanding of the immunological underpinnings of depression and suggests potential therapeutic avenues targeting these cells.

Adoptive transfer of CD8+ T$_{SCM}$ cells solidified their pathogenic role in depression, as evidenced by the induction of neuroinflammation and subsequent depressive-like behaviors in *Rag1*$^{-/-}$ recipient mice upon adoptive transfer of CD8+ T$_{SCM}$ cells isolated from stressed mice. Intriguingly, despite the evident neurological impact, no adoptively transferred CD8+ T$_{SCM}$ cells were detected in the brain parenchyma of *Rag1*$^{-/-}$ recipient mice, challenging prevailing hypotheses regarding direct CNS infiltration[57–59]. Therefore, we presented an exhaustive and detailed in vivo atlas of adoptively transferred T cells within the murine model, utilizing vDISCO[21]. This meticulous mapping affords an unparalleled perspective on the spatial dynamics and migratory patterns of these T cells, elucidating their interactions with diverse host tissue and organs. Our results unveiled a distinctive migratory pattern of CD8+ T$_{SCM}$ cells, notably directed toward the intestine. This

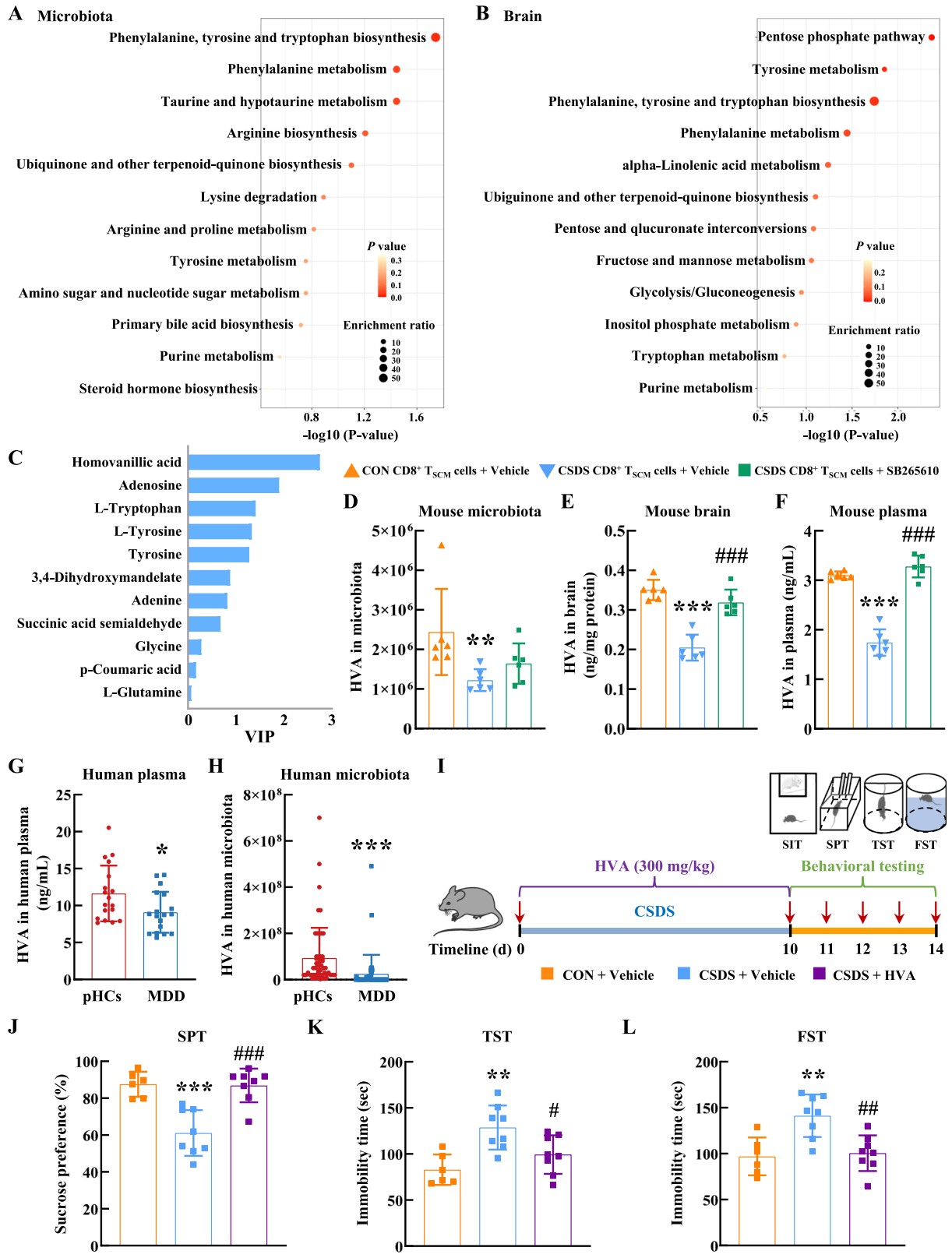

migration was driven by the interaction between PPBP highly expressed on CD8+ $T_{SCM}$ cells and its receptor CXCR2. Administration of the selective CXCR2 inhibitor effectively disrupted this molecular engagement, resulting in a marked reduction of CD8+ $T_{SCM}$ cell localization within the intestinal tissue and thereby highlighting the pivotal role of the PPBP-CXCR2 axis. Beyond the intestine, the presence of CD8+ $T_{SCM}$ cells was also observed in other organs including heart, liver

and lung. These inquiries will constitute a direction of our subsequent research endeavors, aiming to comprehensively delineate the multifaceted roles of CD8+ $T_{SCM}$ cells in depression pathophysiology.

Metabolites serve as pivotal mediators along the gut-brain axis, playing a cardinal role in the pathophysiology of depression[20,60,61]. Through integrative metabolomic profiling of both the gut microbiota and brain, our investigation has unveiled HVA as a mediator between

**Fig. 5 | Blockage of CXCR2 alleviated pathological CD8$^+$ T$_{SCM}$ cell-induced HVA level reduction in brain.** The enriched KEGG pathway of identified differential metabolites ($P < 0.05$ and VIP > 1) in microbiota (**A**) and brain tissue (**B**) of $Rag1^{-/-}$ recipient mice by MetaboAnalyst. **C** The VIP value of metabolites that actively engage in the metabolic pathways. **D** The level of HVA in microbiota of $Rag1^{-/-}$ recipient mice detected by metabolomics analysis. **$P = 0.0035$ versus the control CD8$^+$ T$_{SCM}$ cells group, $n = 6$/group. The level of HVA in brain tissue (**E**) and blood (**F**) of $Rag1^{-/-}$ recipient mice that received CD8$^+$ T$_{SCM}$ cells detected by ELISA. ***$P < 0.0001$ versus the control CD8$^+$ T$_{SCM}$ cells group, ###$P < 0.0001$ versus the CSDS CD8$^+$ T$_{SCM}$ cells group, $n = 6$/group. **G** The HVA level in human plasma samples detected by HPLC. *$P = 0.0371$ versus the pHCs group, $n = 18$/group. **H** The HVA level in human microbiota samples detected by metagenomic analysis. ***$P < 0.0001$ versus the pHCs group, $n = 55$/group. **I** Animal experiment in CSDS model (CON+Vehicle, CSDS+Vehicle, CSDS + HVA). Components of this figure were created using Servier Medical Art templates, which are licensed under a Creative Commons Attribution 3.0 Unported License; https://smart.servier.com. The depressive-like behaviors of recipient $Rag1^{-/-}$ mice were evaluated by the SPT (**J**), TST (**K**), and FST (**L**). CON+Vehicle ($n = 6$), CSDS+Vehicle ($n = 8$), CSDS + HVA ($n = 8$). ***$P = 0.0002$ (**J**), **$P = 0.0022$ (**K**), and **$P = 0.0031$ (**L**) versus the CON +Vehicle group. ### $P = 0.0002$ (**J**), #$P = 0.0234$ (**K**), and ##$P = 0.0031$ (**L**) versus the CSDS+Vehicle group. **D** Kruskal-Wallis test followed by two-sided Dunn's post hoc multiple comparison test. **E–F, J–L** One-way ANOVA followed by two-sided Holm-Sidak post hoc multiple comparison test. **G–H** Mann–Whitney $U$ test (two-sided). **D–H, J–L** Data were presented as mean ± SD. Source data are provided as a Source Data file. CON control, CSDS chronic social defeat stress, CD8$^+$ T$_{SCM}$ cells stem cell-like memory CD8$^+$ T cells, HVA homovanillic acid, pHCs psychiatrically healthy controls, MDD major depressive disorder, SPT sucrose preference test, TST tail suspension test, FST forced swim test.

CD8$^+$ T$_{SCM}$ cells and the pathogenesis of depression. HVA is a metabolite of dopamine degradation, serves as a biomarker for dopaminergic activity, and has been implicated in the neuropathology of depression[47,62]. Clinical evidence consistently indicates lower HVA concentrations in the cerebrospinal fluid of patients with MDD compared to healthy controls[49,50,63]. Lower HVA significantly correlated with higher levels of neuroinflammation and depression in people with HIV disease[49]. In the context of depression, HVA exhibited neurotrophic and neuroprotective effects by preventing SYN1 degradation and inhibiting autophagic death in synapses[47]. In our investigation, we observed that CD8$^+$ T$_{SCM}$ cells induce a reduction in HVA levels by migrating to the intestine and depleting the microbiota essential for HVA production. These findings not only underscore the pivotal role of CD8$^+$ T$_{SCM}$ cells in depression pathophysiology but also pave the way for personalized treatment approaches aimed at restoring metabolic balance in the gut-brain axis.

Extensive investigations have highlighted significant perturbations in the composition of the gut microbiota, leading to alterations in metabolite profiles among individuals afflicted with depression[64]. Our finding indicated that the deficiency of eight tyrosine-metabolizing bacteria may contribute to the reduced levels of HVA and the induction of depressive behavior by adoptively transferred CD8$^+$ T$_{SCM}$ cells isolated from stressed mice. Of particular importance is the role of *B. scardovii*. Treatment with *B. scardovii* elevated HVA concentrations in both plasma and brain and ameliorated depression-like behavior in CSDS mice, underscoring its critical role in mitigating depressive phenotypes in depression. In previous studies, *Bacteroides acidifaciens* has been found to be associated with depression[65]. Additionally, it has been reported that *Bifidobacterium longum* produces HVA using tyrosine as a substrate, and it further participates in the pathogenesis of depression by regulating HVA production[47]. These findings suggest a complex interplay between the gut microbiota and the immune system in the context of depression and highlight the potential role of specific bacterial taxa in modulating HVA biosynthesis and depressive behaviors.

In summary, our comprehensive analysis elucidates the profound association between CD8$^+$ T$_{SCM}$ cells and MDD. Notably, we pioneered the mapping of adoptively transferred CD8$^+$ T$_{SCM}$ cell migration throughout the organism, with an emphasis on their migratory patterns toward the intestine, orchestrated by the PPBP-CXCR2 axis. An additional revelation of our study was the impact of CD8$^+$ T$_{SCM}$ cells on the equilibrium of gut microbiota metabolism. Our data revealed that the administration of CXCR2 inhibitor ameliorated the decrease of tyrosine-metabolizing bacteria associated with HVA biosynthesis, thereby recovering the HVA levels. The restoration of HVA attenuated neuroinflammatory cascades with concomitant alleviation of depressive symptoms. Collectively, these findings underscore the pivotal role of CD8$^+$ T$_{SCM}$ cells in depression pathophysiology and highlight the potential of CXCR2 inhibitor as a therapeutic strategy to modulate gut microbiota metabolism and neuroinflammation in MDD.

## Methods
### Participants
Of 583 potential participants screened by phone, 296 were assessed for eligibility on site, and 230 were included and separated into four cohorts (cohorts 1, 2, 3 and 4). The 230 subjects included 115 patients with major depressive disorder (MDD) and 115 psychiatrically healthy controls (pHCs). All subjects were of Chinese Han ethnicity. MDD patients were recruited from ZhongDa Hospital, Southeast University, the Third People's Hospital of Huzhou, and the Third People's Hospital of Huai'an, and pHCs were recruited through socially-oriented advertising in Nanjing and Huai'an from January 2021 to December 2022. Diagnoses of neuropsychiatric pathology were conducted by experienced psychiatrists using the Structured Clinical Interview for the Diagnostic and Statistical Manual of Mental Disorders-5 (SCID-5). The inclusion criteria for MDD were as follows: (1) 18–60 years old, (2) met the diagnostic criteria of DSM-5 with MDD, (3) either first-/recent-onset drug-naive (44 patients) or drug-free for at least 2 weeks (16 patients) before entering the trial, and (4) 24-item Hamilton Depression Scale (HAMD-24) score ≥20. Subjects without any history of DSM-Axis I disorder and with HAMD-24 < 8 were included as pHCs. The exclusion criteria for both MDD and pHCs included: (1) pregnancy or puerperium, (2) a history of manic or hypomanic episodes, (3) comorbidity of other mental disorders, including alcohol, tobacco, or drug abuse and dependence, (4) physical diseases (i.e., neurological diseases including epilepsy, mental retardation, and multiple sclerosis; immune/autoimmune diseases; infectious diseases; metabolic diseases including diabetes and obesity; cardiovascular diseases; liver and renal insufficiency; endocrine diseases; respiratory diseases; cancer; and serious injury), and (5) a history of somatic drug therapy (including glucocorticoids and anti-inflammatory/immunomodulatory drugs during the past three months) with central nervous system side effects or immune system side effects.

There was a heavy propensity toward female participants. It is well known that females are at higher risk for developing depressive disorders compared to males[3]. Empirical investigations dating back many years have demonstrated that females are typically twice as likely as males to develop depressive disorders[66–68]. Similar ratios have been reported in a wide range of patient and non-patient populations, including clinical and community samples with major depression and community samples with sub-threshold depressive symptoms[66–69]. Furthermore, the results of the latest China Mental Health Survey showed that the prevalence of any type of depressive disorder in females is higher than that in males, and the lifetime prevalence among females is 1.20- to 1.72-fold greater than that among males[70]. In addition, the chapter on MDD in the DSM-5 mentions that the most reproducible findings in the epidemiology of MDD have a greater prevalence in females, and female experience 1.5- to 3-fold higher rates of morbidity than male beginning in early adolescence[71]. Therefore, a heavy propensity toward females is consistent with the characteristics of epidemiology.

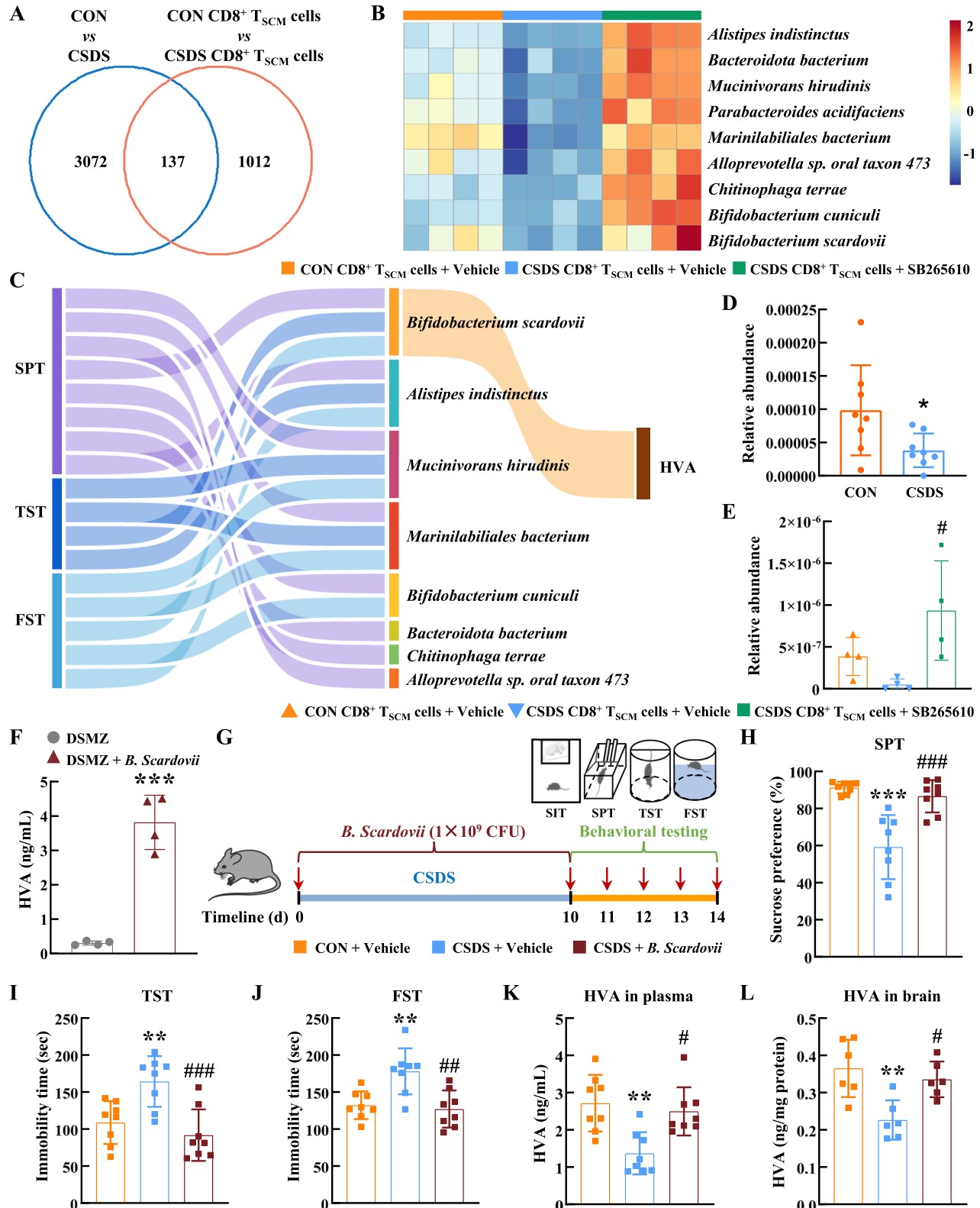

## Clinical assessment

Except for collecting all subjects' sociodemographic data including age, sex, marital status, years of education, body mass index (BMI), family history of mental disorder and so on, we also collected information about their childhood and recent experiences, as well as anhedonia. Childhood Trauma Questionnaire (CTQ) and Life Event Scale (LES) were applied to collect subjects' experiences. Snaith-Hamilton Pleasure Scale (SHAPS) and Temporal Experience of Pleasure Scale (TEPS) were employed to assess all subjects' severity of anhedonia. The 9-item Patient Health Questionnaire (PHQ-9) and the 7-item Generalized Anxiety Disorder (GAD-7) were used to evaluate subjects' depressive and anxious symptoms themselves. In addition, the severity of depressive and anxious symptoms in MDD patients and pHCs was also evaluated by trained and experienced senior psychiatrists using HAMD-24, Montgomery Asberg Depression Rating Scale (MADRS), and Hamilton Anxiety Scale (HAMA), respectively.

**Fig. 6 | Blockage of CXCR2 inhibited the decrease of tyrosine-metabolizing bacteria contributing to the recovery of HVA levels. A** The Venn Diagram of differentially bacteria both in the CSDS mice and the *Rag1⁻/⁻* mice that received CD8⁺ T_SCM cells. **B** Heatmap of identified differential microbiota involved in tyrosine metabolism pathway. **C** Sankey diagram showing correlations between bacteria, HVA, and depressive behavior. **D** The abundance of *B. scardovii* in the control and CSDS mice. *$P = 0.0335$ versus the CON group, $n = 8$/group. **E** The abundance of *B. scardovii* in the recipient *Rag1⁻/⁻* mice. #$P = 0.0237$ versus the CSDS CD8⁺ T_SCM cells group, $n = 4$/group. **F** HVA production by *B. scardovii* with DSMZ medium in vitro. ***$P = 0.0001$ versus the DSMZ group, $n = 4$/group. **G** Animal experiment in CSDS model (CON+Vehicle, CSDS+Vehicle, CSDS + *B. scardovii*). Components of this figure were created using Servier Medical Art templates, which are licensed under a Creative Commons Attribution 3.0 Unported License; https://smart.servier. com. The depressive-like behaviors of recipient *Rag1⁻/⁻* mice were evaluated by the

SPT (**H**), TST (**I**), and FST (**J**). ***$P < 0.0001$ (**H**), **$P = 0.0056$ (**I**) and **$P = 0.0034$ (**J**) versus the CON+Vehicle group, ###$P = 0.0002$ (**H**), ###$P = 0.0007$ (**I**) and ##$P = 0.002$ (**J**) versus the CSDS+Vehicle group, $n = 8$/group. The level of HVA in plasma (**K**, $n = 8$/group) and brain tissue (**L**, $n = 6$/group) detected by ELISA. **$P = 0.0079$ (**K**) and **$P = 0.0038$ (**L**) versus the CON+Vehicle group, #$P = 0.0266$ (**K**) and #$P = 0.0141$ (**L**) versus the CSDS+Vehicle group. **C** two-sided Pearson's or Spearman's correlation coefficient. **D, F** Student's *t* test (two-sided). **E, H–J, L** one-way ANOVA followed by two-sided Holm-Sidak post hoc multiple comparison test. **K** Kruskal-Wallis test followed by two-sided Dunn's post hoc multiple comparison test. **D–F, H–L** Data were presented as mean ± SD. Source data are provided as a Source Data file. CON control, CSDS chronic social defeat stress, CD8⁺ T_SCM cells stem cell-like memory CD8⁺ T cells, SPT sucrose preference test, TST tail suspension test, FST forced swim test, HVA homovanillic acid, DSMZ DSMZ medium 104, *B. Scardovii Bifidobacterium scardovii*.

## Animals

Adult male C57BL/6J mice, EGFP and *Rag1⁻/⁻* mice on C57BL/6J background (25.0–30.0 g, 6–8 weeks old) were purchased from Gem-Pharmatech (Nanjing, China) and randomly assigned to experimental groups. Mice were maintained in a specific pathogen-free facility with a constant temperature and humidity and a 12-h light/12-h dark cycle with the lights on at 7:00 a.m. Food and water were available ad libitum. All animal procedures were performed in strict accordance with the Animal Research: Reporting of In Vivo Experiments Guidelines. The care and use of animals were reviewed and approved by the Institutional Animal Care and Use Committee at the Medical School of Southeast University (20190222004).

## Single-cell suspension preparation

Peripheral blood mononuclear cells (PBMCs) were obtained from whole blood samples by density gradient centrifugation. After the plasma was collected by centrifugation, the Ficoll-Paque Plus (17144002, Cytiva, Sweden) was placed at the bottom of the centrifuge tube, the whole blood was diluted to 1× PBS in a 1:1 ratio and then tiled on the top of Ficoll-Paque Plus, centrifuged at $500 \times g$ for 30 min. The centrifugal PBMCs layer was then collected, and incubated with ACK lysis buffer (C3702, Beyotime Biotech, China) for 5 min at 4 °C to remove red blood cells. PBMCs were collected by centrifugation and then re-suspended with 1 mL 1× PBS. Finally, cell suspensions were filtered through a 70 μm cell strainer. Cell viability >90% was required for subsequent construction of libraries.

## Single-cell library generation and sequencing

Single-cell libraries were prepared per the Chromium Single Cell 3′ library preparation kit user guide (10× Genomics). All libraries prepared for this study were sequenced on a NovaSeq 6000 system (Illumina) with 150 bp paired-end sequencing, and each library was sequenced for a whole lane. Briefly, cellular suspension, barcoded Single Cell 3′ Gel Beads, and partitioning oil were loaded onto a Chromium Chip A to generate single-cell gel bead-in-emulsions (GEMs). GEM-reverse transcriptions (GEM-RTs) were performed in a Veriti 96-well thermal cycler (Thermo Fisher Scientific, USA). Silane magnetic beads were used to remove leftover biochemical reagents and primers from the post-GEM reaction mixture. Full-length, barcoded cDNA was then amplified by PCR to generate sufficient mass for library construction. Indexed sequencing libraries were constructed using the Chromium Single Cell 3′ Library Construction Kit according to the user guide. Barcoded sequencing libraries were quantified by quantitative PCR on ABI StepOnePlus Real-Time PCR System (Life Technologies, USA).

## Pre-processing scRNA-seq data

The sequences from scRNA-seq were aligned to human transcriptome GRCh38, generated matrix of unique molecular identifier (UMI) counts per gene and associated cell barcode by using Cell Ranger (v3.1.0, 10× Genomics). Raw scRNA-seq data were deposited in the Genome

Sequence Archive database (HRA002269). We further analyzed these metrics using the R (v3.6.3) package Seurat (v3.1.1). Only genes expressed in at least three cells and cells with a minimum of 400 genes were kept. Low-quality cells meeting one of the following thresholds were further excluded: 1) the number of expressed genes was lower than 500 or larger than 4000; 2) the UMI counts were lower than 2000 or larger than 15,000 per cell; 3) more than 8% of UMI counts were mapped to mitochondrial or ribosomal genes. After filtering, we detected 36,785 genes in a total of 100,522 cells.

## Dimension reduction and identification of cell clusters

The qualified data was normalized via a global-scaling normalization method, "LogNormalize", and cell clusters were identified through dimensionality reduction clustering. Canonical correspondence analysis (CCA) was conducted to correct the batch effect, and then the data were integrated. Z-score normalization was performed on the integrated data, and the normalized expression was used for Principal Component Analysis (PCA). Through PCA dimensionality reduction, the clustering algorithm based on the graph theory was used to cluster the cells (resolution = 0.5). On the basis of the classification results of cell subsets, the single-cell subpopulation classification was visualized using the t-Distributed Stochastic Neighbor Embedding (t-SNE) non-linear clustering method. A total of 27 clusters were obtained using cell marker genes for cell type identification. We removed a cluster with no notable cell-type-specific markers.

## Specific marker gene analysis

To dissect the specific gene markers from each subpopulation, we screened up-regulated genes from each cluster. Seurat's Wilcoxon rank sum test was used to analyze gene differential expression in different cell populations. The screening condition required that the percentage of gene-expressed cells of all cells in specific cluster >25%. $P \leq 0.01$ was considered significant. Log fold-change of the average expression (logFC) $\geq 0.360674$; that is, gene up-regulation multiple ≥1.28. Cell clusters and their marker genes were described in detail in the figures and text. Heat maps or violin plots were then used to show the distribution of marker genes. Identities of clusters of cells were manually annotated using known marker genes in published articles with the help of the R package *SingleR* (v1.0.1)[72]. Sub-clustering of major immune cell clusters was performed in the same workflow.

## Differential expression and analysis of signaling pathways

For specific cluster comparisons between MDD patients and pHCs, we used the Wilcoxon rank sum test in Seurat (v3.1.1) to detect the fold change of different expression. The package MAST[73] was used to assess the significance of differential expression, then we utilized the Benjamini-Hochberg method to correct for multiple testing on the $P$-values and obtained adjusted $P$-values. We identified differentially expressed genes (DEGs) as following criteria: (1) $P < 0.05$; (2) $|\log_2 FC| \geq 1$, logFC represented log fold-change of the average expression between

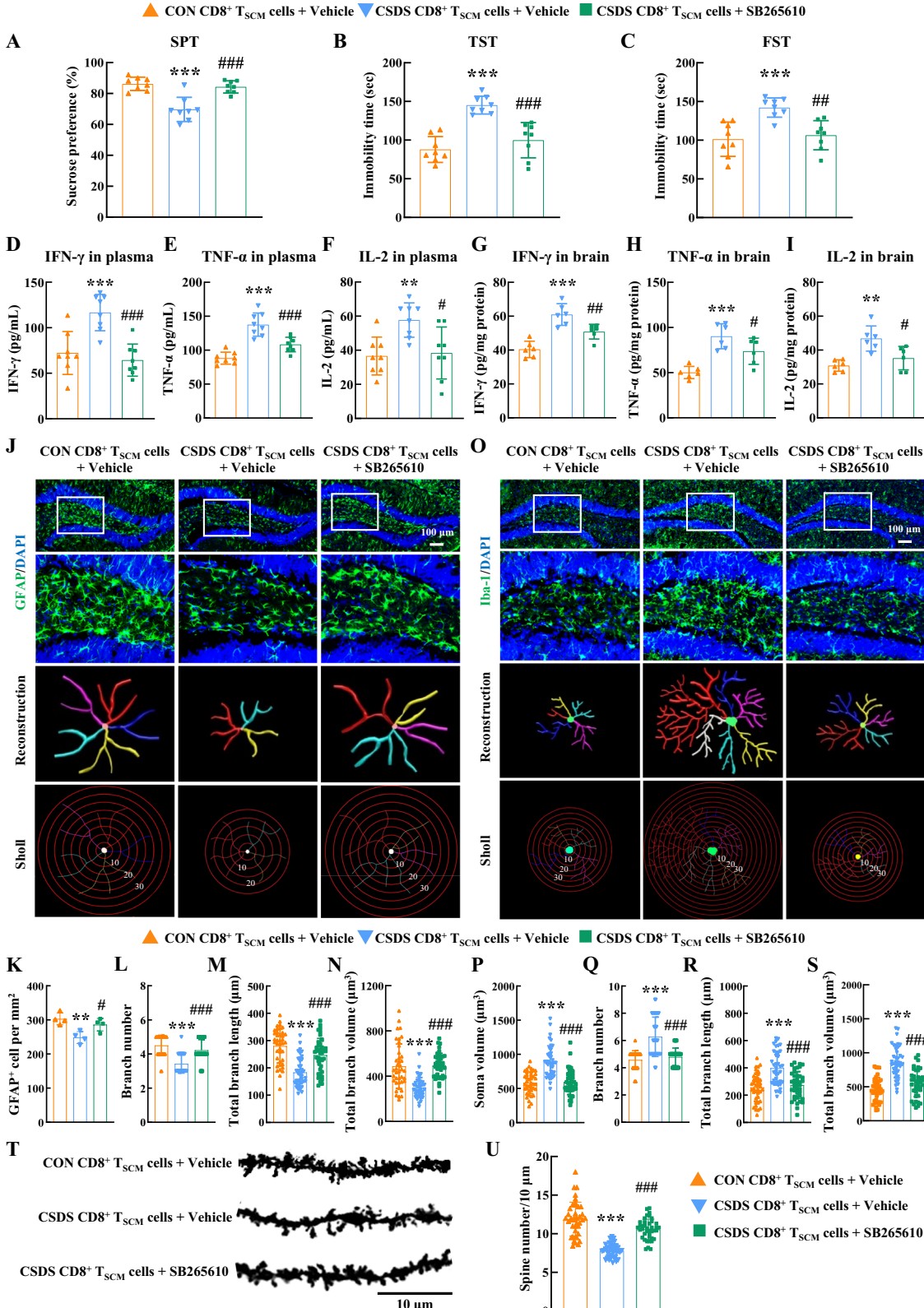

the two groups; (3) The percentage of cells where the gene was detected in specific cluster >10%. Pathway annotation and enrichment for the list of DEGs were calculated using Metascape (www.metascape.org)[74].

### Ligand-receptor expression and cell interactions

To enable a systematic analysis of intercellular interactions, cell-cell communication was predicted on the receptor−ligand pairs using CellPhoneDB (https://www.CellPhoneBD.org/)[75] and CellChat (http://www.cellchat.org/)[76]. Only receptors and ligands expressed in more than the user-specified threshold percentage of the cells in the specific cluster were considered for the analysis (default is 10%). Pairwise comparisons between all T cell types were performed, and only those with $P < 0.05$ were used for subsequent prediction of cell-cell communication.

**Fig. 7 | Blockage of CXCR2 alleviated neuroinflammation and improved depressive symptoms.** The depressive-like behaviors of recipient *Rag1*$^{-/-}$ mice were evaluated by the SPT (**A**), TST (**B**), and FST (**C**). ***$P$ < 0.0001 (**A**), ***$P$ < 0.0001 (**B**) and ***$P$ = 0.0006 (**C**) versus the control CD8$^+$ T$_{SCM}$ cells+Vehicle group, ###$P$ < 0.0001 (**A**), ###$P$ < 0.0001 (**B**) and ##$P$ = 0.0016 (**C**) versus the CSDS CD8$^+$ T$_{SCM}$ cells+Vehicle group, $n$ = 8/group. The level of IFN-γ (**D**), TNF-α (**E**) and IL-2 (**F**) in plasma detected by ELISA. ***$P$ = 0.0006 (**D**), ***$P$ < 0.0001 (**E**) and **$P$ = 0.0078 (**F**) versus the control CD8$^+$ T$_{SCM}$ cells+Vehicle group, ###$P$ = 0.0001 (**D**), ###$P$ = 0.0003 (**E**) and #$P$ = 0.0102 (**F**) versus the CSDS CD8$^+$ T$_{SCM}$ cells+Vehicle group, $n$ = 8/group. The level of IFN-γ (**G**), TNF-α (**H**) and IL-2 (**I**) in brain tissue detected by ELISA. ***$P$ < 0.0001 (**G**), ***$P$ = 0.0001 (**H**) and **$P$ = 0.0025 (**I**) versus the control CD8$^+$ T$_{SCM}$ cells+Vehicle group, ##$P$ = 0.0074 (**G**), #$P$ = 0.0332 (**H**) and #$P$ = 0.0408 (**I**) versus the CSDS CD8$^+$ T$_{SCM}$ cells+Vehicle group, $n$ = 6/group. SB265610 treatment alleviated the astrocyte dysfunction induced by CD8$^+$ T$_{SCM}$ cells isolated from stressed mice. Representative images showing astrocyte immunostaining for GFAP in the *Rag1*$^{-/-}$ mouse hippocampus, followed by three-dimensional reconstruction and Sholl analysis (**J**); Quantification of GFAP-positive cells per square millimeter (**K**); Average branch number (**L**), total branch length (**M**), and total branch volume (**N**) in the hippocampus. **$P$ = 0.0055 (**K**), ***$P$ < 0.0001 (**L**), ***$P$ < 0.0001 (**M**) and ***$P$ < 0.0001 (**N**) versus the control CD8$^+$ T$_{SCM}$ cells+Vehicle group, #$P$ = 0.0286 (**K**), ###$P$ < 0.0001 (**L**), ###$P$ < 0.0001 (**M**) and ###$P$ < 0.0001 (**N**) versus the CSDS CD8$^+$ T$_{SCM}$ cells+Vehicle group, $n$ = 40 cells from 4 mice per group. SB265610 treatment inhibited pathological CD8$^+$ T$_{SCM}$ cell-induced microglial activation. Representative images showing microglial immunostaining for Iba-1 in the *Rag1*$^{-/-}$ mouse hippocampus, followed by three-dimensional reconstruction and Sholl analysis (**O**); Average soma size (**P**), branch number (**Q**), total branch length (**R**), and total branch volume (**S**). ***$P$ < 0.0001 (**P**), ***$P$ < 0.0001 (**Q**), ***$P$ < 0.0001 (**R**) and ***$P$ < 0.0001 (**S**) versus the control CD8$^+$ T$_{SCM}$ cells+Vehicle group, ###$P$ < 0.0001 (**P**), ###$P$ < 0.0001 (**Q**), ###$P$ < 0.0001 (**R**) and ###$P$ < 0.0001 (**S**) versus the CSDS CD8$^+$ T$_{SCM}$ cells+Vehicle group, $n$ = 40 cells from 4 mice per group. Golgi staining (**T**) and quantification (**U**) of dendritic spines in the hippocampal. ***$P$ < 0.0001 (**U**) versus the control CD8$^+$ T$_{SCM}$ cells+ Vehicle group, ###$P$ < 0.0001 (**U**) versus the CSDS CD8$^+$ T$_{SCM}$ cells+Vehicle group, $n$ = 40 neurons from 4 mice per group. **A**–**I**, **K**–**N**, **P**–**S**, **U** One-way ANOVA followed by two-sided Holm-Sidak post hoc multiple comparison test. **A**–**I**, **K**–**N**, **P**–**S**, **U** Data were presented as mean ± SD. Source data are provided as a Source Data file. CON control, CSDS chronic social defeat stress, CD8$^+$ T$_{SCM}$ cells stem cell-like memory CD8$^+$ T cells, SPT sucrose preference test, TST tail suspension test, FST forced swim test, GFAP glial fibrillary acidic protein, Iba-1 Ionized calcium-binding adapter molecule 1.

## Flow cytometry analysis

The prepared PBMCs were suspended in 1× PBS with 2% fetal bovine serum (FBS). Then, single cells were collected by centrifugated and incubated for 15 min with Blocking Reagent (Miltenyi Biotec). Finally, single cells were stained with fluorochrome-conjugated antibodies or their corresponding isotopic controls for 30 min at 4 °C in darkness.

To investigate the proportion of cell immune subsets in PBMCs of human, PBMCs samples were gated for T cells (CD45$^+$CD3$^+$), Myeloid cells (CD45$^+$CD3$^-$CD19$^-$HLA-DR$^+$), NK cells (CD45$^+$CD3$^-$CD56$^+$), B cells (CD45$^+$CD3$^-$CD19$^+$), regulatory CD4$^+$ T (CD4$^+$ T$_{reg}$) cells (CD45$^+$CD3$^+$ CD56$^-$CD4$^+$CD25$^+$CD127$^{low}$), CD4$^+$ Naïve T (T$_{Naïve}$) cells (CD45$^+$CD3$^+$ CD56$^-$CD4$^+$CD25$^-$CD45RA$^+$CCR7$^+$), CD4$^+$ central memory T (T$_{CM}$) cells (CD45$^+$CD3$^+$CD56$^-$CD4$^+$CD25$^-$CD45RA$^-$CCR7$^+$), CD4$^+$ effector memory T (T$_{EM}$) cells (CD45$^+$CD3$^+$CD56$^-$CD4$^+$CD25$^-$CD45RA$^-$CCR7$^-$), CD8$^+$ T$_{Naïve}$ cells (CD45$^+$CD3$^+$CD56$^-$CD8$^+$CD45RA$^+$CCR7$^+$CD95$^-$), stem cell-like memory CD8$^+$ T (CD8$^+$ T$_{SCM}$) cells (CD45$^+$CD3$^+$CD56$^-$CD8$^+$CD45RA$^+$ CCR7$^+$CD95$^+$), CD8$^+$ T$_{CM}$ cells (CD45$^+$CD3$^+$ CD56$^-$CD8$^+$CD45RA$^-$CCR7$^+$), CD8$^+$ T$_{EM}$ cells (CD45$^+$CD3$^+$CD56$^-$CD8$^+$CD45RA$^-$CCR7$^-$), and terminally differentiated effector memory CD8$^+$ T (CD8$^+$ T$_{EMRA}$) cells (CD45$^+$CD3$^+$ CD56$^-$CD8$^+$CD45RA$^+$CCR7$^-$).

To investigate the proportion of CD8$^+$T cell subsets in PBMCs of mice, samples were gated for CD8$^+$ T$_{Naïve}$ cells (CD45$^+$CD3$^+$NK1.1$^-$CD8$^+$ CD62L$^+$CD44$^-$Sca-1$^-$), CD8$^+$ T$_{SCM}$ cells (CD45$^+$CD3$^+$NK1.1$^-$CD8$^+$CD62L$^+$ CD44$^-$Sca-1$^+$), CD8$^+$ T$_{CM}$ cells (CD45$^+$CD3$^+$NK1.1$^-$CD8$^+$CD62L$^+$CD44$^+$), CD8$^+$ T$_{EM}$ cells (CD45$^+$CD3$^+$NK1.1$^-$CD8$^+$CD62L$^-$CD44$^+$), and tissue-resident memory CD8$^+$ T cells (CD8$^+$ T$_{RM}$) cells (CD45$^+$CD3$^+$NK1.1$^-$CD8$^+$ CD62L$^-$CD44$^-$).

All the above samples were sorted by the BD FACSCelesta$^{TM}$ Flow Cytometer. And data were acquired by FlowJo software 10.6.

## Chronic social defeat stress (CSDS) model

The CSDS procedure was performed as follows: C57BL/6J mice were exposed to social defeat stress from CD1 mice for a consecutive period of 10 days. Each intruder mouse (C57BL/6J) was introduced into the home cage of an unfamiliar resident CD1 aggressor mouse for a duration of 5–10 min and experienced physical defeat. This was followed by cohousing with the aggressor across a perforated plexiglass divider for 24 h. Control C57BL/6J mice were housed in pairs on either side of the plexiglass partition. The CSDS model is a well-established paradigm to investigate the neurobiology underlying stress following social trauma, specifically inducing depressive-like behavior in male mice[77,78]. Therefore, our study exclusively utilized male mice.

For homovanillic acid (HVA) or *Bifidobacterium scardovii* (*B. scardovii*) treatment experiment, mice were randomly divided into three group: (1) CON+vehicle group, (2) CSDS+vehicle group, (3) CSDS + HVA (300 mg/kg) or CSDS + *B. scardovii* ($1 \times 10^9$ colony-forming units [CFU]/day) group. After exposed to CSDS for 10 d, behavior experiments were performed.

## Chronic unpredictable stress (CUS) model

To induce chronic stress in mice, mice were first exposed to various, randomly scheduled, low-intensity social and environmental stressors 2–3 times a day for 4 weeks. The stressors included the following: (1) food or water deprivation for 24 h, (2) absence of sawdust in cage for 24 h, (3) forced swimming at 8 °C for 5 min, (4) sawdust moistened with water for 24 h, (5) overnight illumination, (6) tail nipping (1 cm from the tip of the tail), (7) physical restraint for 6 h, and (8) 45° cage-tilt along the vertical axis for 3 h.

## Social interaction test (SIT)

The SIT consisted of two stages. In the first stage, the experimental mice were given 2.5 min to explore a chamber measuring 40 cm wide by 40 cm high by 40 cm deep, containing an empty wire cage. In the second stage, an unfamiliar CD1 mouse was introduced into the wire cage and their interaction was recorded for another 2.5 min. The duration spent within the interaction zone surrounding the wire cages was meticulously recorded and subjected to analysis.

## Sucrose preference test (SPT)

SPT was used to test the preference of mice for sugar and evaluate anhedonia in mice. Briefly, it was conducted in three phases as follows: phase 1 habituation, phase 2 sucrose preference baseline, and phase 3 sucrose preference testing. In phase 1, 1% weight/volume sucrose solution was used in the cage for 3 d to habituate mice to the solution. In phase 2, each mouse was transferred to a single cage and was exposed to both tap water and sucrose solution for 24 h to obtain the sucrose preference baseline. In phase 3, sucrose preference was tested via a two-bottle choice test using standard bottles, one filled with tap water and one with 1% sucrose solution, supplied to mice for 24 h. The positions of the two bottles were switched every 6 h, and sucrose and water consumptions were simultaneously measured. The preference to consume sucrose solution was then calculated as percentage preference = [(sucrose intake/total intake) × 100]. Tests were performed by an individual blind to the animal's treatment status.

## Forced swim test (FST)

Mice were individually dropped into a cylinder (diameter: 20 cm; height: 25 cm) filled with 15 cm water and keep at 23–25 °C. After

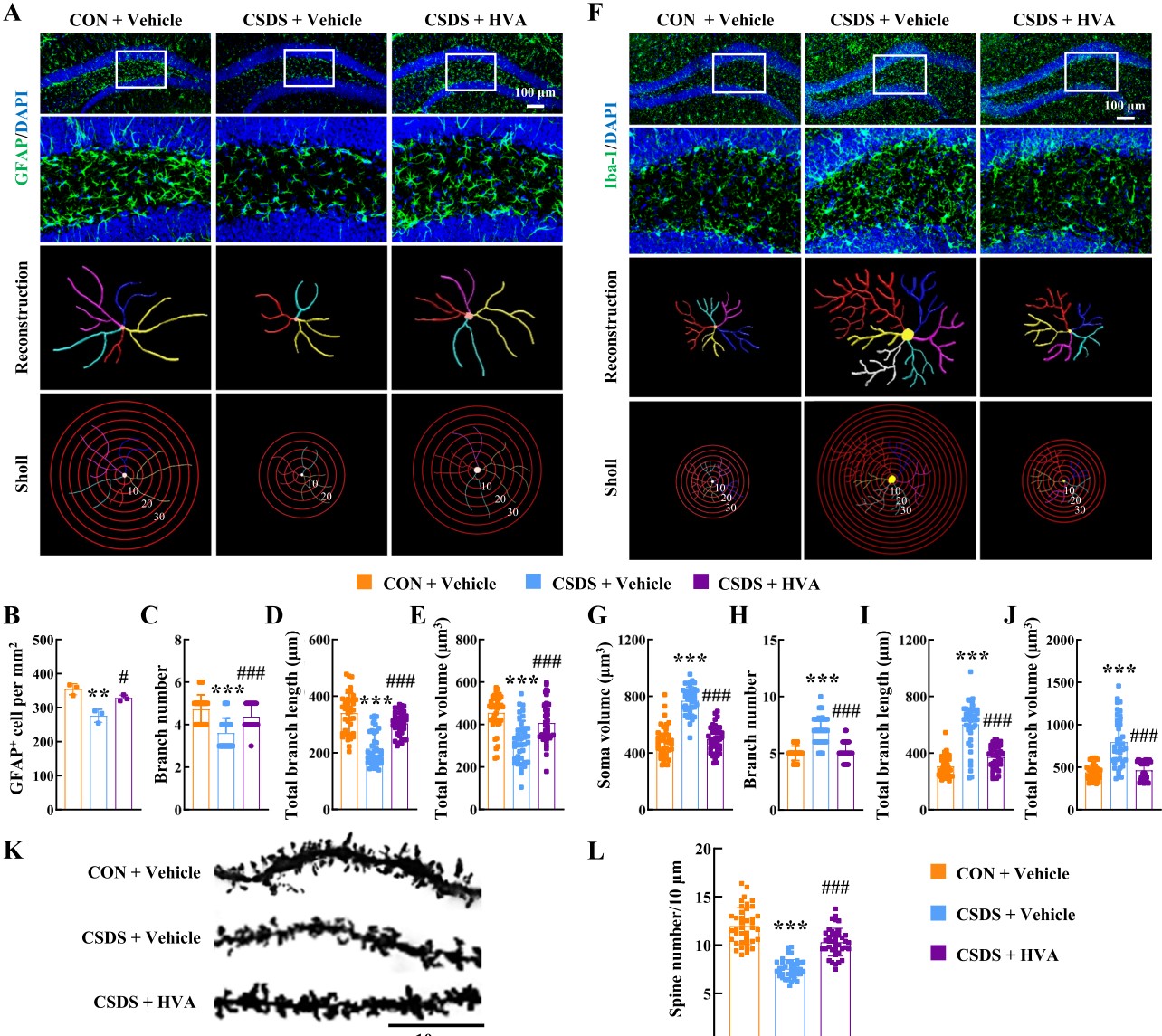

**Fig. 8 | HVA alleviates depression through inhibition of neuroinflammation and enhancement of neuronal plasticity.** HVA treatment alleviated the astrocyte dysfunction induced by CSDS. Representative images showing astrocyte immunostaining for GFAP in hippocampus, followed by three-dimensional reconstruction and Sholl analysis (**A**); Quantification of GFAP-positive cells per square millimeter (**B**); Average branch number (**C**), total branch length (**D**), and total branch volume (**E**) in the hippocampus. ***P** = 0.0025 (**B**), ****P** < 0.0001 (**C**), ****P** < 0.0001 (**D**) and ****P** < 0.0001 (**E**) versus the CON+Vehicle group, #*P* = 0.0126 (**B**), ###*P* < 0.0001 (**C**), ###*P* < 0.0001 (**D**) and ###*P* < 0.0001 (**E**) versus the CSDS +Vehicle group, *n* = 40 cells from 3 mice per group. HVA treatment inhibited CSDS-induced microglial activation. Representative images showing microglial immunostaining for Iba-1 in hippocampus, followed by three-dimensional reconstruction and Sholl analysis (**F**); Average soma size (**G**), branch number (**H**), total branch length (**I**), and total branch volume (**J**). ****P** < 0.0001 (**G**), ****P** < 0.0001 (**H**), ****P** < 0.0001 (**I**) and ****P** < 0.0001 (**J**) versus the CON+Vehicle group, ###*P* < 0.0001 (**G**), ###*P* < 0.0001 (**H**), ###*P* < 0.0001 (**I**) and ###*P* < 0.0001 (**J**) versus the CSDS +Vehicle group, *n* = 40 cells from 3 mice per group. Golgi staining (**K**) and quantification (**L**) of dendritic spines in the hippocampal. ****P** < 0.0001 (**L**) versus the CON +Vehicle group, ###*P* < 0.0001 (**L**) versus the CSDS +Vehicle group, *n* = 40 neurons from 4 mice per group. **B–E**, **G–J**, **L** one-way ANOVA followed by two-sided Holm-Sidak post hoc multiple comparison test. **B–E**, **G–J**, **L** data were presented as mean ± SD. Source data are provided as a Source Data file. CON control, CSDS chronic social defeat stress, HVA homovanillic acid, GFAP glial fibrillary acidic protein, Iba-1 Ionized calcium-binding adapter molecule 1.

engaging in vigorous activities in the first 2 min, the mice acquired an immobile posture, which were characterized by motionless floating in the water with only the necessary movements to keep their heads above water. The duration of immobility was recorded during the last 4 min of the test. Tests were performed by an individual blind to the animal's treatment status.

### Tail suspension test (TST)
The immobility time during TST was considered to mirror despair/depression-like behaviors. Mice were suspended 50 cm above the floor in an apparatus box (50 × 50 × 50 cm) by adhesive tape placed -1 cm

from the tip of the tail. Each test session lasted 6 min and was recorded. The first 2 min served as the habituation period. During the last 4 min, duration of immobility (hanging passively without body movement) was recorded. Tests were performed by an individual blind to the animal's treatment status.

### Mouse CD8⁺ T_SCM cell and EGFP cell isolation and labeling
Spleens from C57BL/6J or EGFP mice were harvested and gently pressed through a 70 μm filter to yield a single-cell suspension. Red blood cells were moved by incubation with ACK lysis buffer (C3702, Beyotime Biotech, China) for 5 min at 4 °C. Living spleen cells were

collected by centrifugation at $300 \times g$, 4 °C for 5 min. For sorting of CD8⁺ T$_{SCM}$ subset, cells were stained with FVS780 Viability staining solution and antibodies directed to CD3, CD8, CD62L, CD44, and Sca-1 in a run using the FACS Aria II SORP (BD Bio-Sciences, USA).

Intestine were removed the fat, Peyer's plaque and contents and separated. Make a longitudinal incision in the intestine, wash it with PBS and then cut it into 1 cm sections. Place the tissues in 50 mL centrifuge tubes, add HBSS without $Ca^{2+}/Mg^{2+}$ (containing 10 mM HEPES, 8% FBS, 4 mM EDTA and 0.5 mM DTT), ensure complete coverage, place at a 45° Angle on a 37 °C shaker and 250 rpm. for 20 min, and repeat 3 times. After each operation, vortex the sample for 20 s, collect the supernatant, and combine it. First, filter the supernatant through pre-wetted nylon cotton, followed by filtration through a 70 µm filter screen. Prepare two Petri dishes and add 20 mL of cold RPMI-1640 medium to each dish. After drying the tissue with absorbent paper, rinse it with cold RPMI-1640 and dry it again. Subsequently, wash the remaining tissues with PBS (without $Ca^{2+}/Mg^{2+}$) to remove EDTA. After chopping, transfer the tissues into a 50 mL centrifuge tube and add 5 mL of HBSS without $Ca^{2+}/Mg^{2+}$ (containing 10 mM HEPES, 5% FBS, and 0.2 mg/mL collagenase (D). Place the tube at a 45° angle on a 37 °C shaker and 250 rpm. for 40 min, shaking it once every 15 min to enhance digestion efficiency. Filter the resulting cell suspension through a 40 µm filter screen into a new centrifuge tube and wash it with 10 mL of PBS. The suspension of lamina propria monocytes and intestinal epithelial lymphocytes was centrifuged at 4 °C and 450 g for 10 min, and the supernatant was discarded. Lymphocytes were subsequently isolated using a freshly prepared Percoll separation solution (30%/70%), followed by centrifugation at 21 °C and 860 g for 20 min. The cell layer located at the interface was carefully aspirated. After resuspending the cells in 10 mL of HBSS, they were centrifuged again at 4 °C and $450 \times g$ for 10 min, and the supernatant was discarded. Finally, the cells were subjected to flow cytometry antibody staining for 30 min. Flow cytometry analysis was performed after washing.

### Mouse CD8⁺ T$_{SCM}$ cells adoptive transfer

Sorted CD8⁺ T$_{SCM}$ cells were spun down and washed with PBS and prepared for adoptive transfer experiments. On day 0, 3, and 6, $2 \times 10^5$ CD8⁺ T$_{SCM}$ cells isolated from control or CUS/CSDS mice were intravenously injected[56,79]. PBS control group simultaneously received an equal volume of sterile PBS. Starting from day 8, recipient mice were subjected to behavioral tests for 4 d[79].

For drug treatment experiment, the CXCR2 inhibitor SB265610 was intraperitoneally injected at a dose of 2 mg/kg, with 1% DMSO in saline applied as vehicle. SB265610 or vehicle was injected 30 min before the daily CD8⁺ T$_{SCM}$ cells adoptive transfer was applied.

### Mouse CD8⁺ T$_{SCM}$ cells depletion

After acclimatizing the mice for three days, CD8⁺ T cell depletion was achieved by intraperitoneal injection of the anti-CD8 neutralizing antibody (αCD8) (BioXcell, BE0061) and an isotype control IgG antibody at a dose of 500 µg on day 1, 3, and 5. The efficiency of depletion was assessed on day 7. In the second week, adoptive transfer was performed, followed by an additional injection of the depletion antibody on day 10 to maintain the depletion effect. Subsequent behavioral tests, including the SPT, TST, and FST, were conducted to evaluate the impact of CD8⁺ T cell depletion.

### Metagenomic sequencing

The total DNA was extracted from samples using the Fecal Genome DNA Extraction Kit (AU46111-96, BioTeke, China). Subsequently, DNA libraries were constructed utilizing the TruSeq Nano DNA Library Preparation Kit-Set (#FC-121–4001, Illumina, USA) following the manufacturer's instructions. Metagenome libraries were then sequenced on an Illumina NovaSeq 6000 platform with PE150 at LC-Bio Technology Co., Ltd. (Hangzhou, China). Fastp software (v0.23.4) was

employed to eliminate reads containing adapter contamination, low quality bases, and undetermined bases. Additionally, sequence quality was verified using fastp. Quality-filtered reads underwent initial alignment via bowtie (v2.2) to exclude host contaminations. The remaining reads were subjected to denovo assembly for each sample using MEGAHIT (v1.2.9), which facilitated microbial function and taxonomy assignment analysis. Metaprodiga (v2.6.3) was utilized for predicting coding regions (CDS) of assembled contigs, while CDS sequences of all samples were clustered through MMseq2 (v15-6f452) to obtain unigenes. DIAMOND (v 0. 9. 14) was used for taxonomic assessment of microbiota based on NR database. The Wilcox test was applied to identify differentially abundant species with significance declared at $P < 0.05$. An assignment of microbial functions was performed employing Kyoto Encyclopedia of Genes and Genomes (KEGG). Bioinformatic analysis was performed with ggplot2 (3.4.0) and pheatmap (1.0.12) using the OmicStudio tools at https://www.omicstudio.cn.

### Metabolomics sequencing

The collected samples were thawed on ice, and metabolites were extracted using 80% methanol buffer. Briefly, 50 mg of the sample was extracted with 0.5 mL of pre-cooled 80% methanol. The extraction mixture was then stored at −20 °C for 30 min. After centrifugation at $20,000 \times g$ for 15 min, the supernatants were transferred into a new tube and vacuum dried. The samples were reconstituted with 100 µL of 80% methanol and stored at −80 °C prior to LC-MS analysis. Additionally, pooled quality control samples were prepared by combining 10 µL from each extraction mixture.

The eluted metabolites from the column were detected using a high-resolution tandem mass spectrometer, Q-Exactive (Thermo Scientific). The Q-Exactive was operated in both positive and negative ion modes. Precursor spectra (70–1050 m/z) were collected at a resolution of 70,000 to achieve an AGC target of 3e6. The maximum injection time was set to 100 ms. Data acquisition was performed in DDA mode with a top 3 configuration for acquiring data. Fragment spectra were collected at a resolution of 17,500 to achieve an AGC target of 1e5 with a maximum injection time of 80 ms. To assess the stability of the LC-MS system throughout the entire acquisition process, a quality control sample (pooling all samples) was acquired after every 10 samples.

Proteowizard's MSConvert software is utilized for the conversion of raw mass spectrum data into a readable mzXML format. XCMS software is employed for peak extraction and quality control during the process. The extracted compounds are annotated using CAMERA, incorporating addition and ion information, followed by identification through metaX software. Primary mass spectrum information is used for initial identification, while secondary mass spectrum information is matched against an in-house standards database. Candidate compounds are further annotated using HMDB, KEGG, and other databases to elucidate their physicochemical properties and biological functions. Finally, metaX software facilitates quantification and screening of differential metabolites. Bioinformatic analysis was performed using the OmicStudio tools at https://www.omicstudio.cn. Pathway analyses were conducted using the pathway analysis and enrichment analysis modules in MetaboAnalyst 6.0 at www.metaboanalyst.ca.

### HVA quantification in human plasma

The level of HVA was quantified using ultra-high performance liquid chromatography tandem mass spectrometry (UHPLC-MS/MS). Plasma samples from pHCs and patients with MDD were thawed, mixed with isotopically labeled internal standard solutions, and then transferred to new tubes after vortexing and centrifugation at $29,700 \times g$ for 10 min at 4 °C. For further derivatization, a cryogenic vacuum centrifugal enrichment and evaporation system (CentriVap Labconco) was employed. A mixture of 50 µL dansyl chloride solution (2 mg/mL, diluted with acetone) and sodium bicarbonate buffer (pH = 9.0, 0.2 M)

was added to the dry matter in 1:1 volume ratio, followed by vortexing for 1 min. After water-bath heating, evaporation, re-dissolution, and centrifugation steps were performed, a supernatant of 5 µL was injected.

## Immunofluorescence staining and image analysis

30-µm thickness frozen sections containing the intact hippocampus were prepared with a cryostat and incubated subsequently with 0.3% Triton X-100 for 25 min, and then blocked with 10% NGS in 0.3% Triton X-100 solution for 1 h at room temperature. Next, the sections were incubated with anti-GFAP antibody (1:400, G3893, Sigma-Aldrich, USA) or a rabbit anti-Iba-1 antibody (1:250, 019-19741, Wako Pure Chemicals, Japan) overnight at 4 °C. On the following day, the sections were washed and incubated with Alexa 488-conjugated goat anti-mouse IgG (1:300, A11001, Invitrogen) or anti-rabbit IgG (1:300, A32731, Invitrogen, USA) in 1 × PBS for 1 h at room temperature. Images were captured using a microscope (ImagerM2, Zeiss, Germany). After a final washing with 1 × PBS, the sections were mounted onto glass slides, and ProLong Gold Anti-fade reagent containing DAPI (0100-20, SouthernBiotech, USA) was applied for visualization of nuclei. Immunofluorescence images were captured by microscopy (Olympus DP73, Olympus, Japan). Average intensities of GFAP were calculated using Image J software (NIH) by sampling a 28 × 28 pixel area and capturing 36 images from 6 consecutive sections. The values were reported as the average intensity above the background ± SD. Computer-based cell tracing software Neurolucida 360 (MBF Bioscience, USA) was used for three-dimensional (3D) reconstruction of GFAP or Iba-1 positive cells within the hippocampus of mice. NeuroExplorer (MBF Bioscience, USA) was used to analyze 40 cells per animal. Sholl analysis was used to determine branch tree morphology by placing three-dimensional concentric circles in 5 mm increments starting at 5 mm from the soma.

The paraffin-embedded intestinal tissue sections of *Rag1*$^{-/-}$ mice were incubated in xylene for 2 times, 2 min each time, and the sections were degummed. The slides were incubated at the ratio of 100%, 95%, 80%, 70% and 50% successively for 3 min, and then water was added until the slices were transparent. Rinse with 1× PBS twice, 5 min each time, boil in 10 mM improved citrate antigen retrieval solution (P0083, Beyotime Biotechnology, pH = 7.0) for 20 min, after cooling, rinse with PBS for 5 min × 2. The blocking solution (10% NGS, 0.3% Triton X-100, PBS) was closed at room temperature for 2 h, incubated with CD8a Monoclonal Antibody (1:50, 42-0081-82, Thermo fisher, USA) overnight, rinsed with 1× PBS the next day, sealed with DAPI, and fluorescence imaging and statistics were performed.

## Western blot

Lysates were harvested from hippocampus and colon tissues in RIPA lysis buffer (P0013B, Beyotime, China) containing a protease inhibitor cocktail. Equal amounts of protein were separated via sodium dodecyl sulfate-polyacrylamide gel electrophoresis, and then transferred to polyvinylidene fluoride membranes electrophoretically. After incubation with blocking buffer, the membrane was incubated with antibodies against ZO-1 (1:1000, 21773-1-AP, Proteintech Group, China), Occludin (1:1000, 27260-1-AP, Proteintech Group, China), Claudin-5 (1:1000, AF5216, Affinity, USA), SYN1 (1:500, 20258-1-AP, Proteintech Group, China), BDNF (1:500, 28205-1-AP, Proteintech Group, China), and β-actin (1:2000, 60008-1-AP, Proteintech Group, China) overnight at 4 °C. After 3 washes, the membrane was incubated with a horse-radish peroxidase-conjugated goat anti-mouse/rabbit IgG secondary antibody (1:2000, 7076P2/7074P2, Cell Signaling, USA). Signals were detected by Automatic Chemiluminescence Image Analysis System (Tanon 5200, Tanon Science & Technology, China). Quantification of the individual protein bands was performed via densitometry using Image J software.

## vDISCO whole body immunolabeling, clearing, and imaging

The mouse bodies were placed inside a 300 mL glass chamber (Omnilab, 5163279) and filled with the appropriate solution according to the protocol to cover the entire body of the animal (250–300 mL). A transcardial circulator system was established to enable peristaltic pumping of the solutions (ISMATEC, REGLO Digital MS-4/8 ISM 834; reference tubing, SC0266), with a pressure set at 180–230 mmHg (50–60 rpm). The tubing was positioned to allow pumping of the solutions through the heart using a perfusion needle (Leica, 39471024) into the vasculature via the same entry point used for PBS and PFA perfusion steps described above. The other end of the tube was immersed in the chamber with an open end to facilitate the suction of the solution into the body. Initially, decolorization solution (25% CUBIC reagent 1 consisting of 25 wt% urea (Carl Roth, 39413), 25 wt% N,N,N′,N′-tetrakis (2-hydroxypropyl) ethylenediamine (Sigma-Aldrich Co., LLC.,122262), and 15 wt% Triton X-100 (AppliChem GmbH, A49751000) in 0.1 M PBS) was used for perfusing samples over a period of two days while refreshing solutions every 12 h. Samples were washed with PBS for 3 × 2 h. Then, decalcification solution (10 w/v% EDTA in 0.01 PBS, pH~8–9, Carl Roth, 1702922685) was perfused for 2 d followed by half a day with permeabilization solution composed of 0.5% Triton X-100, 1.5% goat serum (GIBCO, 16210072), 0.5 mM of Methyl-beta-cyclodextrin (Sigma, 332615), 0.2% trans-1-Acetyl-4-hydroxy-L-proline (Sigma, 441562), 0.05% sodium azide (Sigma, 71290) in 0.01 M PBS. The setup was adjusted to initiate the PI labeling and boosting. The free end of the perfusion tube was connected to a 0.22 µm syringe filter (Sartorius, 16532) and an infrared lamp (Beuer, IL21) was aimed at the chamber to enable the solution's temperature to be around 26–28 °C. This setup was then left running for 6 d after the addition of 35 µL of nanobooster (stock concentration 0.5–1 mg/mL) which was added directly into the refreshed permeabilization solution. Next, the body was placed into a 50 mL tube (Falcon, 352070), with the same permeabilization and labeling solution, and an extra 5 µL of nanobooster was added. The tube was then put on a shaker at room temperature (RT) for 2 d for labeling. Atto647N conjugated anti GFP nanobooster (Chromotek, gba647n-100) was used to boost the signal from the EGFP mice in the study. Then, the animals were placed back into the initial perfusion setup, where washing solution was perfused for 2 × 12 h, which was composed of; 1.5% goat serum, 0.5% Triton X-100, 0.05% of sodium azide in 0.1 M PBS. 0.1 M PBS was used to wash the sample 3 × 2 h. 3DISCO protocol was applied for whole body clearing[80]. The animals were demounted from the perfusion system but kept in glass chambers and placed on top of shakers (IKA, 2D digital) at RT inside a fume hood. Glass chambers were sealed with parafilm and covered with aluminum foil along with the 3DISCO application. For dehydration, sequential immersion of tetrahydrofuran (THF) (Sigma,186562) (50 vol% THF, 70 vol% THF, 80 vol% THF, 100 vol% THF and again 100 vol% THF) was applied every 12 h. Then 3 h of dichloromethane (DCM) (Sigma, 270997) immersion for delipidation was followed by indefinite immersion in BABB (benzyl alcohol + benzyl benzoate 1:2, Sigma, 24122 and W213802) solution for refractive index matching.

## RNA sequencing analysis

Whole RNA-sequencing analysis was completed by LC-Bio (Hangzhou, China). Briefly, hippocampus tissue of *Rag1*$^{-/-}$ mice transplanted intravenously with CON CD8$^+$ T$_{SCM}$ cells or CUS CD8$^+$ T$_{SCM}$ cells was collected in TRIzol reagent (9109, TaKaRa, Japan). UMI technology was used to label each sequence fragment with sequence tags, which minimized the interference of duplication generated by PCR amplification on the quantitative accuracy of the transcriptome. RNA sequencing reads were aligned to the mouse genome (GRCh37/hg19) using the software Hisat2 (2.0.4). Transcript abundance was quantified as fragments per kilo base of exon per million fragments mapped (FPKM). The threshold of significantly differential expression was set at

$P < 0.05$ and $|\log_2(\text{fold change})| \geq 1$. The volcano plot and pathway enrichment analysis of differential genes were performed using the OmicStudio tools at https://www.omicstudio.cn/tool. Cytoscape software (v3.4.0) was then used to analyze the interactions of the DEGs.

## Enzyme-linked immunosorbent assay (ELISA)

The plasma and hippocampus of recipient $Rag1^{-/-}$ mice were isolated for cytokine detection. The plasma was centrifuged for 15 min at $1000 \times g$, 4 °C. The level of HVA was analyzed using commercial ELISA kits according to the manufacturer's instructions. The ELISA kit of HVA (LA182246H), TNF-α (LA128801H), IL-2 (LA128808H), and IFN-γ (LA128805H) were purchased from Nanjing Lapuda Biotechnology Co., Ltd.

## Microbial strains cultivation

*B. Scardovii* were purchased from the DSMZ (DSM 13734). The strains were cultured using the DSMZ medium 104 at 37 °C under anaerobic conditions (80% $N_2$, 10% $CO_2$, 10% $H_2$). For the animal experiment, bacteria were enriched by centrifuging at $600 \times g$ for 5 min at room temperature and resuspended in oxygen-free sterilized PBS.

## qPCR

RNA was extracted using VeZol Reagent (R411-01, Vazyme, China) according to the manufacturer's protocol. Reverse transcription was performed with a HiScript Q RT SuperMix for qPCR Kit (R123-01, Vazyme, China) and quantified using SYBR Green qPCR Master Mix (Q141-02, Vazyme, China) following the manufacturer's instructions in an Applied Biosystems qPCR System (StepOne, Thermo Fisher, USA). GAPDH was used as an internal control. The primers used to amplify mRNA transcripts were synthesized by Invitrogen. The primers were as follows: *TBX21* (F: AGGCTGAGTTTCGAGCAGTC, R: TGGCCTCGGTAG-TAGGACAT); *GNLY* (F: TCTCTCGTCTGAGCCC, R: GCAGCATTGGAAA-CACT); *PRF1* (F: GGGATTCCAGAGCCCAAGTG, R: GTGTGTCCACTGGG-AAGGAG); *GAPDH* (F: ACCATCTTCCAGGAGCGAGAT, R: GGGCAGAGA-TGATGACCCTTT); *Gzmb* (F: AGGACAACACTCTTGACGCTG, R: CCGAT-GATCTCCCCTGCCTTT); *Gzmh* (F: GAGCTTCCTTTGAGGAGGATTC, R: TCAGTTTGCCCGTAGGAGAC); *Nkg7* (F: TCAAGTCCAGACATTCTTCTC-CT, R: CACAAGGTTTCATACTCAGCCC); *Fcgr3A* (F: CCATCACTGTCCA-AGACCCAG, R: GTGTCCACTGCAAACAGGAG); *Klrd1* (F: ACTTCTCAT-GGCAGTTTCTAGG, R: CCAGGCAAACACAGCATTCA); *Hopx* (F: GAGCA-GACGCAGAAATGGTT, R: CCTGGCTCCCTAGTCCGTAA); *Prf1* (F: TTGG-TGGGACTTCAGCTTTCC, R: TCCATACACCTGGCACGAAC); *Clic3* (F: C-AGCTCTTTGTGAAGGCCAGT, R: GCACACCCTTGAGGAGTAGG); *Cx3cr1* (F: GAGTATGACGATTCTGCTGAGG, R: CAGACCGAACGTGAAGAC-GAG); *Ccl4* (F: AACCTAACCCCGAGCAACAC, R: AGGGTCAGAGCCCA-TTGGTG); *Adgrg1* (F: GTTTTCTCTGGTGCAAGGTGC, R: CTAGGCTCT-GGGAAGAAGCG); *Cst7* (F: TGCCTTGAAGCGGACTCTAT, R: AGTGGG-GAAAACACGAGTGG); *Spon2* (F: GAAGAGCTGGGGCCTATCTG, R: ACG-TTTTCCATCACCTGGCAA); *Efhd2* (F: TTCTTCGAGGCCAAGGTACAG, R: AGTCTCCCACAGATCCCCTG); *Samd3* (F: GCTATTGAAGCTGACG-GATGC, R: CCCACTGACTTCTTCCTCTTGA); *Tgfbr3* (F: TGTGTTCCTG-CTCAACTCCC, R: AACCCTCCGAAACCAGGAAG); *Fcrl6* (F: CACCTC-CAGCTCAGAACTACC, R: CTTCAGCCATGGACTCACAGA); *S1pr5* (F: TG-CCGGTTACAGGAGACTTTTG, R: GAGCTTGCCGGTGTAGTTGT); *Tbx21* (F: CACTAAGCAAGGACGGCGA, R: TGTAATGGCTTGTGGGCTCC); *Lgals1* (F: CTTCAATCATGGCCTGTGGTC, R: GTAGGCACAGGTTGTTG-CTG); *Ttc38* (F: TCAGTACGTCAAGTGGACCAA, R: AAGGCCGTTAGAGA-TGGCAAG); *Prss23* (F: GGTGAGTCCCTACACCGTTC, R: GGCGTCGAAG-TCTGCCTTAG); *Rps26* (F: GCCCAAGGATAAGGCCATCA, R: GAAG-CACGTAGGCGTCGAA); *Ypel1* (F: ATGGTGAAAATGACCAAGTCCAA, R: CAAAGGCATGTTCGTATTTCCAC); *Fgr* (F: ACCCCCAACAAGGAACC-AAG, R: ATCCCAAAGGACCACACGTC); *Cep78* (F: ATGGCTGGCATA-GATCAGTCA, R: GGTGGCAGTCTGTTTCGGAT); *C1orf21* (F: CTCCGGG-GTTTTCGGCTT, R: GTGTGTACACCTTGGGCG); *Gapdh* (F: AGGTCGG-TGTGAACGGATTTG, R: TGTAGACCATGTAGTTGAGGTCA).

## Statistics

The statistical analyses used for the different experiments were described in the respective figure legends. Power analysis software (Hintze, J. NCSS, LLC., Kaysville, Utah, USA. www.ncss.com) was used to calculate the sample power for the clinical study. For comparison between two groups, the Shapiro-Wilk test was used to assess normality in distribution for each group. If the Shapiro-Wilk test passed, the two groups were compared using the Student's *t* test (two-sided). When the Shapiro-Wilk test failed, the Mann-Whitney *U* test (two-sided) was used for the statistical analysis of two groups. The Chi-square test was performed to compare discrete variables, such as gender, marital status, and family history. The differential gene expressions from scRNA-seq were examined by the Benjamini-Hochberg procedure. For comparison of three groups, the Shapiro-Wilk test was performed to determine the variances between multiple groups. One-way ANOVA followed by the two-sided Holm-Sidak test was used for parametric comparisons, while the Kruskal-Wallis test followed by the two-sided Dunn's test was used for nonparametric comparisons. Statistical analysis was performed using GraphPad Prism 8.0.2 Software. For correlations, the Shapiro-Wilk test was used to assess normality of data distribution, and then the two-sided Pearson's correlation coefficient was used for data with normal distribution, while the two-sided Spearman's correlation coefficient was used for data with non-normal distribution. Statistical analysis was performed using SPSS Statistics 22. The results were considered statistically significant if $P < 0.05$. All data were presented as mean ± SD.

## Ethics statement

The study procedure was approved by the Clinical Research Ethics Committee of ZhongDa Hospital, Southeast University (ID: 2020ZDSYLL247-P01 and 2022ZDSYLL046-P01). A prospective multi-cohort clinical study (ChiCTR2200061705 and ChiCTR2100051157) was registered on the Chinese Clinical Trial Registry (http://www.chictr.org.cn). This study was in accordance with the international human research ethical standards as stated in the Helsinki Declaration and the relevant clinical research guidelines and management practices in China. All participants or their legally authorized representatives submitted written informed consent to participate in this study.

## Reporting summary

Further information on research design is available in the Nature Portfolio Reporting Summary linked to this article.

# Data availability

The scRNA-seq data generated in this study have been deposited in the National Genomics Data Center database under accession code PRJCA009116 (HRA002269) [https://ngdc.cncb.ac.cn/bioproject/browse/PRJCA009116]. The metagenomics data generated in this study have been deposited in the National Genomics Data Center database under accession code PRJCA029686 (HRA008457, human microbiota) [https://ngdc.cncb.ac.cn/bioproject/browse/PRJCA029686] and PRJCA029610 (mouse microbiota) [https://ngdc.cncb.ac.cn/bioproject/browse/PRJCA029610]. The metabolomics data used in this study are available in the National Genomics Data Center database under accession code PRJCA029658 (mouse brain tissue) [https://ngdc.cncb.ac.cn/bioproject/browse/PRJCA029658] and PRJCA029659 (mouse microbiota) [https://ngdc.cncb.ac.cn/bioproject/browse/PRJCA029659]. The raw data of RNA-seq are accessible on Gene Expression Omnibus using the accession number GSE200957. All the data supporting this study are available within the article, the Supplementary file, and the Source data file, as indicated in the Reporting summary for this article. Source data are provided with this paper.

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

## Acknowledgements

This work was supported by grants from the Science and Technology Innovation 2030-Major Project of the Ministry of Science and Technology of China (2021ZD0202904/2021ZD0202900 to H.Y.), the National Natural Science Foundation of Distinguished Young Scholars (82025033 to H.Y.), the National Natural Science Foundation of China (82230115 to H.Y., 82373857 to Y.Z., 82473903 to B.H., 82273914 to Y.B., and 82304471 to Y.C.), the Major Science and Technology Projects in Jiangsu Province (BG2024045 to H.Y.), the Basic Research Program of Jiangsu (BK20253013 to H.Y. and B.H.), the Natural Science Foundation Outstanding Youth Fund of Jiangsu Province (BK20240175 to B.H.), the Noncommunicable Chronic Diseases-National Science and Technology Major Project (2024ZD0526600/2024ZD0526601/2024ZD0526603 to H.Y.), the Nanjing U35 Strong Foundation Program to B.H., and the Jiangsu Provincial Key Laboratory of Critical Care Medicine (JSKLCCM-2022-02-008 to Y.B.).

## Author contributions

H. Yao conceived and supervised this project and wrote the manuscript. B. Han performed analyses of scRNA-seq, RNA-seq, and intercellular communication. Y. Yuan designed the clinical research. Y. Zhang performed western blot, ELISA, adoptive transfer, flow cytometry analysis, immunostaining, metabolome analysis, metagenomic analysis, clinical sample analysis and wrote the manuscript. M. Ju. W. Yang performed CSDS and CUS mouse model, behavioral tests, adoptive transfer, PBMC isolation, and flow cytometry analysis. Y. Cai performed metabolome analysis. X. Yu performed flow cytometry analysis. S. Chen, G. Chen, Z. Shen, and Y. Li recruited MDD patients and collected clinical samples. Y. Bai contributed to the flow cytometry sorting assay. H. Ren and L. Shen contributed to adoptive transfer, CUS, and behavioral tests. J. Li and P. Shi contributed to the experiment design and the discussion of the manuscript.

## Competing interests

The authors declare no competing interests.
