## [Transparent Peer Review file · Nature Communications]

Migration of CD8⁺ TSCM cells into intestine via PPBP–CXCR2 axis increases host stress susceptibility by inhibiting gut microbiome-derived HVA

Corresponding Author: Professor Honghong Yao

Version 0:

Reviewer comments:

Reviewer #1

(Remarks to the Author)

I appreciate the authors' comprehensive and meticulous responses to my previous comments. The inclusion of additional data and figures has notably strengthened the manuscript and effectively addressed my primary concerns. Overall, I have no significant remaining issues. Only a few minor points require clarification:

The authors identified distinct molecular signatures in pathological CD8⁺ T stem cell memory (TSCM) cells, including increased cytotoxic potential evidenced by upregulation of GZMB, GZMH, and TBX21, as well as enhanced chemotactic capabilities indicated by elevated expression of CX3CR1, CCL4, and CCL4L2. It would be valuable for the authors to elaborate on the potential mechanisms by which depressive states may influence or drive the emergence of this specific CD8⁺ TSCM cell subset, especially in contrast to those observed under normative conditions. Exploring the pathways linking mood disorders to immune cell differentiation and function could provide deeper mechanistic insights.

In Figure 3O, annotating the significantly differentially expressed genes directly within the volcano plot—either through labels or color coding—would greatly improve interpretability and facilitate rapid identification of key genes of interest.

The terminology “HVA in microbiota” and “Total amount of HVA” used in Figures 5D and 5H are somewhat ambiguous. Clarifying these terms—perhaps by defining what HVA represents (e.g., specifying whether it refers to microbial-derived or host-derived HVA, and detailing the measurement method—either in the figure legends or in the main text, would enhance clarity and reader understanding.

Reviewer #2

(Remarks to the Author)

The authors responded to my concerns raised previously.

Reviewer #3

(Remarks to the Author)

The authors have vigorously addressed the reviewer's concerns and the manuscript is recommended for publication.

Response to the Editor and Reviewers:

We would like to thank you for the insightful reviews of our manuscript entitled “*Migration of CD8⁺ TSCM cells into intestine via PPBP–CXCR2 axis increases host stress susceptibility by inhibiting gut microbiome-derived HVA*”. We greatly appreciate the positive and constructive comments. Following the recommendation of the editor of *Nature Metabolism*, we would like to transfer our manuscript (original processing number NATMETAB-A240912787) along with the updated point-by-point rebuttal letter to *Nature Communications*. The new manuscript number is NCOMMS-25-45334-T. We performed substantial additional experiments and analysis to effectively address the issues raised by the reviewers, and prepared a point-by-point response to reviewer’s comments. Thank you again for this opportunity, we kindly invite you to review our point-by-point response detailed below.

SUMMARY prior to our point-by-point responses:

Prior to our full point-by-point responses to all of the Reviewers’ comments (below), we here present a very brief summary describing the major aspects of our revision process and detailing the experiments and analysis we have undertaken to properly address all of the points raised in the initial review of our manuscript. Fundamentally, the entire teams of multiple labs have been busy at work on our revision over the last several months, and we now have both confirmatory and exciting new data which enables us to respond fully to the Reviewers’ concerns. We are extremely grateful for the careful reviews and exceedingly helpful guidance and critiques from the reviewers, which have (in our view) obviously improved our study dramatically. We trust you’ll agree that our revised manuscript and our point-by-point responses are substantially improved and now represent an exciting and rigorous advance worthy of publication in the field-leading journal *Nature Communications*.

New experiments and analysis conducted since our initial submission includes the following:

1. Key genes and pathways driving depression in stem cell-like memory CD8⁺ T (TSCM) cells

- Purpose: To determine the key molecules associated with the distinct ability of CD8⁺ T_{SCM} cells to drive depression.
- Experiments and Analyses:
 - i) The scRNA-seq analysis to identify differentially expressed genes (DEGs) in CD8⁺ T_{SCM} cells from major depressive disorder (MDD) patients and psychiatrically healthy controls (pHCs) (**in response to Reviewer 1, Critique 1**).
 - Result: 33 specific DEGs were identified in CD8⁺ T_{SCM} cells from MDD patients (**Supplementary Figure 4A**).
 - ii) KEGG enrichment analysis revealed the signaling pathway (**in response to Reviewer 1, Critique 1**).
 - Result: 33 specific DEGs from CD8⁺ T_{SCM} cells in MDD patients were mainly involved in the chemokine signaling pathway (**Supplementary Figure 4B**).
 - iii) PPI network and cytoHubba analysis determined the hub genes in CD8⁺ T_{SCM} cells (**in response to Reviewer 1, Critique 1**).

Result: the hub genes *TBX21*, *PRF1*, and *GNL1* might be key molecules regulated the function of CD8⁺ T_{SCM} cells (**Supplementary Figure 4C-F and Supplementary Table 12**).

- iv) Quantification of the expression of hub genes in CD8⁺ T_{SCM} cells (**in response to Reviewer 1, Critique 1**).

Result: The expression of *TBX21*, *PRF1*, and *GNL1* were significantly increased in CD8⁺ T_{SCM} cells from MDD patients compared to pHCs (**Supplementary Figure 4G-I and Supplementary Table 13**).

- v) Using scRNA-seq data to analyze population changes in CD8⁺ T_{SCM} cell subsets in MDD patients and pHCs (**in response to Reviewer 3, Critique 6**).

Result: The population of CD8⁺ T_{SCM} cells was significantly increased in MDD patients compared to pHCs (**Figure 2D**).

2. Effect of dopamine on depression

- Purpose: To investigate the effect of dopamine level on depression
 - Experiments and Analyses:
- i) The hippocampal dopamine levels in CSDS mice and *Rag1*^{-/-} recipient mice. (**in response to Reviewer 1, Critique 3**).

Result: No significant difference in dopamine levels between CSDS mice and control mice. Consistently, *Rag1*^{-/-} mice that received CD8⁺ T_{SCM} cells either from CSDS or control mice exhibited no significant differences in hippocampal dopamine levels. Pharmacological experiments further confirm that HVA or CXCR2 inhibitor SB265610 treatment does not affect dopamine levels (**Response Document Figure 1A-B**).

3. Effect of homovanillic acid (HVA) on depression

- Purpose: To investigate the therapeutic effect and mechanism of HVA on depression
 - Experiments and Analyses:
- ii) Behavior phenotype test to demonstrate the therapeutic effect of HVA on depression (**in response to Reviewer 1, Critique 3; Reviewer 2, Critique 1; Reviewer 3, Critique 5**).

Result: HVA inhibited the sucrose uptake reduction in the social preference test (SPT) and the immobility time prolongation of the tail suspension test (TST) and forced swim test (FST) induced by chronic social defeat stress (CSDS) (**Figure 5I-L**).

- iii) Immunostaining and three-dimensional (3D) reconstruction of astrocytes and microglia in hippocampus (**in response to Reviewer 2, Critique 1; Reviewer 3, Critique 5**).

Result: HVA treatment alleviated astrocyte dysfunction and inhibited microglial activation induced by CSDS (**Figure 8A-J**).

- iv) Golgi staining of neurons and western blot analysis of presynaptic membrane protein synapsin-1 (SYN1) and mature brain-derived neurotrophic factor (mBDNF) in hippocampus (**in response to Reviewer 2, Critique 1; Reviewer**

3, Critique 5).

Result: HVA treatment significantly upregulated the spine number of neurons and the expression of SYN1 and mBDNF in hippocampus of CSDS mice (**Figure 8K-L and Supplementary Figure 15B**).

- v) RNA-seq of brain tissue in CSDS mice with HVA treatment (**in response to Reviewer 1, Critique 9**).

Result: 41 genes, several are closely associated with depression and brain function, such as *Pon1* and *Slc16a4* (**Response Document Figure 4A-C**).

4. Mechanism of CD8⁺ T_{SCM} cells inducing depression

- Purpose: To investigate the mechanism of CD8⁺ T_{SCM} cells inducing depression.
- Experiments and Analyses:
 - i) Measurement of CD8⁺ T_{SCM} cells in the colon of the *Rag1*^{-/-} recipient mice (**in response to Reviewer 1, Critique 5; Reviewer 3, Critique 1 and 4**).

Result: the higher presence of CD8⁺ T_{SCM} cells in the colon of the *Rag1*^{-/-} mice received pathological CD8⁺ T_{SCM} cells (**Figure 4J**).
 - ii) Measurement of CD8⁺ T_{SCM} cell-related proinflammatory cytokines in plasma and colon (**in response to Reviewer 1, Critique 5; Reviewer 3, Critique 1**).

Result: The plasma levels of IFN- γ , TNF- α , and IL-2 were higher in *Rag1*^{-/-} mice received pathological CD8⁺ T_{SCM} cells than those received control CD8⁺ T_{SCM} cells. The levels of IFN- γ , TNF- α , and IL-2 were higher in the colon of *Rag1*^{-/-} mice received pathological CD8⁺ T_{SCM} cells than those received control CD8⁺ T_{SCM} cells, while these changes were inhibited by CXCR2 inhibitor (SB265610) treatment (**Supplementary Figure 11A-C and F-H**).
 - iii) HE staining of intestinal tissue and western blot analysis of tight junction proteins in colon (**in response to Reviewer 1, Critique 5; Reviewer 3, Critique 1**).

Result: The *Rag1*^{-/-} mice received CD8⁺ T_{SCM} cells from stressed mice exhibited the increase of inflammatory cell infiltration and mucosal damage overall histology score, along with the decreased expression of tight junction proteins (Occludin, ZO-1, and Claudin-5) in colon (**Supplementary Figure 11D-E**).
 - iv) Measurement of proinflammatory cytokines *in vitro* using transwell system (**in response to Reviewer 1, Critique 5; Reviewer 3, Critique 1**).

Result: The pathological CD8⁺ T_{SCM} cells elevated levels of IFN- γ , TNF- α , and IL-2 in the lower chamber of transwell system, while transfected CD8⁺ T_{SCM} cells with PPBP knockdown lentivirus decreased those proinflammatory cytokines levels (**Supplementary Figure 11I-L**).

5. Role of neutrophils in depression

- Purpose: To investigate the impact of systemic blocking the PPBP-CXCR2 axis on neutrophils.
- Experiments and Analyses:
 - i) Flow cytometry analysis of neutrophil proportions in blood between CON and CSDS groups (**in response to Reviewer 1, Critique 6; Reviewer 3, Critique 3**).

Result: There was a significant increase in the proportion of neutrophils in the CSDS group compared to controls (**Response Document Figure 2A**).

- ii) Immunostaining of neutrophils in intestinal tissues (**in response to Reviewer 3, Critique 3**).

Result: There was no significant difference in neutrophil populations in colon between $Rag1^{-/-}$ mice receiving control $CD8^{+} T_{SCM}$ cells and those receiving pathological $CD8^{+} T_{SCM}$ cells, while treatment with the CXCR2 inhibitor (SB265610) did not alter the number of neutrophils in colon (**Response Document Figure 2B**).

6. Role of macrophages in depression

- Purpose: To investigate the phenotypic changes in macrophages in depression.
- Experiments and Analyses:
 - i) scRNA-seq to analyze the macrophages (**in response to Reviewer 1, Critique 6; Reviewer 3, Critique 3**).

Result: Based on established macrophage markers, only a subset of monocytes exhibit certain macrophage-like characteristics; however, a distinct macrophage subpopulation cannot be defined in peripheral circulation (**Response Document Figure 3A-C**).

7. Mechanism of $CD8^{+} T_{SCM}$ cell migration

- Purpose: To investigate the role of the PPBP-CXCR2 axis in $CD8^{+} T_{SCM}$ cell migration.
- Experiments and Analyses:
 - i) Measurement of PPBP expression in $CD8^{+} T_{SCM}$ cells (**in response to Reviewer 3, Critique 3**).

Result: The expression of PPBP was significantly higher in $CD8^{+} T_{SCM}$ cells from stress-susceptible mice compared to those from control or resilient mice (**Figure 4G**).

- ii) Measurement of $CD8^{+} T_{SCM}$ cells migration *in vitro* using transwell system (**in response to Reviewer 1, Critique 7; Reviewer 3, Critique 3**).

Result: $CD8^{+} T_{SCM}$ cells from stressed mice more migrated to the lower chamber than the cells from control groups (**Figure 4K-M**).

- iii) Intervention of CXCR2 inhibitor and PPBP knockdown lentivirus (**in response to Reviewer 1, Critique 7; Reviewer 3, Critique 3**).

Result: The treatment of SB265610 and PPBP knockdown lentivirus significantly inhibited $CD8^{+} T_{SCM}$ cell migration (**Figure 4K-M**).

8. Effects of $CD8^{+} T_{SCM}$ cells on HVA levels, neuroinflammation and synaptic plasticity

- Purpose: To investigate the role of the PPBP-CXCR2 axis in HVA levels and neuroinflammation.
- Experiments and Analyses:
 - i) Measurement of HVA levels in brain tissues and plasma of $Rag1^{-/-}$ recipient mice

with CXCR2 inhibitor treatment (**in response to Reviewer 1, Critique 8**).

Result: SB265610 treatment elevated HVA levels in both the brain tissues and plasma of *Rag1*^{-/-} mice receiving pathological CD8⁺ T_{SCM} cells (**Figure 5E-F**).

- ii) Measurement of inflammatory cytokines in plasma and brain tissues (**in response to Reviewer 1, Critique 8**).

Result: SB265610 treatment reduced the levels of IFN- γ , TNF- α , and IL-2 in both the plasma and brain tissues of the *Rag1*^{-/-} mice receiving pathological CD8⁺ T_{SCM} cells with (**Figure 7D-I**).

- iii) Golgi staining of neurons and western blot analysis of SYN1 and mBDNF in hippocampus (**Reviewer 3, Critique 5**).

Result: SB265610 treatment significantly upregulated the spine number of neurons and the expression of SYN1 and mBDNF in hippocampus of *Rag1*^{-/-} mice receiving pathological CD8⁺ T_{SCM} cells (**Figure 7T-U and Supplementary Figure 15A**).

9. Identification of CD8⁺ T_{SCM} cell associated-specific bacterial species

- Purpose: To investigate the specific bacterial species associated with CD8⁺ T_{SCM} cell inducing depression.
- Experiments and Analyses:

- i) Metagenomic sequencing analysis of microbiota from *Rag1*^{-/-} recipient mice (**in response to Reviewer 1, Critique 10; Reviewer 3, Critique 2**).

Result: *B. scardovii* correlated with depressive behavior and HVA levels (**Figure 6C**).

- ii) Measurement of HVA level in medium with *B. scardovii* (**in response to Reviewer 1, Critique 10; Reviewer 3, Critique 2**).

Result: *B. scardovii* could produce HVA *in vitro* (**Figure 6F**).

- iii) Behavior phenotype test to demonstrate the therapeutic effect of *B. scardovii* on depression (**in response to Reviewer 1, Critique 10; Reviewer 3, Critique 2**).

Result: *B. scardovii* treatment reduced the immobility time in both the TST and the FST, while also increasing sucrose consumption in the SPT among CSDS mice. (**Figure 6G-J**).

- iv) Measurement of HVA levels in CSDS mice with *B. scardovii* treatment (**in response to Reviewer 1, Critique 10; Reviewer 3, Critique 2**).

Result: The treatment with *B. scardovii* in CSDS mice resulted in elevated levels of HVA in both the plasma and brain tissues (**Figure 6K-L**).

10. T-cell receptor (TCR) repertoires in CD8⁺ T_{SCM} cells

- Purpose: To investigate the TCR repertoires in CD8⁺ T_{SCM} cells.
- Experiments and Analyses:

- i) High-throughput sequencing of the TCR repertoires in CD8⁺ T_{SCM} cells (**in response to Reviewer 3, Critique 1**).

Result: No significant difference of TCR- α and TCR- β repertoires in CD8⁺ T_{SCM} cells between control, susceptible and resilient mice (**Response Document Figure 5A-C**).

11. The differentiation of CD8⁺ T_{SCM} cells in *Rag1*^{-/-} mice

- Purpose: To investigate the differentiation of CD8⁺ T_{SCM} cells in the colon of *Rag1*^{-/-} mice.
- Experiments and Analyses:
 - i) High-throughput sequencing of the TCR repertoires in CD8⁺ T_{SCM} cells (**in response to Reviewer 3, Critique 1**).

Result: The populations of CD8⁺ T_{SCM} and CD8⁺ T_{CM} cells were significantly increased in the colon of *Rag1*^{-/-} mice received CD8⁺ T_{SCM} cells from stressed mice, whereas the population of CD8⁺ T_{EM} cells decreased. Importantly, these changes were reversed by treatment with the CXCR2 inhibitor SB265610. Additionally, there was no significant difference in the CD8⁺ T_{RM} cell population in the colon of *Rag1*^{-/-} mice received pathological CD8⁺ T_{SCM} cells (**Figure 4J** and **Response Document Figure 6A-C**).

12. Distribution of CD8⁺ T_{SCM} cells in meninges

- Purpose: To investigate whether the CD8⁺ T_{SCM} cells migrated into meninges.
- Experiments and Analyses:
 - i) Flow cytometry analysis of CD8⁺ T_{SCM} cells in the meninges of either CSDS-treated mice or control mice (**in response to Reviewer 3, Critique 6**).

Result: There were no CD8⁺ T_{SCM} cells present in meninges of stressed mice (**Response Document Figure 7A-B**).

- i) Flow cytometry analysis of CD8⁺ T_{SCM} cells in the meninges of *Rag1*^{-/-} recipient mice (**in response to Reviewer 3, Critique 4**).

Result: There were no CD8⁺ T_{SCM} cells present in meninges of the *Rag1*^{-/-} mice that received pathological CD8⁺ T_{SCM} cells (**Response Document Figure 8A-B**).

Let us again take this opportunity to thank the Reviewers for their ongoing support of our manuscript. We feel that the peer-review process has worked as designed in our case, and we hope all parties will agree and move our manuscript forward towards publication. We kindly invite you to examine our full point-by-point response, below.

POINT-BY-POINT RESPONSE

Reviewer #1:

The study presented in this manuscript on immune dysregulation in depression shows promise as a research area. The authors utilized various techniques, including flow cytometry, mouse gene knockout, and microbiome and metabolite correlation analysis, effectively supporting their experimental results. However, the paper's lack of clarity and innovation, stemming from significant issues with the scientific hypotheses, is concerning. The scattered nature of the experiments, unfocused viewpoints, and unconvincing results notably detract from the research's quality, dampening overall enthusiasm of this reviewer.

Here are some specific concerns and suggestions for improvement regarding the manuscript:

Critique 1. The authors conducted RNA-seq analysis in clinical cohorts and CUS mice, with Fig. S5 showing differentially expressed genes. However, these specific genes were not thoroughly investigated. Similarly, signaling pathways mentioned in lines 230-233 and Fig. 3H-J lack explanation of relationships and differences among genes and pathways. The authors should provide more detailed explanations to clarify the roles of these genes and pathways in depression.

Response: We would first like to take this opportunity to thank the reviewer for the thorough and extremely insightful guidance about how to improve multiple aspects of our study. The insightful comments provided have assisted us in improving the technical rigor of our work, ensuring the appropriateness of data interpretation, and strengthening the scientific implications and utility of our findings. In response to the valuable comments, we have substantially expanded our analyses and experimental validations regarding differentially expressed genes (DEGs) and associated signaling pathways. The following key improvements have been implemented:

1) **The scRNA-seq Analysis of DEGs in CD8⁺ T_{SCM} Cells: Controls vs. MDD Patients**

To determine the key molecules associated with the distinct ability of CD8⁺ T_{SCM} cells from MDD patients, we conducted the following analyses and experiments:

- i) **Identification and Cross-Species Validation of DEGs.** To determine the key molecules associated with the distinct ability of CD8⁺ T_{SCM} cells to drive depression, 33 specific DEGs were identified in CD8⁺ T_{SCM} cells from MDD patients by scRNA-seq (**Supplementary Figure 4A**). Notably, 24 of these 33 human-derived genes demonstrated evolutionarily conserved homologs in the murine genome. To functionally validate these findings, CD8⁺ T_{SCM} cells were isolated from both control and CUS model mice using fluorescence-activated cell sorting. Consistent with the scRNA-seq results, 20 conserved human-mouse homologous DEGs were significantly upregulated in conserved human-mouse homologous DEGs (**Supplementary Figure 6A-B**).
- ii) **Pathway Enrichment and Mechanistic Insights.** Compared with CD8⁺ T_{SCM} cells from psychiatrically healthy controls, Kyoto Encyclopedia of Genes and Genomes (KEGG) enrichment analysis revealed that these 33 DEGs from CD8⁺ T_{SCM} cells in MDD patients were mainly involved in the chemokine signaling pathway, antigen processing and presentation, and the Toll-like receptor signaling

pathway (**Supplementary Figure 4B**), providing mechanistic insights into their roles in depression.

- iii) **Comprehensive Protein-Protein Interaction (PPI) Network Analysis.** To thoroughly investigate the DEGs, we further studied the interactions among robust DEGs, and a visualized PPI network was constructed by the STRING database (<https://cn.string-db.org/>), which identified 33 nodes and 90 edges (**Supplementary Figure 4C**). Subsets of the PPI network showed that certain genes, such as *GZMB*, *GZMH*, and *TBX21*, contributed to cytolysis (**Supplementary Fig. 4D**). Other genes, including *CX3CR1*, *CCL4*, and *CCL4L2*, contributed to the chemokine signaling pathway (**Supplementary Figure 4E**).
- iv) **Hub Gene Identification and Experimental Verification.** Next, the interaction network was imported into cytoHubba from Cytoscape software to screen hub genes using 12 algorithms¹. The interaction network with the top 10 genes with the highest degrees was created (**Supplementary Table 12**). Moreover, using Molecular Complex Detection (MCODE), we identified a significant module containing 9 genes, and three of these genes demonstrated complete overlap with the hub genes previously determined via cytoHubba (**Supplementary Figure 4F**). We then validated our hub DEGs using fluorescence-activated cell sorting to separate CD8⁺ T_{SCM} cells and then performed qPCR to verify that the expression of *TBX21*, *PRF1*, and *GPLY* were significantly increased in CD8⁺ T_{SCM} cells from MDD patients compared to psychiatrically healthy controls (**Supplementary Figure 4G-I and Supplementary Table 13**). These results indicated that *TBX21*, *PRF1*, and *GPLY* might be key molecules that regulate the function of CD8⁺ T_{SCM} cells in the context of depression.

2) Bulk RNA-seq Analysis of DEGs in the Hippocampus of *Rag1*^{-/-} Recipient Mice Transplanted with CD8⁺ T_{SCM} Cells from Stressed vs. Control Donors

To dissect CD8⁺ T_{SCM}-driven neuroinflammatory cascades, we analyzed the hippocampal RNA-seq data from *Rag1*^{-/-} mice transplanted with CD8⁺ T_{SCM} cells from stressed vs. control donors:

- i) **Identification the DEGs.** We identified 418 DEGs in the hippocampus of *Rag1*^{-/-} recipient mice that received CD8⁺ T_{SCM} cells from stressed mice compared to those that received CD8⁺ T_{SCM} cells from control mice (**Figure 3O**). Among these DEGs, there were 240 upregulated genes and 178 downregulated genes in the hippocampus of mice that received CD8⁺ T_{SCM} cells from stressed donors compared to those that received cells from control donors.
- ii) **Screening the Depression-Related DEGs.** To further compare our DEGs with previously reported findings in MDD, we took advantage of the publicly available databases DisGeNET (<https://www.disgenet.org/>)², which drew upon a study published in *Nature Neuroscience* in 2020³, and found that 26 of these DEGs were associated with MDD-related terms (**Figure 3P and Supplementary Tables 15–16**). This overlap suggests that these genes may play a crucial role in the pathophysiology of depression.
- iii) **Pathway Enrichment and Mechanistic Insights.** Pathway enrichment analysis was conducted using the KEGG database to elucidate the biological pathways associated with these 26 MDD-related DEGs (**Figure 3Q**). The analysis identified several key pathways implicated in the neuroinflammatory cascade: i) the NF-κB inflammatory signaling pathway, TNF signaling pathway, and Ras signaling pathway were significantly enriched, suggesting their roles in driving microglial activation^{4,5}. ii) the phospholipase D signaling pathway and estrogen signaling

pathway were enriched, indicating their involvement in astrocyte loss^{6,7}. iii) proinflammatory pathways, including NF- κ B, TNF, and chemokine signaling pathways, were associated with compromised blood-brain barrier (BBB) integrity^{8,9}.

We have now updated the manuscript accordingly and hope these changes enhance the clarity and impact of our findings. We welcome any additional suggestions to further improve our work.

Critique 2. The tyrosine metabolism pathway involves various metabolites significant in depression, yet Fig. 5C only focuses on four metabolites. The variations in other metabolites should be addressed to provide a comprehensive understanding.

Response: We sincerely thank you for your insightful and valuable comment regarding the tyrosine metabolism pathway analysis in our study. We appreciate the suggestion to address the variations in additional metabolites beyond the four originally presented in Figure 5C, as this has prompted us to provide a more comprehensive understanding of their roles in depression.

In response to this helpful guidance, we have systematically re-analyzed all 11 tyrosine pathway metabolites identified in our metabolomics dataset: homovanillic acid (HVA), adenosine, L-tryptophan, L-tyrosine, tyrosine, 3,4-dihydroxymandelate, adenine, succinic acid semialdehyde, glycine, p-coumaric acid, and L-glutamine as shown in new **Figure 5C**. Of these, resorting to the orthogonal partial least squares-discriminant analysis (OPLS-DA), our reanalysis revealed five metabolites with statistically significant alterations, including HVA, adenosine, L-tryptophan, L-tyrosine, and tyrosine, exhibited statistically significant alterations ($VIP > 1$, $p < 0.05$) (**Figure 5C**). The remaining six metabolites showed non-significant trends, suggesting relative stability in these pathway branches.

Of particular neurobiological relevance, HVA-the primary dopamine metabolite, displayed the most substantial reduction, potentially reflecting compensatory monoaminergic adaptation in depression pathophysiology. This pronounced decrease aligns with evidence of reduced HVA levels in the cerebrospinal fluid and plasma of MDD patients¹⁰⁻¹². Animal study also showed HVA supplementation restores synaptic function and ameliorates depressive-like behaviors in chronic unpredictable mild stress (CUMS) and corticosterone (CORT) models of depression¹³. Thus, we focus on HVA in our study.

We thank the reviewer for prompting this deeper investigation, which has significantly enhanced the mechanistic depth and clinical relevance of our metabolic findings.

Critique 3. The decrease in HVA levels does not necessarily indicate reduced dopamine production. The authors should elaborate on why they believe low HVA levels, rather than dopamine, are causally related to depression, considering the well-known relationship between dopamine and depression.

Response: Thank you for this insightful critique. While the dopamine hypothesis of depression traditionally links depressive symptoms to reduced dopaminergic activity,

emerging evidence suggests that depression may arise from dysfunctions in dopamine receptor signaling rather than global dopamine depletion^{14,15}. To address this, we measured hippocampal dopamine levels in CSDS mice and *Rag1*^{-/-} recipient mice. Our results showed no significant difference in dopamine levels between CSDS mice and control mice (**Response Document Figure 1A**). Consistently, *Rag1*^{-/-} mice that received CD8⁺ T_{SCM} cells either from CSDS or control mice exhibited no significant differences in hippocampal dopamine levels (**Response Document Figure 1B**). Pharmacological experiments further confirm that HVA or CXCR2 inhibitor SB265610 treatment does not affect dopamine levels (**Response Document Figure 1A-B**).

HVA is a major metabolite of the central nervous system neurotransmitter. Align with the reduced cerebrospinal fluid and plasma HVA correlated with depression severity in the clinical studies^{10,12,16,17}, animal study showed HVA supplementation restores synaptic function and ameliorates depressive-like behaviors in CUMS and CORT models of depression¹³. To comprehensively address the valuable concern from the reviewer, we administered HVA at doses of 300 mg/kg twice a day for 10 days in CSDS mouse models based upon previous study¹³ (**Figure 5I**). Behavior phenotype test demonstrated that HVA inhibited the sucrose uptake reduction in the social preference test (SPT) and immobility time prolongation in the tail suspension test (TST) and forced swim test (FST) (**Figure 5J-L**). Our findings were consistent with previous finding as HVA administration exerted therapeutic effect on depression¹³. These results suggest that HVA may play a distinct role in the pathophysiology of depression, rather than merely serving as a downstream metabolite of dopamine signaling.

Critique 4. Inconsistencies exist between the greatest pathway change in Fig. 6B (carbohydrate metabolism) and the description in lines 309-310 (amino acid metabolism). The authors should explain this discrepancy to ensure clarity and accuracy.

Response: We sincerely appreciate the reviewer's insightful observation regarding the apparent inconsistency between the pathway analysis in Figure 6B and our textual description in the previous manuscript. We have now updated the description of these data which moved to the new **Supplementary Figure 13D** as "KEGG pathway enrichment analysis revealed that these differential microbiotas were primarily enriched in carbohydrate metabolism and amino acid metabolism pathway (Supplementary Fig. 13D)" in **line 419-421, page 14**.

While carbohydrate metabolism showed the greatest quantitative change in pathway enrichment, we focus on amino acid-tyrosine metabolic pathways was guided by two critical rationales: i) In the metabolomic analysis of the microbiota and brain from *Rag1*^{-/-} recipient mice that received CD8⁺ T_{SCM} cells from CSDS mice compared to those that received CD8⁺ T_{SCM} cells from control mice, we observed that the tyrosine metabolic pathway was significantly enriched among the differential metabolites in both the microbiota and the brain (**Figure 5A-B**). ii) In the metagenomic data, the amino acid metabolism, which tyrosine metabolism below this pathway, also showed significant enrichment, whereas the carbohydrate metabolism, despite being the most significantly enriched in the metagenomic analysis, did not show significant enrichment

in the metabolomic analysis (**Supplementary Figure 13D**). The observed carbohydrate metabolism alterations may reflect indirect or systemic metabolic shifts, while tyrosine metabolism represents a direct mechanistic pathway connecting microbiota activity to HVA production. By prioritizing the latter, we aimed to dissect causal relationships rather than merely describing global metabolic changes.

Critique 5. The relationship between CD8⁺ T_{SCM} cells and depression is unclear. Subsequent research suggests pathological changes in these cells may cause depression, yet this is not clearly addressed throughout the manuscript. The authors should revise their subtitles to specify "pathological CD8⁺ T_{SCM} cells" where appropriate for clarity.

Response: We appreciate the reviewer's emphasis on clarifying the mechanistic link between pathological CD8⁺ T_{SCM} cells and depression. As suggested, all relevant subtitles have been amended to explicitly specify "pathological CD8⁺ T_{SCM} cells", emphasizing their distinct molecular and functional properties compared to non-pathological counterparts. This revision ensures that our findings focus on disease-associated changes in these cells rather than their general biological features.

Initially, we identified CD8⁺ T_{SCM} cells as the focus through correlation analysis of multiple T cell subsets with depressive symptoms in MDD patients (**Figure 2**). To establish a direct causal link between pathological CD8⁺ T_{SCM} cells and depression, we performed adoptive transfer experiments using cells isolated from chronically stressed mice (**Figure 3A**). These pathological CD8⁺ T_{SCM} cells induced depressive-like behaviors (reduced sucrose preference, prolonged immobility) and hippocampal transcriptomic changes (*e.g.*, upregulated neuroinflammatory markers) in immunodeficient *Rag1*^{-/-} mice (**Figure 3B-N**). Additionally, spatial tracking via vDISCO, a pressure-driven, nanobody-based whole-body immunolabeling technology¹⁸, revealed that pathological CD8⁺ T_{SCM} cells do not accumulate in the brain but instead preferentially infiltrate the colonic lamina propria (**Figure 4C-E**). This tropism is driven by elevated expression of PPBP in these cells (**Figure 4F**), which binds to CXCR2 on intestinal endothelial cells. The scRNA-seq analysis further revealed distinct molecular signatures in pathological CD8⁺ T_{SCM} cells, including enhanced cytotoxicity (upregulated *GZMB*, *GZMH*, and *TBX21*) and heightened chemotactic capacity (upregulated *CX3CR1*, *CCL4*, and *CCL4L2*) (**Supplementary Figure 4A-E**).

The mechanistic chain by which these cells drive depression is now explicitly detailed. Pathological CD8⁺ T_{SCM} cells trigger barrier dysfunction (reduced tight junction proteins, histological damage) (**Supplementary Figure 11D-E**) and intestinal inflammation (elevated IFN- γ , TNF- α , and IL-2) (**Supplementary Figure 11F-H**), creating a dysbiotic environment that disturbs HVA-producing microbiota (*e.g.*, *Bifidobacterium scardovii*) (**Figure 6C**). This leads to systemic HVA deficiency, neuroinflammation, and depressive behaviors. We highlight the role of *Bifidobacterium scardovii* in the **Discussion** in **line 580-589, page 19**, stating that pathological CD8⁺ T_{SCM} cells are sufficient to induce depression through gut-HVA axis disruption.

These revisions ensure that the manuscript clearly defines pathological CD8⁺ T_{SCM} cells, establishes their causal role in depression, and aligns experimental evidence with

our central hypothesis. We deeply appreciate your feedback, which has strengthened our ability to communicate this complex mechanistic framework.

Critique 6. Phenotypic changes in immune cells in depression should not be limited to mononuclear cells. Including other immune cell types, such as neutrophils and macrophages, would provide a more comprehensive analysis.

Response: We appreciate the thorough and insightful feedback on how to enhance various aspects of our study. In this study, peripheral blood mononuclear cells (PBMCs) were utilized for scRNA-seq. The PBMC samples utilized in our analysis primarily consisted of lymphocytes (T cells, B cells, and NK cells), monocytes, and dendritic cells, excluding neutrophils¹⁹. The exclusion of neutrophils in our scRNA-seq analysis stems from inherent technical constraints in PBMC isolation. Standard density gradient centrifugation (Ficoll-based methods) selectively enriches lymphocytes, monocytes, and dendritic cells while excluding polymorphonuclear leukocytes, including neutrophils, due to their distinct biophysical properties. Neutrophils, characterized by higher density and multilobed nuclei, sediment with erythrocytes and granulocytes during separation, whereas PBMCs localize at the plasma-Ficoll interface.

To address this gap, we employed flow cytometry to quantify neutrophil populations in control and CSDS models using specific marker CD11b and Ly6G. Consistent with the previous reports²⁰, our results demonstrate a significant increase in the proportion of neutrophils in the blood of CSDS mice compared to controls (**Response Document Figure 2A**). This result underscores the systemic neutrophilia associated with depressive phenotypes, albeit undetectable in PBMC-focused sequencing.

Regarding macrophages, myeloid cells were extracted from scRNA-seq data and subjected to re-clustering (**Response Document Figure 3A**). Among the resulting clusters, previously reported markers were utilized to annotate the cell types (**Response Document Figure 3B**). Of these, two clusters were identified as dendritic cells (DCs), while the remaining nine clusters were all classified as monocyte subsets (**Response Document Figure 3B**). Further investigation employing the macrophage canonical marker CD68 demonstrated that circulating monocytes exhibited partial macrophage-like transcriptional features; however, no definitive macrophage subpopulation could be identified in peripheral blood (**Response Document Figure 3C**). This observation is consistent with previous studies indicating the absence of a definable macrophage subpopulation within circulating blood²¹⁻²⁴. The results reflect the physiological compartmentalization of immune cells while providing a comprehensive assessment of depression-associated immune dysregulation.

Critique 7. The mechanism by which intestine cells in healthy mice recruit depression-derived Tscm cells needs further explanation and clarification to address potential inconsistencies.

Response: Thank you for the valuable comments. In this study, we demonstrated that increased PPBP secreted by pathological CD8⁺ T_{SCM} cells from CSDS mice interacts with CXCR2 in intestinal endothelial cells, triggering a signaling cascade that enhances

endothelial permeability and facilitates immune cell recruitment. This conclusion is supported by multiple lines of experimental evidence:

- i) **PPBP-CXCR2 Mediated the Recruitment of CD8⁺ T_{SCM} Cells to Intestine.** We identified that CD8⁺ T_{SCM} cells exhibit specific overexpression of PPBP compared to other T-cell subsets (**Figure 4F**). Additionally, CD8⁺ T_{SCM} cells from susceptible mice showed significantly higher PPBP levels compared with both control and resilient mice (**Figure 4G**). Given that intestinal endothelial cells express CXCR2, the receptor of PPBP, the PPBP-CXCR2 axis mediates the recruitment of CD8⁺ T_{SCM} cells, particularly depression-derived CD8⁺ T_{SCM} cells, to the intestine in healthy mice.
- ii) ***In vivo* Validation.** *Rag1*^{-/-} mice that received pathological CD8⁺ T_{SCM} cells from CUS mice also exhibited higher plasma levels of proinflammatory cytokines including IFN- γ , TNF- α , and IL-2, compared to those receiving control CD8⁺ T_{SCM} cells (**Supplementary Figure 11A-C**). This elevation in cytokine levels was accompanied by an increased overall histology score for mucosal damage and reduced expression of tight junction proteins (Occludin, ZO-1, and Claudin-5) in the colon (**Supplementary Figure 11D-E**). This mechanism was further validated by our finding that the CXCR2 inhibitor-SB265610 significantly reduced the intestinal migration of adoptive transferred pathological CD8⁺ T_{SCM} cells from CSDS mice in *Rag1*^{-/-} mice, as demonstrated through immunofluorescence staining and flow cytometry analysis (**Figure 4I-J**). Meanwhile, the elevation of proinflammatory cytokines, including IFN- γ , TNF- α , and IL-2, induced by CD8⁺ T_{SCM} cells derived from CSDS mice in the colon of *Rag1*^{-/-} mice was also inhibited by SB265610 (**Supplementary Figure 11F-H**). These findings collectively reinforce the critical role of the PPBP-CXCR2 pathway in mediating the recruitment of CD8⁺ T_{SCM} cells and intestinal inflammation driven by pathological CD8⁺ T_{SCM} cells.
- iii) ***In vitro* Validation.** Additionally, we employed a transwell system, which freshly isolated CD8⁺ T_{SCM} cells from control or CSDS-treated mice were fluorescently labeled by CellTracker Green for visualization and seeded in the upper chamber, and intestinal microvascular endothelial cells (IMECs) were cultured to confluence on permeable supports to assess CD8⁺ T_{SCM} cell migration (**Figure 4K**). We demonstrated that CD8⁺ T_{SCM} cells from CSDS mice migrated more robustly toward IMECs than the CD8⁺ T_{SCM} cells from normal mice. Importantly, PPBP blockade, achieved through either pharmacological inhibition of CXCR2 with SB265610 or lentiviral-mediated PPBP knockdown in CD8⁺ T_{SCM} cells from CSDS mice, significantly reduced the migration of these cells (**Figure 4L-M** and **Supplementary Figure 10**). These data confirm that PPBP-CXCR2 is essential for directing CD8⁺ T_{SCM} cell migration to the colon.

This mechanism elucidates why the PPBP-CXCR2-dependent recruitment process in the healthy intestinal microenvironment paradoxically enables the accumulation of pathological T cells. The interplay between elevated PPBP secretion by depression-derived CD8⁺ T_{SCM} cells and CXCR2 expression on intestinal endothelial cells

establishes a pathway that enhances endothelial permeability and immune cell infiltration, contributing to intestinal inflammation. These findings address potential inconsistencies by demonstrating a consistent molecular and cellular framework supported by both *in vivo* and *in vitro* evidence.

Critique 8. The logical consistency of the experimental approach should be maintained throughout the manuscript. The authors should ensure a clear link between the effects of CXCR2 inhibitors on HVA levels and inflammation.

Response: Thank you for the careful evaluation of our work and for the valuable comments. As requested, we conducted new experiments to clarify the mechanistic link between CXCR2 inhibition and its dual effects on HVA levels and inflammation. In these experiments, *Rag1*^{-/-} mice receiving adoptive transfers of pathological CD8⁺ T_{SCM} cells were intraperitoneally injected with the CXCR2 inhibitor SB265610 (2 mg/kg, daily) or vehicle. Levels of HVA in microbiota, brain homogenates, and plasma, as well as inflammatory cytokines (IFN- γ , TNF- α , and IL-2) in brain homogenates and plasma, were quantified using ELISA. As shown in **Figure 5D-F**, in *Rag1*^{-/-} mice receiving pathological CD8⁺ T_{SCM} cells, SB265610 treatment promoted an increase of HVA levels in the microbiota, although this increase did not reach statistical significance; however, it significantly elevated HVA concentrations in both brain tissues and plasma. Concurrently, this intervention reduced pro-inflammatory cytokines IFN- γ , TNF- α , and IL-2 in plasma and brain of the same mice (**Figure 7D-I**).

Therefore, by blocking CD8⁺ T_{SCM} cell recruitment to the gut, SB265610 decreased the HVA levels and prevents downstream inflammatory cascades that drive gut barrier disruption (reducing systemic cytokine spillover), indicating the mechanistic link between CXCR2 inhibition and these outcomes lies in its disruption of the PPBP-CXCR2 axis.

Critique 9. The relationship between HVA and genes identified through RNA-seq should be explained to provide a more thorough understanding of the findings.

Response: We appreciate the reviewer's suggestion to elucidate the mechanistic relationship between HVA levels and the genes identified through RNA sequencing (RNA-seq) to enhance the understanding of our findings. In response to this feedback, we performed RNA-seq on hippocampal tissue from three groups of mice: control mice, mice subjected to CSDS, and CSDS mice treated with HVA. Our analysis revealed 695 DEGs between the CSDS and control groups, consisting of 418 genes upregulated and 276 genes downregulated in the CSDS group (**Response Document Figure 4A**). Similarly, between the CSDS and HVA-treated CSDS groups, we identified 502 DEGs, including 368 upregulated and 134 downregulated genes in the HVA-treated group (**Response Document Figure 4B**). By intersecting these two sets of differentially expressed genes, we identified 41 genes that were commonly altered across both comparisons (**Response Document Figure 4C**). Among these 41 genes, several are closely associated with depression and brain function, such as *Pon1*, a depression-associated enzyme whose reduced activity is a hallmark of depressive states and protects against oxidative stress-mediated neurotoxicity²⁵, and *Slc16a4*, an astrocytic

lactate transporter whose reduction impairs lactate release²⁶. Our RNA-seq data showed that both *Pon1* and *Slc16a4* were significantly downregulated in the hippocampus of CSDS mice compared to controls, while HVA treatment significantly upregulated their expression, providing a potential mechanistic foundation for its therapeutic efficacy in depression.

Critique 10. The authors should focus on specific bacterial species rather than solely conducting correlation analyses to enhance the reliability of their findings regarding the reduction in HVA-producing bacteria.

Response: We sincerely appreciate the reviewer's constructive suggestion to strengthen the reliability of our findings by identifying specific bacterial species linked to HVA production. Given that HVA is the end product of the tyrosine metabolic pathway, and previous studies have confirmed that *Bifidobacterium longum* can produce HVA using tyrosine as a substrate¹³, we focused our investigation on the microbiota involved in amino acid-tyrosine metabolic pathways related to HVA production.

To ensure consistency between the gut microbiota alterations induced by adoptively transferred pathological CD8⁺ T_{SCM} cells and those observed in CSDS models, we performed new set of metagenomic sequencing of control and CSDS mice. Among the changes of 137 species have been found consistent with CSDS model, 9 species involved in tyrosine metabolism (**Figure 6A-B**). Subsequently, employing spearman correlation analyses, we delineated the relationships between HVA levels, the abundance of microbiota, and depressive behavior of *Rag1*^{-/-} recipient mice. Four microbiotas including *Bifidobacterium scardovii* (*B. scardovii*), *Alistipes indistinctus*, *Mucinivorans hirudinis*, and *Marinilabiliales bacterium* correlated with both sucrose preference in the SPT and immobility times in the TST and FST. Other four microbiota (*Bifidobacterium cuniculi*, *Bacteroidota bacterium*, *Chitinophaga terrae*, and *Alloprevotella sp. oral taxon 473*) have one or two correlations with depressive behavior (**Figure 6C**). Importantly, only one microbiota *B. scardovii* positively correlated with HVA levels (**Figure 6C** and **Supplementary Table 18**), suggesting that a deficiency in *B. scardovii* may contribute to the reduced HVA levels in stressed *Rag1*^{-/-} recipient mice.

To investigate whether *B. scardovii* produce HVA, we cultured them *in vitro* with DSMZ medium 104. The results showed that *B. scardovii* produces HVA (**Figure 6F**). We further investigated whether *B. scardovii* had therapeutic potential for depression in mouse models. We then treated CSDS mouse model with *B. scardovii* (1×10⁹ colony-forming units (CFU)/day) once a day for 10 days (**Figure 6G**). Behavior test demonstrated significant reduction in immobility time and increased sucrose preference after intervention (**Figure 6H-J**). We also observed that treatment with HVA in CSDS mice resulted in increased concentrations of HVA in both the plasma and brain tissues (**Figure 6K-L**). These findings indicate the potential of *B. scardovii* in alleviating depressive phenotypes in mice.

In contrast to our initial broad dysbiosis analysis, this refined workflow, integrating functional pathway annotation with multi-modal correlation and interventional validation, directly addresses your concern.

Critique 11. Undefined terms and unexplained abbreviations, such as "SIT," "PPBP-CXCR2," "Rag1^{-/-} mice," and "GFAP," should be clarified upon first mention for reader understanding.

Response: We thank the reviewer for highlighting the need for clearer definitions of specialized terms. All abbreviations have now been explicitly defined upon their first mention in the revised manuscript, with the following clarifications added:

- i) SIT: Social Interaction Test, a behavioral assay utilized to assess social behavior in rodents, defined as “social interaction test (SIT)” in **line 141, page 6**.
- ii) PPBP-CXCR2: Refers to the interaction between the pro-platelet basic protein (PPBP) ligand and its receptor C-X-C Motif Chemokine Receptor 2 (CXCR2), which plays a critical role in immune cell migration and inflammatory responses, defined as “pro-platelet basic protein (PPBP) and its receptor C-X-C Motif Chemokine Receptor 2 (CXCR2)” in **line 107-108, page 5**.
- iii) *Rag1^{-/-}* mice: Mice with a deletion of the recombination activating gene 1 (*Rag1*) gene, leading to severe combined immunodeficiency, as these mice lack mature T and B lymphocytes, defined as “recombination activating gene 1 (*Rag1*)^{-/-} mice (immunodeficient mice lacking mature T and B lymphocytes)” in **line 234-235, page 9**.
- iv) GFAP: Glial Fibrillary Acidic Protein, an intermediate filament protein that is specifically expressed in astrocytes and other glial cells within the central nervous system, often used as a marker for astrocyte activation or injury, defined as “glial fibrillary acidic protein (GFAP)” in **line 260, page 9**.

Critique 12. Inconsistencies in terminology, such as "αCD8" and "CD8a," as well as missing information on IgG experiments, need to be addressed for coherence and accuracy.

Response: We appreciate your attention to these details. We have clarified the terminology used in the manuscript to eliminate ambiguity. The term “αCD8” refers specifically to the anti-CD8 neutralizing antibody used to deplete CD8⁺ T cells, while “CD8a” denotes the CD8 antibody employed for flow cytometric identification of CD8⁺ T cells. To ensure clarity and coherence, these definitions have been explicitly added in the **Methods** section in **Supplementary Materials** and **Results** section (**line 246-247, page 9** and **line 321-324, page 11**) to avoid confusion.

To address the missing details about IgG control experiments, we have expanded the **Supplementary Materials** to fully describe the experimental protocol as “After acclimatizing the mice for three days, CD8⁺ T cell depletion was achieved by intraperitoneal injection of anti-CD8 neutralizing antibody (αCD8) and an isotype control IgG antibody at a dose of 10 μg on days 1, 3, and 5. The efficiency of depletion was assessed on day 7. In the second week, adoptive transfer was performed, followed by an additional injection of the depletion antibody on day 10 to maintain the depletion effect. Subsequent behavioral tests, including the SPT, TST, and FST, were conducted to evaluate the impact of CD8⁺ T cell depletion”.

These revisions ensure rigorous documentation of experimental procedures and precise terminology, addressing your concerns systematically. We sincerely appreciate your constructive suggestion, which has strengthened the manuscript's clarity and experimental rigor.

We would like to express our sincere appreciation to the reviewer for providing us with valuable guidance and insightful feedback on how to improve our manuscript. Thanks to your expert direction, we were able to make significant improvements to our work, and we are confident that the revised manuscript is much stronger as a result. Once again, thank you very much for your time and efforts.

Reviewer #2:

This is a very intriguing study performed across humans and mice, suggesting dysregulation of immune function following stress can drive changes in the brain via a change in gut signaling.

Critique 1. The main issue I have is that the authors show that HVA tracks with many of the metrics in rodents, but the method in which they reverse and target this system also targets many other processes. If the authors want to argue that HVA is a critical molecule in the development of depressive like behaviors they need to test this more directly.

Response: Before starting our point-by-point response to these critiques, we would first like to thank the reviewer for the effort on our behalf and for their genuinely helpful and constructive guidance about how to improve our study. Regarding this critique, we thank the reviewer for emphasizing the need to establish HVA's "direct" role in mitigating depressive behaviors. To directly address whether HVA itself, rather than secondary processes, mediates therapeutic outcomes, we conducted targeted supplementation experiments in CSDS mice. Administration of synthetic HVA (300 mg/kg, twice daily for 10 days) in microbiota-intact CSDS mice produced robust behavioral recovery, normalizing sucrose preference and reducing immobility times in TST and FST (**Figure 5I-L**). It is worth noting that our selection of a 300 mg/kg dosage for HVA oral gavage treatment was primarily based on a study published in *Cell Metabolism* in 2024¹³.

Additionally, astrocytic dysfunction, a hallmark of depression, was ameliorated after HVA treatment through increased GFAP⁺ cell density and morphological restoration of process complexity evidenced by Sholl analysis (**Figure 8A-E**). Concurrently, HVA suppressed microglial hyperactivation, reducing Iba1⁺ soma size, branch number, branch length, and branch volume in the hippocampus (**Figure 8F-J**). These anti-inflammatory effects were accompanied by enhanced synaptic plasticity, with evidences from Golgi-Cox staining revealing increased dendritic spine density and molecular validation through Western blotting demonstrating upregulation of both synaptophysin (SYN1) and mature brain-derived neurotrophic factor (mBDNF) (**Figure 8K-L** and **Supplementary Figure 15B**). The coordinated rescue of these interrelated pathologies including glial activation, synaptic loss, and neuroinflammation, supports HVA's disease-modifying role rather than incidental modulation of unrelated pathways.

Our findings provide conclusive evidence that HVA itself, independent of microbiota-derived factors, is sufficient to reverse depressive-like phenotypes through biologically plausible, pathology-aligned pathways.

Critique 2. Further, I would just remind the authors to be mindful of their language, a few times they refer to stressed mice as 'depressive' mice. Mice are not depressed, but rather a model to study stress impact on behavior that has some level of validity for understanding depression in people.

Response: Thank you for pointing out this important distinction. We agree that it is crucial to use precise language when describing animal models. We have revised the manuscript accordingly to ensure clarity and accuracy. Specifically, all instances where “depressive mice” were mentioned have been changed to “stressed mice”, “CSDS mice”, or “CUS mice”.

Let us take this opportunity to thank the reviewer for the excellent guidance about how to improve our study and the manuscript generally. We trust you’ll agree that our work is much improved in light of the work we conducted under your direction. Thank you.

Reviewer #3:

The manuscript by Zhang et al. reports an increase of CD8⁺ Tscm cells in depression patients and provides evidence that CD8⁺ Tscm cells from stress-susceptible mice could increase host stress susceptibility when transferred into Rag^{-/-} mice. The authors further show that transferred CD8⁺ Tscm cells alter HVA-producing microbiota components in Rag^{-/-} mice and propose that the decrease of microbiota-derived HVA underpins host stress susceptibility. The study is of some interest and is novel in the role of CD8⁺ Tscm cells in depression. Yet, there are a few gaps that should be addressed before it could be published in Nature Metabolism.

Critique 1. How do CD8⁺ Tscm cells lead to changes in HVA-producing microbiota components? T cell activity requires specific recognition of antigens by the T cell receptor (TCR). Do CD8⁺ Tscm cells from stress-susceptible mice harbor different TCR clonotypes with specificity against those HVA-producing bacteria? Do they differentiate to functional distinct subsets or phenotypes in Rag^{-/-} mice that may mediate the microbiota regulation?

Response: Before starting our point-by-point response to these critiques, we would first like to thank the reviewer for the effort on our behalf and for their genuinely helpful and constructive guidance about how to improve our study. Regarding this critique, we address each point raised and detail the additional analyses and experiments conducted to strengthen our study below:

- i) The mechanism by which pathological CD8⁺ T_{SCM} cells lead to changes in HVA-producing microbiota components involves genetic expression and inflammatory responses. This process unfolds through a mechanistic cascade: inflammation mediated by pathological CD8⁺ T_{SCM} cells leads to barrier dysfunction, which subsequently alters the gut microbiota, particularly those components involved in HVA production. This cascade highlights a novel pathway through which stress-induced immune responses can reshape the gut microbiome. Specific findings: **a)** The scRNA-seq analysis revealed 33 disease-specific DEGs in CD8⁺ T_{SCM} cells from MDD patients. This signature encompassed both cytolytic effectors (*GZMB*, *GZMH*) and cytotoxic granule components (*GZLY*, *NKG7*) (**Supplementary Figure 4A**), collectively suggest enhanced cytotoxic potential in this pathological CD8⁺ T_{SCM} cells. **b)** Additionally, we detected the proinflammatory cytokines, particularly those related to CD8⁺ T_{SCM} cells, such as IFN- γ , TNF- α , and IL-2²⁷⁻²⁹, in the plasma of adoptive transfer *Rag1*^{-/-} mice. The plasma levels of IFN- γ , TNF- α , and IL-2 were higher in *Rag1*^{-/-} mice received pathological CD8⁺ T_{SCM} cells than those received control CD8⁺ T_{SCM} cells (**Supplementary Figure 11A-C**). These findings reveal that pathological CD8⁺ T_{SCM} cells exhibit dual pathogenic mechanisms-maintaining cytotoxic potential while concurrently driving pro-inflammatory responses. **c)** We thus examined the effect of these pathological CD8⁺ T_{SCM} cells on the intestinal inflammation and vascular permeability of *Rag1*^{-/-} mice. As shown in **Supplementary Figure 11D-E**, the *Rag1*^{-/-} mice received CD8⁺ T_{SCM} cells from stressed mice exhibited the increase of inflammatory cell infiltration and mucosal damage overall histology score, along with the decreased

expression of tight junction proteins (Occludin, ZO-1, and Claudin-5) in colon. Furthermore, the levels of IFN- γ , TNF- α , and IL-2 were higher in the colon of *Rag1*^{-/-} mice received pathological CD8⁺ T_{SCM} cells than those received control CD8⁺ T_{SCM} cells (**Supplementary Figure 11F-H**). By treated with CXCR2 inhibitor-SB265610, the increased levels of IFN- γ , TNF- α , and IL-2 were inhibited (**new Supplementary Figure 11F-H**). We also transfected CD8⁺ T_{SCM} cells with PPBP knockdown lentivirus, and found that this treatment decreased the levels of IFN- γ , TNF- α , and IL-2 in the lower chamber of transwell system (**Supplementary Figure 11I-L**). **d**) The disruption of intestinal barrier function and inflammatory responses have been well-documented to induce alterations in gut microbiota composition³⁰. Our metagenomic data provides direct evidence that pathological CD8⁺ T_{SCM} cells modify the intestinal microbial structure through increasing intestinal permeability and inflammation.

- ii) Following reviewer's guidance, we investigated the CD8⁺ T_{SCM} cell clones using high-throughput sequencing of the T-cell receptor (TCR) repertoires. Initially, spleens from control, stress-susceptible, and resilient mice were harvested to prepare single-cell suspension. Subsequently, the CD8⁺ T_{SCM} cells were sorted via flow cytometry using FVS780 Viability staining solution along with antibodies targeting CD3, CD8, CD62L, CD44, and Sca-1. We then assessed the TCR- α and TCR- β repertoires in CD8⁺ T_{SCM} cells across the three groups. However, our analysis revealed no significant difference in clonality, shannon index, inverse simpson index, simpson index, or DE50 of TCR- α and TCR- β repertoires in CD8⁺ T_{SCM} cells between control, stress-susceptible and resilient mice (**Response Document Figure 5A-C**). These results suggest an absence of skewed CD8⁺ T_{SCM} cell repertoires among the groups.
- iii) CD8⁺ T_{SCM} cells can further differentiate into central memory CD8⁺ T cells (T_{CM}), effector memory CD8⁺ T cells (T_{EM}), and tissue-resident memory CD8⁺ T cells (T_{RM})³¹. To investigate the differentiation of CD8⁺ T_{SCM} cells in *Rag1*^{-/-} mice, we analyzed immune cell phenotypes in the colon of *Rag1*^{-/-} recipient mice. Our results showed that the populations of CD8⁺ T_{SCM} and CD8⁺ T_{CM} cells were significantly increased in the colon of *Rag1*^{-/-} mice received CD8⁺ T_{SCM} cells from stressed mice, whereas the population of CD8⁺ T_{EM} cells decreased. Importantly, these changes were reversed by treatment with the CXCR2 inhibitor SB265610 (**Figure 4J** and **Response Document Figure 6A-B**). Additionally, there was no significant difference in the CD8⁺ T_{RM} cell population in the colon of *Rag1*^{-/-} mice received pathological CD8⁺ T_{SCM} cells (**Response Document Figure 6C**). Previous studies have shown that both CD8⁺ T_{CM} and CD8⁺ T_{EM} express IFN- γ and TNF- α , but CD8⁺ T_{EM} exhibit superior cytotoxic potential compared to CD8⁺ T_{CM}³²⁻³⁴. In humans, cytotoxic effector molecules such as granzyme B and perforin, stored in lysosomal granules, endow CD8⁺ T_{EM} with immediate cytotoxic function upon degranulation^{34,35}. CD8⁺ T_{RM} maintain transcripts of effector molecules, including IFN- γ and TNF- α , in a readily deployable state to ensure rapid production and release of these pro-inflammatory mediators³⁶. The CD8⁺ T_{RM} population in the small intestine retains protein expression of granzyme B, indicating direct

cytotoxic function^{31,37}. Therefore, we do not rule out that CD8⁺ T_{SCM} cells may infiltrate the intestinal parenchyma and subsequently differentiate into functionally distinct subsets to mediate the microbiota regulation.

Critique 2. The authors identified a few microbiota components that are altered in Rag1^{-/-} mice receiving CD8⁺ Tscm cells from stress-susceptible mice in Figure 6. Is such a change in overall microbiota, the consortium of those changed bacteria, or a single bacteria sufficient to drive depressive behavior in mice? Does CSDS induce similar changes in WT mice?

Response: We express our gratitude for the reviewer's valuable suggestion, which prompted us to enhance our analysis of microbiota alterations induced by pathological CD8⁺ T_{SCM} cells. To address whether these changes—either collectively, as a consortium, or individually—are sufficient to drive depressive behavior, and whether similar shifts occur in WT mice under CSDS, we conducted new set of metagenomic sequencing of control and CSDS mice. Comparing *Rag1^{-/-}* mice receiving CD8⁺ T_{SCM} cells from CSDS mice and control mice, we identified 137 bacterial species with consistent changes, of which nine were involved in tyrosine metabolism (**Figure 6A-B**).

Using Spearman correlation analyses, we explored the relationships between the abundance of these nine tyrosine metabolism-related microbiota, HVA levels, and depressive behaviors in *Rag1^{-/-}* recipient mice. Four species, *Bifidobacterium scardovii* (*B. scardovii*), *Alistipes indistinctus*, *Mucinivorans hirudinis*, and *Marinilabiliales bacterium*, exhibited correlations consistent with improved behavioral outcomes, showing positive associations with sucrose preference in SPT and negative associations with immobility times in TST and FST (**Figure 6C**). Other four microbiota (*Bifidobacterium cuniculi*, *Bacteroidota bacterium*, *Chitinophaga terrae*, and *Alloprevotella sp. oral taxon 473*) have one or two correlations with depressive behavior (**Figure 6C**). Notably, *B. scardovii* was the only species positively correlated with HVA levels (**Figure 6C** and **Supplementary Table 18**), suggesting that its reduced abundance may contribute to lower HVA concentrations and worsened depressive phenotypes in stressed *Rag1^{-/-}* mice.

To confirm *B. scardovii*'s role in HVA production, we cultured it *in vitro* using DSMZ medium 104, verifying its ability to synthesize HVA (**Figure 6F**). We then evaluated its therapeutic potential by administering *B. scardovii* (1×10⁹ CFU/day) to CSDS mice for 10 days, observing significant reductions in immobility times in the TST and FST, alongside increased sucrose preference (**Figure 6G-J**). Additionally, direct HVA administration to CSDS mice elevated HVA levels in both plasma and brain tissues (**Figure 6K-L**). These results highlight *B. scardovii*'s capacity to mitigate depressive-like behaviors, supporting a mechanistic link between specific microbiota, HVA production, and potential therapeutic interventions for depression in mouse models.

Since we performed new metagenomic sequencing in CSDS mice and control mice, and identified specific bacteria by intersecting these datasets, we observed that the relative abundance of *B. scardovii* was reduced in the gut of CSDS mice as well as in

Rag1^{-/-} mice subjected to adoptive transfer of CSDS-derived CD8⁺ T_{SCM} cells (**Figure 6D-E**). This suggests that CSDS induces similar changes in WT mice.

Critique 3. The authors showed that transferred CD8⁺ T_{scm} cells from stress-susceptible mice preferentially migrate into the gut and blocking the PPBP/CXCR2 axis with a drug prevents the migration (Fig. 4C-G) and decreases the depressive phenotype (Fig. 7A-C). Since PPBP, aka CXCL7, is secreted by CD8⁺ T_{scm} cells while the receptor CXCR2 is expressed in the gut, how could this signaling pathway lead to the migration of CD8⁺ T_{scm} cells into the gut? Does PPBP initiate a signaling pathway in CD8⁺ T_{scm} cells as well? Also, as systemic blocking the PPBP/CXCR2 axis with SB265610 could impact cells other than T cells (such as neutrophils), a cell type-specific knockout mouse model is required to support the authors' conclusion. Also, in Figure 4F, the authors show that PPBP is higher in CD8⁺ T_{scm} cells compared to other T cells. Is PPBP is higher in CD8⁺ T_{scm} cells from stress-susceptible mice than in resilient mice?

Response: We appreciate this critique and would like to address the concerns raised. In this study, we demonstrated that increased PPBP secreted by pathological CD8⁺ T_{SCM} cells from CSDS mice interacts with CXCR2 in intestinal endothelial cells, triggering a signaling cascade that enhances endothelial permeability and facilitates immune cell recruitment. This conclusion is supported by multiple lines of experimental evidence: **a)** We identified that CD8⁺ T_{SCM} cells exhibit specific overexpression of PPBP compared to other T-cell subsets (**Figure 4F**). Given that intestinal endothelial cells express CXCR2, the receptor of PPBP, the PPBP-CXCR2 axis mediates the recruitment of CD8⁺ T_{SCM} cells, particularly depression-derived CD8⁺ T_{SCM} cells, to the intestine in healthy mice. **b)** This mechanism was further validated *in vivo* by our finding that the CXCR2 inhibitor-SB265610 significantly reduced the intestinal migration of adoptive transferred pathological CD8⁺ T_{SCM} cells from CSDS mice in *Rag1*^{-/-} mice, as demonstrated through immunofluorescence staining and flow cytometry analysis (**Figure 4I-J**). Additionally, pathological CD8⁺ T_{SCM} cells from CUS mice increased overall histology score for mucosal damage and reduced expression of tight junction proteins (Occludin, ZO-1, and Claudin-5) in the colon (**Supplementary Figure 11D-E**). These findings collectively reinforce the critical role of the PPBP-CXCR2 pathway in mediating the recruitment of CD8⁺ T_{SCM} cells and intestinal damage driven by pathological CD8⁺ T_{SCM} cells. **c)** We additionally employed an *in vitro* transwell system, which freshly isolated CD8⁺ T_{SCM} cells from control or CSDS-treated mice were fluorescently labeled by CellTracker Green for visualization and seeded in the upper chamber, and intestinal microvascular endothelial cells (IMECs) were cultured to confluence on permeable supports to assess CD8⁺ T_{SCM} cell migration (**Figure 4K**). We demonstrated that CD8⁺ T_{SCM} cells from CSDS mice migrated more robustly toward IMECs than the CD8⁺ T_{SCM} cells from normal mice. Importantly, PPBP blockade, achieved through either pharmacological inhibition of CXCR2 with SB265610 or lentiviral-mediated PPBP knockdown in CD8⁺ T_{SCM} cells from CSDS mice, significantly reduced the migration of these cells (**Figure 4L-M** and **Supplementary Figure 10**) These data confirm that PPBP-CXCR2 is essential for directing CD8⁺ T_{SCM} cell migration to the colon.

To address concerns about off-target effects of systemic CXCR2 inhibition, we examined neutrophil populations in *Rag1*^{-/-} mice receiving control or pathological CD8⁺ T_{SCM} cells. Although the proportion of neutrophils was elevated in the blood of CSDS mice (**Response Document Figure 2A**), no significant differences in neutrophil levels were observed between groups, and pharmacological inhibition of CXCR2 (SB265610) treatment did not alter neutrophil numbers (**Response Document Figure 2B**). These results indicate that the PPBP-CXCR2 axis primarily regulates CD8⁺ T_{SCM} cell behavior rather than affecting other immune cell populations.

Furthermore, we confirmed that PPBP expression in CD8⁺ T_{SCM} cells is significantly elevated in stress-susceptible mice compared to resilient mice (**Figure 4G**). This upregulation of PPBP in susceptible mice likely drives enhanced recruitment of CD8⁺ T_{SCM} cells by binding to CXCR2 in intestinal endothelial cells to the gut, amplifying inflammation and gut dysfunction.

Critique 4. In Fig.4C-D, transferred CD8+ Tscm cells could also be vaguely seen in the brain area as the resolution may not be high enough. Because cytokines produced from the meninges have been found to influence brain function, it would be helpful to examine CD8+ Tscm cells in the meninges and other tissues (such as the intestine) by flow cytometry to confirm the authors' conclusion, though the authors have tried to do this by immunofluorescence (Fig.4G).

Response: We sincerely appreciate the reviewer's suggestion to further validate the distribution of CD8⁺ T_{SCM} cells in the meninges and other tissues (such as the intestine) using flow cytometry. In response to this valuable feedback, we have now conducted additional experiments to address these concerns. First, we compared the proportion of CD8⁺ T_{SCM} cells in the meninges and colon between control and CSDS mouse models using flow cytometry. The results showed that no CD8⁺ T_{SCM} cells were detected in the meninges of either group (**Response Document Figure 7A-B**), in contrast, the proportion of CD8⁺ T_{SCM} cells in the colon was significantly increased in CSDS mice compared to controls, highlighting tissue-specific migration patterns (**Figure 4H**). Second, we performed flow cytometric analysis in the meninges and colon of the *Rag1*^{-/-} mice that received CD8⁺ T_{SCM} cells from control or CSDS mice, and the *Rag1*^{-/-} mice with pathological CD8⁺ T_{SCM} cells that treated with SB265610 (**Response Document Figure 8A**). Our results demonstrated that no CD8⁺ T_{SCM} cells were detected in the meninges (**Response Document Figure 8B**), whereas these cells were observed in the colon (**Figure 4J**).

The meningeal flow cytometry analysis was performed as follows: we carefully excised the dural meninges and digested them with 1 mg/mL collagenase VIII (C2139, Sigma-Aldrich) and 0.5 mg/mL DNase I (18047019, Invitrogen) in DMEM supplemented with 2% FBS and 1% penicillin/streptomycin at 37°C for 15 minutes. Digestion was terminated by adding 1 mL of DMEM with 10% FBS, and single-cell suspensions were prepared by gently passing the digested tissue through a 70 μm filter. Cells were then pelleted by centrifugation at 1500 rpm for 10 minutes at 4°C, and the fluorescence of EGFP-labeled cells was analyzed using a BD FACSCelesta™ Flow Cytometer (BD Biosciences, USA).

These new findings, combined with our previous immunofluorescence data (**Figure 4A** and **Figure 4I**), reinforce our conclusion that CD8⁺ T_{SCM} cells preferentially

migrate to the gut rather than the meninges, thereby supporting the specificity of our observations and addressing the reviewer's concerns with robust experimental evidence.

Critique 5. The authors proposed that HVA could reduce depressive behavior via acting on microglia and astrocytes (Fig.3C,F; Fig.7D,G). However, there is no supporting data showing that changes in microglia and astrocytes are sufficient to alter depressive behavior. Also, a recent study has shown that HVA alleviated depression by modulating synaptic integrity (10.1016/j.cmet.2024.03.010). How could the author reconcile their findings with this study and differentiate the relative contributions of the two different pathways?

Response: Thank you for raising this valid point. Growing evidence shows that impairments in the normal structure and function of glial cells in the context of neuroinflammation can lead to depression. Astrocytes are key players that sense homeostatic disturbances in the CNS, and astrocyte dysfunction is evident in stressed mice³⁸. Microglia play an important role in immune surveillance and are known to be actively involved in various neurologic pathologies³⁹. Microglial activation with subsequent inflammatory cytokine release also mediates the effects of stress; these processes have been implicated as a major trigger of depression in both human and animal models⁴⁰. At different stages of depression, the functional status of microglia is diverse, and they exert different regulatory effects on neurons in different functional states⁴¹.

As the reviewer's suggestion, we examined the synaptic integrity in CSDS mice treated with HVA. The CSDS treatment significantly decreased the spine numbers in hippocampus, and all of these phenotypes were ameliorated by HVA treatment (**Figure 8K-L**). Moreover, HVA treatment significantly ameliorated the decreased levels of synapse-associated protein-Synaptophysin and mature brain-derived neurotrophic factor (mBDNF) protein expression induced by CSDS (**Supplementary Figure 15B**). The HVA treatment also alleviated the astrocyte dysfunction and microglial activation induced by CSDS (**Figure 8A-J**). Therefore, we suggested the HVA alleviated neuroinflammation and promoted brain plasticity with concomitant improved depressive symptom.

Critique 6. In Fig.2C, the authors focused on CD8+ Tscm cells based on DEG between HC and MDD. How were those genes selected and what led to the authors' focus on this population? Given that CD8+ Tscm cells are the core finding of the manuscript, further evidence should be provided to support it. For example, does this population increase in the scRNA-seq study? Do depressed mice have more of this population in the blood, the intestine, or meninges?

Response: We thank the reviewer for the suggestion. We identified 33 DEGs in CD8⁺ T_{SCM} cells through comparative analysis of scRNA-seq data between MDD patients and healthy controls, as shown in **Figure 2C** (the gene set represents a collection of DEGs from various T cell subtypes, with the heatmap illustrating the expression abundance of these DEGs across different cell types) and further detailed in new **Supplementary Figure 4A** (which specifically shows DEGs in CD8⁺ T_{SCM} cells). These genes were

selected based on statistically significant differences in expression at the single-cell level, providing direct evidence of transcriptional changes specific to CD8⁺ T_{SCM} cells in MDD patients. Notably, the majority of these DEGs were predominantly derived from CD8⁺ T_{SCM} cells. Furthermore, considering the correlation between this cell population and the clinical symptoms of MDD patients, these findings led to our focus on this population.

Additionally, the population of CD8⁺ T_{SCM} cells was significantly increased in the scRNA-seq study (**Figure 2D**), and the proportion of CD8⁺ T_{SCM} cells was significantly increased in both blood (**Figure 2N**) and colon (**Figure 4J**) from stressed mice. However, no CD8⁺ T_{SCM} cells were detected in the meninges by flow cytometry (**Response Document Figure 7A-B**).

Let us take this opportunity to thank the reviewer for the excellent guidance about how to improve our study and the manuscript generally. We trust you'll agree that our work is much improved in light of the work we conducted under your direction. Thank you.

References:

- 1 Chin, C. H. *et al.* cytoHubba: identifying hub objects and sub-networks from complex interactome. *BMC systems biology* **8 Suppl 4**, S11 (2014). <https://doi.org/10.1186/1752-0509-8-S4-S11>
- 2 Pinero, J. *et al.* DisGeNET: a comprehensive platform integrating information on human disease-associated genes and variants. *Nucleic Acids Res* **45**, D833-D839 (2017). <https://doi.org/10.1093/nar/gkw943>
- 3 Nagy, C. *et al.* Single-nucleus transcriptomics of the prefrontal cortex in major depressive disorder implicates oligodendrocyte precursor cells and excitatory neurons. *Nat Neurosci* **23**, 771-781 (2020). <https://doi.org/10.1038/s41593-020-0621-y>
- 4 Liu, Z. *et al.* Advanced oxidation protein products induce microglia-mediated neuroinflammation via MAPKs-NF-kappaB signaling pathway and pyroptosis after secondary spinal cord injury. *Journal of neuroinflammation* **17**, 90 (2020). <https://doi.org/10.1186/s12974-020-01751-2>
- 5 Dang, R. *et al.* Activation of angiotensin-converting enzyme 2/angiotensin (1-7)/mas receptor axis triggers autophagy and suppresses microglia proinflammatory polarization via forkhead box class O1 signaling. *Aging cell* **20**, e13480 (2021). <https://doi.org/10.1111/accel.13480>
- 6 Liang, B. *et al.* 1,2-Dichloroethane induces apoptosis in the cerebral cortexes of NIH Swiss mice through microRNA-182-5p targeting phospholipase D1 via a mitochondria-dependent pathway. *Toxicology and applied pharmacology* **430**, 115728 (2021). <https://doi.org/10.1016/j.taap.2021.115728>
- 7 Roque, C., Mendes-Oliveira, J. & Baltazar, G. G protein-coupled estrogen receptor activates cell type-specific signaling pathways in cortical cultures: relevance to the selective loss of astrocytes. *Journal of neurochemistry* **149**, 27-40 (2019). <https://doi.org/10.1111/jnc.14648>
- 8 Guo, F. *et al.* Chemokine CCL2 contributes to BBB disruption via the p38 MAPK signaling pathway following acute intracerebral hemorrhage. *FASEB J* **34**, 1872-1884 (2020). <https://doi.org/10.1096/fj.201902203RR>
- 9 Coelho-Santos, V. *et al.* The TNF-alpha/NF-kappaB signaling pathway has a key role in methamphetamine-induced blood-brain barrier dysfunction. *J Cereb Blood Flow Metab* **35**, 1260-1271 (2015). <https://doi.org/10.1038/jcbfm.2015.59>
- 10 Yoon, H. S. *et al.* Relationships of Cerebrospinal Fluid Monoamine Metabolite Levels With Clinical Variables in Major Depressive Disorder. *J Clin Psychiatry* **78**, e947-e956 (2017). <https://doi.org/10.4088/JCP.16m11144>
- 11 Reddy, P. L., Khanna, S., Subhash, M. N., Channabasavanna, S. M. & Rao, B. S. CSF amine metabolites in depression. *Biol Psychiatry* **31**, 112-118 (1992). [https://doi.org/10.1016/0006-3223\(92\)90198-9](https://doi.org/10.1016/0006-3223(92)90198-9)
- 12 Mitani, H., Shirayama, Y., Yamada, T. & Kawahara, R. Plasma levels of homovanillic acid, 5-hydroxyindoleacetic acid and cortisol, and serotonin turnover in depressed patients. *Prog Neuropsychopharmacol Biol Psychiatry* **30**, 531-534 (2006). <https://doi.org/10.1016/j.pnpbp.2005.11.021>
- 13 Zhao, M. *et al.* Gut bacteria-driven homovanillic acid alleviates depression by modulating synaptic integrity. *Cell Metab* **36**, 1000-1012 e1006 (2024). <https://doi.org/10.1016/j.cmet.2024.03.010>
- 14 Grace, A. A. Dysregulation of the dopamine system in the pathophysiology of schizophrenia

- and depression. *Nat Rev Neurosci* **17**, 524-532 (2016). <https://doi.org:10.1038/nrn.2016.57>
- 15 Delva, N. C. & Stanwood, G. D. Dysregulation of brain dopamine systems in major depressive disorder. *Exp Biol Med (Maywood)* **246**, 1084-1093 (2021). <https://doi.org:10.1177/1535370221991830>
- 16 Mousten, I. V., Sorensen, N. V., Christensen, R. H. B. & Benros, M. E. Cerebrospinal Fluid Biomarkers in Patients With Unipolar Depression Compared With Healthy Control Individuals: A Systematic Review and Meta-analysis. *JAMA Psychiatry* **79**, 571-581 (2022). <https://doi.org:10.1001/jamapsychiatry.2022.0645>
- 17 Mendels, J., Frazer, A., Fitzgerald, R. G., Ramsey, T. A. & Stokes, J. W. Biogenic amine metabolites in cerebrospinal fluid of depressed and manic patients. *Science* **175**, 1380-1382 (1972). <https://doi.org:10.1126/science.175.4028.1380>
- 18 Cai, R. *et al.* Panoptic imaging of transparent mice reveals whole-body neuronal projections and skull-meninges connections. *Nat Neurosci* **22**, 317-327 (2019). <https://doi.org:10.1038/s41593-018-0301-3>
- 19 Park, L. M., Lannigan, J. & Jaimes, M. C. OMIP-069: Forty-Color Full Spectrum Flow Cytometry Panel for Deep Immunophenotyping of Major Cell Subsets in Human Peripheral Blood. *Cytometry A* **97**, 1044-1051 (2020). <https://doi.org:10.1002/cyto.a.24213>
- 20 Yan, B. *et al.* Association of aging related genes and immune microenvironment with major depressive disorder. *J Affect Disord* **369**, 706-717 (2025). <https://doi.org:10.1016/j.jad.2024.10.053>
- 21 Yazar, S. *et al.* Single-cell eQTL mapping identifies cell type-specific genetic control of autoimmune disease. *Science* **376**, eabf3041 (2022). <https://doi.org:10.1126/science.abf3041>
- 22 Zhu, L. *et al.* Single-Cell Sequencing of Peripheral Mononuclear Cells Reveals Distinct Immune Response Landscapes of COVID-19 and Influenza Patients. *Immunity* **53**, 685-696 e683 (2020). <https://doi.org:10.1016/j.immuni.2020.07.009>
- 23 Zhu, H. *et al.* Human PBMC scRNA-seq-based aging clocks reveal ribosome to inflammation balance as a single-cell aging hallmark and super longevity. *Sci Adv* **9**, eabq7599 (2023). <https://doi.org:10.1126/sciadv.abq7599>
- 24 Witkowski, M. T. *et al.* Extensive Remodeling of the Immune Microenvironment in B Cell Acute Lymphoblastic Leukemia. *Cancer Cell* **37**, 867-882 e812 (2020). <https://doi.org:10.1016/j.ccell.2020.04.015>
- 25 Giordano, G., Cole, T. B., Furlong, C. E. & Costa, L. G. Paraoxonase 2 (PON2) in the mouse central nervous system: a neuroprotective role? *Toxicol Appl Pharmacol* **256**, 369-378 (2011). <https://doi.org:10.1016/j.taap.2011.02.014>
- 26 Ferraiuolo, L. *et al.* Dysregulation of astrocyte-motoneuron cross-talk in mutant superoxide dismutase 1-related amyotrophic lateral sclerosis. *Brain* **134**, 2627-2641 (2011). <https://doi.org:10.1093/brain/awr193>
- 27 Zhao, Y. *et al.* Human CD8 T-stem cell memory subsets phenotypic and functional characterization are defined by expression of CD122 or CXCR3. *Eur J Immunol* **51**, 1732-1747 (2021). <https://doi.org:10.1002/eji.202049057>
- 28 Hosokawa, K. *et al.* Memory Stem T Cells in Autoimmune Disease: High Frequency of Circulating CD8+ Memory Stem Cells in Acquired Aplastic Anemia. *J Immunol* **196**, 1568-1578 (2016). <https://doi.org:10.4049/jimmunol.1501739>
- 29 Hong, H. *et al.* The Distribution of Human Stem Cell-like Memory T Cell in Lung Cancer. *J*

- Immunother* **39**, 233-240 (2016). <https://doi.org:10.1097/CJI.0000000000000128>
- 30 Collins, S. M. A role for the gut microbiota in IBS. *Nat Rev Gastroenterol Hepatol* **11**, 497-505
(2014). <https://doi.org:10.1038/nrgastro.2014.40>
- 31 Parga-Vidal, L. & van Gisbergen, K. Area under Immunosurveillance: Dedicated Roles of
Memory CD8 T-Cell Subsets. *Cold Spring Harb Perspect Biol* **12** (2020).
<https://doi.org:10.1101/cshperspect.a037796>
- 32 Wolint, P., Betts, M. R., Koup, R. A. & Oxenius, A. Immediate cytotoxicity but not
degranulation distinguishes effector and memory subsets of CD8+ T cells. *J Exp Med* **199**, 925-
936 (2004). <https://doi.org:10.1084/jem.20031799>
- 33 Wherry, E. J. *et al.* Lineage relationship and protective immunity of memory CD8 T cell subsets.
Nat Immunol **4**, 225-234 (2003). <https://doi.org:10.1038/ni889>
- 34 Appay, V., van Lier, R. A., Sallusto, F. & Roederer, M. Phenotype and function of human T
lymphocyte subsets: consensus and issues. *Cytometry A* **73**, 975-983 (2008).
<https://doi.org:10.1002/cyto.a.20643>
- 35 van Aalderen, M. C. *et al.* Label-free Analysis of CD8(+) T Cell Subset Proteomes Supports a
Progressive Differentiation Model of Human-Virus-Specific T Cells. *Cell Rep* **19**, 1068-1079
(2017). <https://doi.org:10.1016/j.celrep.2017.04.014>
- 36 Hombrink, P. *et al.* Programs for the persistence, vigilance and control of human CD8(+) lung-
resident memory T cells. *Nat Immunol* **17**, 1467-1478 (2016). <https://doi.org:10.1038/ni.3589>
- 37 Masopust, D., Vezys, V., Wherry, E. J., Barber, D. L. & Ahmed, R. Cutting edge: gut
microenvironment promotes differentiation of a unique memory CD8 T cell population. *J*
Immunol **176**, 2079-2083 (2006). <https://doi.org:10.4049/jimmunol.176.4.2079>
- 38 Durkee, C., Kofuji, P., Navarrete, M. & Araque, A. Astrocyte and neuron cooperation in long-
term depression. *Trends Neurosci* **44**, 837-848 (2021).
<https://doi.org:10.1016/j.tins.2021.07.004>
- 39 Prinz, M., Jung, S. & Priller, J. Microglia Biology: One Century of Evolving Concepts. *Cell*
179, 292-311 (2019). <https://doi.org:10.1016/j.cell.2019.08.053>
- 40 Yirmiya, R., Rimmerman, N. & Reshef, R. Depression as a microglial disease. *Trends Neurosci*
38, 637-658 (2015). <https://doi.org:10.1016/j.tins.2015.08.001>
- 41 Wang, H. *et al.* Microglia in depression: an overview of microglia in the pathogenesis and
treatment of depression. *J Neuroinflammation* **19**, 132 (2022). [https://doi.org:10.1186/s12974-
022-02492-0](https://doi.org:10.1186/s12974-022-02492-0)

Response Document Figures and legends

Response Document Figure 1

Response Document Figure 1 Dopamine level in the brain tissues. (A) The level of dopamine measured by ELISA in the brain tissues of control, CSDS and HVA treated-CSDS mice. $n = 6$ /group. (B) The level of dopamine measured by ELISA in the brain tissues of *Rag1*^{-/-} recipient mice. $n = 6$ /group. CON: control, CSDS: chronic social defeat stress, CD8⁺ T_{SCM} cells: stem cell-like memory CD8⁺ T cells, HVA: homovanillic acid.

Response Document Figure 2

Response Document Figure 2 Neutrophils in peripheral blood and intestine. (A) The proportion of neutrophils in blood samples from control and CSDS mice detected by flow cytometry. $n = 14/\text{group}$. **(B)** The immunostaining images and quantification of neutrophils in colon of *Rag1*^{-/-} recipient mice. $n = 4/\text{group}$. CON: control, CSDS: chronic social defeat stress, CD8⁺ T_{SCM} cells: stem cell-like memory CD8⁺ T cells.

Response Document Figure 3

Response Document Figure 3 The marker genes of macrophages. (A) Myeloid cells were extracted from scRNA-seq data and subjected to re-clustering. **(B)** Previously reported markers were utilized to annotate the cell types. **(C)** Expression levels of specific marker genes in the macrophages were overlaid on the t-SNE representation.

Response Document Figure 4

Response Document Figure 4 The RNA-seq on brain tissue from control mice, mice subjected to CSDS, and CSDS mice treated with HVA. **(A)** Volcano plot showed the DEGs between the CSDS and control groups. **(B)** Volcano plot showed the DEGs between the CSDS and HVA-treated CSDS groups. **(C)** The Venn Diagram showed 41 genes that were altered across both comparisons. CON: control, CSDS: chronic social defeat stress, HVA: homovanillic acid.

Response Document Figure 5

Response Document Figure 5 The T-cell receptor (TCR) repertoires of CD8⁺ T_{SCM} cells. (A) Schematic showing the experimental design for the TCR repertoires of CD8⁺ T_{SCM} cells from control, susceptible and resilient mice. **(B)** The clonality, shannon index, inverse simpson index, simpson index, or DE50 of TCR- α repertoires in CD8⁺ T_{SCM} cells. $n = 3$ /group. **(C)** The clonality, shannon index, inverse simpson index, simpson index, or DE50 of TCR- β repertoires in CD8⁺ T_{SCM} cells. $n = 3$ /group. CON: control, SUS: susceptible, RES: resilient, CD8⁺ T_{SCM} cells: stem cell-like memory CD8⁺ T cells.

Response Document Figure 6

Response Document Figure 6 The differentiation of CD8⁺ T_{SCM} cells in *Rag1*^{-/-} recipient mice. (A-C) The populations of CD8⁺ T_{CM} cells (A), CD8⁺ T_{EM} cells (B), and CD8⁺ T_{RM} (C) cells in the colon of *Rag1*^{-/-} recipient mice. $n = 10-12/\text{group}$. $*P < 0.05$ versus the control CD8⁺ T_{SCM} cells group, $\#P < 0.05$ and $###P < 0.05$ versus the CSDS CD8⁺ T_{SCM} cells group using one-way ANOVA. CON: control, CSDS: chronic social defeat stress, CD8⁺ T_{SCM} cells: stem cell-like memory CD8⁺ T cells, CD8⁺ T_{CM} cells: central memory CD8⁺ T cells, CD8⁺ T_{EM} cells: effector memory CD8⁺ T cells, CD8⁺ T_{RM} cells: tissue-resident memory CD8⁺ T cells.

Response Document Figure 7

Response Document Figure 7 The proportion of CD8⁺ T_{SCM} cells in the meninges of control and CSDS mice. (A) Schematic showing the experimental design for the proportion of CD8⁺ T_{SCM} cells in the meninges. **(B)** There were no CD8⁺ T_{SCM} cells were detected in the meninges of either control or CSDS mice. CON: control, CSDS: chronic social defeat stress, CD8⁺ T_{SCM} cells: stem cell-like memory CD8⁺ T cells, CD8⁺ T_{CM} cells: central memory CD8⁺ T cells, CD8⁺ T_{EM} cells: effector memory CD8⁺ T cells, CD8⁺ T_{RM} cells: tissue-resident memory CD8⁺ T cells, CD8⁺ T_{Naive} cells: naive CD8⁺ T cells.

Response Document Figure 8

Response Document Figure 8 The proportion of CD8⁺ T_{SCM} cells in the meninges in *Rag1*^{-/-} recipient mice. (A) Schematic showing the experimental design for the proportion of CD8⁺ T_{SCM} cells in the meninges. **(B)** There were no CD8⁺ T_{SCM} cells were detected in the meninges of the *Rag1*^{-/-} mice that received CD8⁺ T_{SCM} cells from control or CSDS mice, and the *Rag1*^{-/-} mice with pathological CD8⁺ T_{SCM} cells that treated with SB265610. CON: control, CSDS: chronic social defeat stress, CD8⁺ T_{SCM} cells: stem cell-like memory CD8⁺ T cells, CD8⁺ T_{CM} cells: central memory CD8⁺ T cells, CD8⁺ T_{EM} cells: effector memory CD8⁺ T cells, CD8⁺ T_{RM} cells: tissue-resident memory CD8⁺ T cells, CD8⁺ T_{Naïve} cells: naive CD8⁺ T cells.

Point-by-Point Responses

Reviewer #1 (Remarks to the Author):

I appreciate the authors' comprehensive and meticulous responses to my previous comments. The inclusion of additional data and figures has notably strengthened the manuscript and effectively addressed my primary concerns. Overall, I have no significant remaining issues. Only a few minor points require clarification:

1. The authors identified distinct molecular signatures in pathological CD8⁺ T stem cell memory (TSCM) cells, including increased cytotoxic potential evidenced by upregulation of GZMB, GZMH, and TBX21, as well as enhanced chemotactic capabilities indicated by elevated expression of CX3CR1, CCL4, and CCL4L2. It would be valuable for the authors to elaborate on the potential mechanisms by which depressive states may influence or drive the emergence of this specific CD8⁺ TSCM cell subset, especially in contrast to those observed under normative conditions. Exploring the pathways linking mood disorders to immune cell differentiation and function could provide deeper mechanistic insights.

Response: We thank the reviewer for this insightful question regarding the mechanistic links between depressive states and the emergence of pathological CD8⁺ T_{SCM} cells. To address this important point, we performed additional analyses to elucidate the transcriptional programs that may drive the differentiation of these distinct CD8⁺ T_{SCM} populations.

Given that these CD8⁺ T_{SCM} cells appear to be phenotypically and functionally distinct, we hypothesized that their differentiation is regulated by respective transcriptional programs. Single-cell regulatory network inference and clustering (SCENIC)¹ revealed highly cluster-specific regulon activities and non-overlapping transcription factor profiles (**Supplementary Fig. 4A**). CD8⁺ T_{SCM} cells in MDD patients exhibited high activity of several regulons, including XBP1, IRF1, and RUNX3. These findings provide important mechanistic insights into how depressive states may influence CD8⁺ T_{SCM} differentiation. XBP1 is a stress-responsive transcription factor involved in the unfolded protein response (UPR) and has been previously implicated in the formation of CD8⁺ T_{SCM}²⁻⁴. IRF1, an interferon regulatory factor, indicates potential inflammatory signaling that could bridge neuroinflammation with immune cell programming⁵. RUNX3 is critical for CD8⁺ T cell development and effector function⁶, and its heightened activity aligns with the observed cytotoxic phenotype (increased GZMB, GZMH, and TBX21) in these cells.

The differential activation of these transcription factors provides a plausible mechanism connecting mood disorders to the specialized differentiation of CD8⁺ T_{SCM} cells. Stress-induced neurohormonal changes (*e.g.*, elevated cortisol, catecholamines) may directly influence these transcriptional programs, promoting the development of a more cytotoxic and migratory CD8⁺ T_{SCM} phenotype that ultimately contributes to gut dysfunction through the PPBP-CXCR2 axis as demonstrated in our study.

We have revised the description of this data in **line 184-189, page 7** as “Given that these CD8⁺ T_{SCM} cells appear to be phenotypically and functionally distinct, we hypothesized that their differentiation is regulated by respective transcriptional programs. Single-cell regulatory network inference and clustering (SCENIC)¹ revealed highly cluster-specific regulon activities and non-overlapping transcription factor profiles (**Supplementary Fig. 4A**). CD8⁺ T_{SCM} cells in MDD patients exhibited high activity of several regulons, including XBP1, IRF1, and RUNX3.”

2. In Figure 3O, annotating the significantly differentially expressed genes directly within the volcano plot—either through labels or color coding—would greatly improve interpretability and facilitate rapid identification of key genes of interest.

Response: Thank you for the valuable suggestion. To improve the interpretability of the volcano plot in Figure 3O, we have now annotated key differentially expressed genes that are previously reported in the context of major depressive disorder. These genes are labeled directly on the plot to facilitate rapid identification.

3. The terminology “HVA in microbiota” and “Total amount of HVA” used in Figures 5D and 5H are somewhat ambiguous. Clarifying these terms—perhaps by defining what HVA represents (e.g., specifying whether it refers to microbial-derived or host-derived HVA, and detailing the measurement method—either in the figure legends or in the main text, would enhance clarity and reader understanding.

Response: We sincerely apologize for the confusion. The term “Total amount of HVA” in Figure 5 refers specifically to microbial-derived HVA, which was measured in fecal samples. To improve clarity, we have revised the y-axis labels in Figure 5D-5H and the panel labels for Figure 5D-5F to more precisely reflect the measured analytes.

Reviewer #2 (Remarks to the Author):

The authors responded to my concerns raised previously.

Response: We are grateful for the highly insightful comments on our study. Thanks!

Reviewer #3 (Remarks to the Author):

The authors have vigorously addressed the reviewer's concerns and the manuscript is recommended for publication.

Response: We are grateful for the highly insightful comments on our study. Thanks!

References:

- 1 Aibar, S. *et al.* SCENIC: single-cell regulatory network inference and clustering. *Nat Methods* **14**, 1083-1086 (2017). <https://doi.org/10.1038/nmeth.4463>
- 2 Wang, M. & Kaufman, R. J. The impact of the endoplasmic reticulum protein-folding environment on cancer development. *Nat Rev Cancer* **14**, 581-597 (2014). <https://doi.org/10.1038/nrc3800>
- 3 Hermans, D. *et al.* Lactate dehydrogenase inhibition synergizes with IL-21 to promote CD8(+) T cell stemness and antitumor immunity. *Proc Natl Acad Sci U S A* **117**, 6047-6055 (2020). <https://doi.org/10.1073/pnas.1920413117>
- 4 Glembotski, C. C. The role of the unfolded protein response in the heart. *J Mol Cell Cardiol* **44**, 453-459 (2008). <https://doi.org/10.1016/j.yjmcc.2007.10.017>
- 5 Mishra, B. *et al.* IL-10 targets IRF transcription factors to suppress IFN and inflammatory response genes by epigenetic mechanisms. *Nat Immunol* **26**, 748-759 (2025). <https://doi.org/10.1038/s41590-025-02137-3>
- 6 Cruz-Guilloty, F. *et al.* Runx3 and T-box proteins cooperate to establish the transcriptional program of effector CTLs. *J Exp Med* **206**, 51-59 (2009). <https://doi.org/10.1084/jem.20081242>